# Clustering with bandit feedback: breaking down the computation/information gap

**Victor Thuot**                                                    VICTOR.THUOT@INRAE.FR
*INRAE, Mistea, Institut Agro, Univ Montpellier, Montpellier, France.*

**Alexandra Carpentier**                                    CARPENTIER@UNI-POTSDAM.DE
*Institut für Mathematik, Universität Potsdam, Potsdam, Germany.*

**Christophe Giraud**              CHRISTOPHE.GIRAUD@UNIVERSITE-PARIS-SACLAY.FR
*Université Paris-Saclay, Laboratoire de mathématiques d'Orsay, Orsay, France.*

**Nicolas Verzelen**                                          NICOLAS.VERZELEN@INRAE.FR
*INRAE, Mistea, Institut Agro, Univ Montpellier, Montpellier, France.*

**Editors:** Gautam Kamath and Po-Ling Loh

## Abstract

We investigate the Clustering with Bandit feedback Problem (CBP). A learner interacts with an $N$-armed stochastic bandit with $d$-dimensional subGaussian feedback. There exists a hidden partition of the arms into $K$ groups, such that arms within the same group, share the same mean vector. The learner's task is to uncover this hidden partition with the smallest budget - i.e. the least number of observation - and with a probability of error smaller than a prescribed constant $\delta$. In this paper, (i) we derive a non asymptotic lower bound for the budget, and (ii) we introduce the computationally efficient ACB algorithm, whose budget matches the lower bound in most regimes. We improve on the performance of a uniform sampling strategy. Importantly, contrary to the batch setting, we establish that there is no computation-information gap in the bandit setting.

**Keywords:** clustering, bandit theory, pure exploration, information-theoretic bounds, machine learning

## 1. Introduction

We consider a sequential and active clustering problem, the **Clustering with Bandit feedback Problem (CBP)** introduced, for instance in (Yang et al., 2024; Yavas et al., 2024). In this setting, there are $N$ items, represented by a $d$-dimensional mean. At each time $t$, the learner chooses one of the items, and samples it - i.e. obtains a noisy evaluation of the $d$-dimensional mean that characterizes it - until termination of the sampling process at time $\tau$, which we call the budget, and which is chosen by the learner. We assume that the items are clustered into $K$ unknown groups - and two items are in the same group if and only if their (unknown) means are the same. For a prescribed confidence level $\delta$, the aim of the learner is to recover perfectly this clustering, on an event of probability larger than $1-\delta$, and with a final budget $\tau$ that is as small as possible. Clustering problems are ubiquitous in modern data analysis, and CBP arises e.g. in digital marketing, where accurate clustering of the customers is crucial for adapting recommendations to specific groups of customers, and where repeated feedback can be collected online. Since feedback collection is costly, the goal is to recover the clusters with a minimal number $\tau$ of feedback requests. See (Yang et al., 2024) for further motivations.

In the low-dimensional setting, where $K$, $d$ are small, Yang et al. (2024) proves that, when $\delta$ converges to 0, an asymptotic expected budget for perfectly recovering the groups is at most of the order

$$\frac{\sigma^2}{\Delta_*^2} N \log(1/\delta) \ , \tag{1}$$

where $\Delta_*$ is the minimal Euclidean distance between the means, and $\sigma^2$ is the variance of the observations.

**High-dimensional setting.** We consider the high-dimensional setting, where $K$, $d$ can be large, possibly larger than $1/\delta$ or $N$ (for $d$). In the classical clustering setting, where there is no repeated measurements on each item, clustering in high-dimension can be nearly impossible in practice. Indeed, in high-dimension, the best polynomial-time algorithms require a very large separation of the means for successful clustering with no repeated measurements. This requirement has two origins. First, it is difficult to localize the means in high-dimension, making the clustering problem harder when $d$ becomes large compared to $N/K$. Second, a computation-information gap is conjectured (i) for clustering (Lesieur et al., 2016; Even et al., 2024) when $d$ is very large, and (ii) for estimation (Diakonikolas et al., 2017, 2023) in some high-dimensional non-isotropic setting.

For instance, when there is no repeated measurement, that is for vanilla clustering problems where each item is only observed once, clustering a mixture of $N$ isotropic Gaussians with co-variance $I_d$ and balanced size of the groups, in the high-dimensional setting where $d \geq N$ and $K \gg \log(N)$, low-degree polynomial algorithms requires a separation at least $\Delta_*^2 \gtrsim \sigma^2 \sqrt{dK^2/N}$ (see Even et al., 2024, Thm. 1), while a separation $\Delta_*^2 \gtrsim \sigma^2 \sqrt{dK \log(N)/N}$ is enough at the information level (see Even et al., 2024, Thm. 4). This is a strong evidence of a computation-information gap for the problem of clustering isotropic Gaussian mixture in high dimension.

When repeated measurements are possible, let us consider the simple scheme where we sample $T$ times each item. This scheme corresponds to oracle-BOC sampling of (Yang et al., 2024), when the groups have similar sizes, and the clusters are equidistant. Sampling $T$ times each item is equivalent to shrinking the variance from $\sigma^2$ to $\sigma^2/T$, so, applying standard polynomial time algorithms (Giraud and Verzelen, 2019) to the average values for each items, we can recover the clustering in polynomial time with confidence $\delta = 1/N$ when $T \gtrsim \frac{\sigma^2}{\Delta_*^2}\sqrt{dK^2/N}$. A clustering procedure is said to be a batch if it uses one single observation of each item to recover the partition, as it is the case in vanilla clustering problems. The total number of requests of this simple batch algorithm is then

$$\tau = NT \gtrsim N + \frac{\sigma^2}{\Delta_*^2}\sqrt{dK^2N}. \tag{2}$$

This set of results raises two fundamental questions:

1. Can we improve upon the number of requests of the simple batch algorithm, by implementing a more careful sequential design strategy?

2. What is the minimal budget for perfect recovery in high-dimension, and is there a fundamental computation-information gap for clustering with bandit feedback?

**Contributions.** We provide an answer to these two fundamental questions.

1. First, we provide a polynomial-time algorithm that recovers exactly the clustering with probability higher than $1 - \delta$. i.e. balanced case (all groups have a similar size), it has an expected budget of order

$$N + \frac{\sigma^2}{\Delta_*^2}\left[ N \log\left(N/\delta\right) + \sqrt{dKN \log\left(N/\delta\right)}\right], \tag{3}$$

which outperforms the budget (2) required by the simple batch algorithm.

2. Second, we prove that the budget (3) is information-theoretical optimal, meaning that there is no computation-information gap for clustering with bandit feedback in high-dimension, contrary to the classical case with no repeated measurement.

Our results are non-asymptotic in $N$, $K$, $d$, and $\delta$, in order to account for high-dimensional phenomenon, and possible computational barriers –see the discussion for more details. Compared to the asymptotic minimal budget (1) obtained in (Yang et al., 2024) for $\delta \to 0$, an additional term pops up in the non-asymptotic minimal budget (3), which is dominant when $dK > N \log(N/\delta)$. Our algorithm is based on ideas related to sub-sampling, in order to localize in a more efficient way the mean of each group. The possibility of performing sub-sampling enables us to bypass combinatorial problems arising in clustering with no-repeated measurements. Our algorithm has a quasi-linear complexity, and is also order-optimal for all $\delta$, $N$, $K$, and $d$, for a broader family of problems defined below. From a technical perspective, our information-theoretical results use novel techniques as those combine arguments from high-dimensional statistics and from bandit theory.

**Related literature in clustering.** The problem of clustering a mixture of subGaussian is a classical problem, which has lead to a large literature both in statistics and in machine learning (Dasgupta, 1999; Vempala and Wang, 2004; Lesieur et al., 2016; Lu and Zhou, 2016; Diakonikolas et al., 2018; Regev and Vijayaraghavan, 2017; Giraud and Verzelen, 2019; Fei and Chen, 2018; Chen and Yang, 2021; Kwon and Caramanis, 2020; Segol and Nadler, 2021; Romanov et al., 2022; Liu and Li, 2022; Diakonikolas et al., 2023). In low-dimension and for large values of $N$, state-of-the art polynomial-time procedures for recovering the groups have been introduced by (Liu and Li, 2022), and are based on generalisation of higher moments methods –see also (Diakonikolas et al., 2018; Kothari and Steinhardt, 2017). In high-dimension, the best known conditions for exact reconstruction in polynomial-time are based on an SDP relaxation of K-means (Peng and Wei, 2007; Giraud and Verzelen, 2019). For $K = 2$, a simple Lloyd algorithm achieves perfect recovery at the information level (Ndaoud, 2022), thereby establishing the absence of computation-information gap for $K = 2$. For larger $K$, (Lesieur et al., 2016) conjectures a computation-information gap in high-dimension, and (Even et al., 2024) exhibits a low-degree computational barrier for the clustering of a mixture of isotropic Gaussians, when $d \geqslant N$. Some computation-information gaps have also been shown for Statistical-Query algorithms for learning mixture of non-isotropic Gaussian, with unknown covariance, in moderately high-dimension – see Diakonikolas et al. (2017, 2023). In the sequel, we refer to clustering with no repeated measurements as batch clustering.

**Sequential literature related to CBP.** When turning to the sequential learning literature, the CBP belongs to the family of pure exploration problems in the sequential active learning framework. An iconic such problem is the best-arm identification problem – see (Jamieson and Nowak, 2014) for a survey. In this stream of literature, the Thresholding Bandit Problem (TBP) is quite related – see (Chen and Li, 2015; Chen et al., 2014; Locatelli et al., 2016). This is a specific instance of our setting in dimension $d = 1$ and for two groups, i.e. $K = 2$. In this active binary classification problem, the learner aims at finding the arms that have a mean larger than a given threshold (here $d = 1$), and to divide them in $K = 2$ groups. Note that (Katariya et al., 2018) propose a generalisation of these ideas to multiple groups, albeit still in dimension 1. The optimal asymptotic budget $\tau$ for perfect recovery in the TBP is $\Delta_*^{-2} N \log(1/\delta)$ when $\delta$ goes to 0, and there are no computational gaps, see (Tirinzoni and Degenne, 2022) for state of the art results on TBP.

The CBP, first introduced in (Yang et al., 2024), can be seen as a generalisation of the TBP in dimension $d$. This generalisation is highly non-trivial : subtle phenomenons make clustering problems with $d \geq 2$ very different from clustering in dimension 1. (Yang et al., 2024) provides

an algorithm called BOC, which perfectly recovers the groups with probability higher than $1 - \delta$, and which has an expected budget at most of the order (1) in the asymptotic regime where $\delta$ goes to zero. Note that this rate is reminiscent of the TBP (where $d = 1$, $K = 2$). A closer look at the proofs in (Yang et al., 2024) exhibits an exponential dependence of second-order terms (in $\delta$) on $K$, $d$ so that BOC - or at least its current analysis - is effective only in the asymptotic regime, when $K$, $d$ are considered as being constants. In fact, since the oracle version of BOC samples equally all the arms when the clusters are balanced and equidistant, the BOC budget in this case is at least (2) in high-dimension (Even et al., 2024), which is suboptimal. Our non-asymptotic analysis allows to recover the shape of the optimal budget in the so-called high-dimensional regimes where $d$ or $K$ are not considered as constants. Quite recently, Yavas et al. (2024) have extended the analysis of Yang et al. (2024) to other distributions beyond subGaussian ones.

A somewhat related problem was studied in (Yun and Proutière, 2019), in the Stochastic Block Model within the fixed-budget setting. To extract hidden structure, the interaction between pairs of nodes can be sampled several times, in an active manner. The setting is however quite distinct from our work, and is also focusing on the asymptotic regime where $\delta$ goes to 0. In the paper (Ariu et al., 2024), the related problem of clustering items based on binary feedback is studied - but therein, the feedback corresponds to a single coordinate of a chosen vector. In our work, we observe the full $d$-dimensional vector at each time, so that the settings differ. Finally, it is worth mentioning that our problem should not be confused with that of online clustering, for example studied in (Cohen-Addad et al., 2021).

**Outline.** We formally introduce the CBP in Section 2. An information-theoretical lower bound on the minimal budget for exact recovery is established in Section 3. We introduce and analyze our procedure ACB in Section 4. Numerical experiments are provided in Section 5. All the results are discussed in Section 6.

## 2. Setting and notation

**The sequential and active setting.** We consider a set of $N$ arms, indexed by $[N]$. Each arm $a \in [N]$ is associated to an unknown probability distribution $\nu_a$ on $\mathbb{R}^d$. At each time $t$, the learner chooses an arm $A_t \in [N]$ based on the past observations. Conditionally on the chosen arm $A_t$, she receives from the environment a random observation $X_t \in \mathbb{R}^d$, distributed as $\nu_{A_t}$.

For each arm $a \in [N]$, we write $\mu_a \in \mathbb{R}^d$ for the mean of the distribution $\nu_a$. Both in the context of multi-armed bandits, and in the context of clustering, it is common to assume that the distributions are subGaussian.

**Assumption 1 ($\sigma$-subGaussian arm observations)** *For any arm $a \in [N]$, we assume that there exists a symmetric $d \times d$ matrix $\Sigma_a$ such that, (i) $\max_{a \in [N]} \|\Sigma_a\|_{op} \leq \sigma^2$, where $\|.\|_{op}$ is the operator norm; (ii) the coordinates $(E_i)$ of $E = \Sigma_a^{-1/2}[X - \mu_a]$ are independent and fulfills $\mathbb{E}[\exp(tE_i)] \leq \exp(t^2/2)$ for all $t \in \mathbb{R}$.*

**Remark 2** *This assumption encompasses the emblematic settings where the data are Gaussian, and where the data are bounded. If the distributions $(\nu_a)$ are Gaussian, then Assumption 1 holds by e.g. choosing $\Sigma_a$'s to be the covariance matrices, and associate $\sigma$. If the distributions $(\nu_a)_a$ are such that the coordinates are independent and lie in $[0, 1]$, the collection $(\nu_a)$ is $1/4$-subGaussian.*

**The clustering problem with bandit feedback.** As for the vanilla clustering problem, our objective is to partition the set of arms into groups of arms that share the same expectation $\mu_a$. For this purpose, we make the following modeling assumption.

**Assumption 3 (Hidden partition $G^*$ of the arms into $K$ groups)** *Consider $N \geq K \geq 1$. We assume that there exists a partition $G^* = \{G_1^*, \ldots, G_K^*\}$ of $[N]$ into $K$ groups such that any two arms $a$ and $b$ are in the same group if and only if they share the same expectation ($\mu_a = \mu_b$). For notation purpose, we introduce the vectors $\mu(1), \ldots, \mu(K) \in \mathbb{R}^p$ such $\mu(k)$ corresponds to the common expectation in $G_k^*$. Henceforth, $\mu(k)$ is called the center of the group $G_k^*$.*

In CBP, the goal of the learner is to uncover the true partition $G^*$ of the arms, while using as few samples as possible. The learner samples arms sequentially and, when reaching some stopping time $\tau$, she returns a partition $\hat{G}$ of $[N]$ into $K$ groups, which should ideally be equal to $G^*$. More precisely, let $\pi$ be an algorithm for the clustering problem with bandit feedback, also called the strategy of the learner. We write $(\mathcal{F}_t)_{t \geq 0}$ for the filtration $\mathcal{F}_t = \sigma(A_1, X_1, \ldots, A_t, X_t)$. A strategy $\pi$ consists on three rules:

- A **selection rule** that chooses the next arm $A_t$ to sample, based on the previously sampled arms and observations; $A_t$ is $\mathcal{F}_t$-measurable.

- A **stopping rule** that controls when the learner stops sampling the arms, and which quantifies the budget of the strategy. This is modeled by a stopping time $\tau$ with respect to the filtration $(\mathcal{F}_t)_{t \geq 0}$.

- A **recommendation rule**. Once the stopping time $\tau$ is reached, the learner outputs an estimated partition of the arms $\hat{G}$. This partition is $\mathcal{F}_\tau$-measurable.

For an environment $\nu$ and an algorithm $\pi$, we write $\mathbb{P}_{\pi,\nu}$ for the probability induced by the interaction between the algorithm $\pi$ and the environment.

In this paper, we aim at exactly recovering the partition $G^*$ in the fixed confidence setting. While the partition $G^*$ is identifiable, the groups $(G_k^*)$ and the means $\mu(k)$ are identifiable only up to relabelling, i.e. up to a permutation of $[K]$. We denote by $G \sim G'$ two equivalent partitions of $[N]$, i.e. two partitions such that, for some permutation $\rho$ of $[K]$, $G_k = G'_{\rho(k)}$ for all $k \in [K]$. For a fixed confidence level $\delta \in (0, 1)$, and a given set of environments $\mathcal{E}$, a strategy $\pi = \pi(\delta)$ fulfilling

$$\mathbb{P}_{\pi,\nu}(\hat{G} \sim G^*) \geqslant 1 - \delta \ , \tag{4}$$

is said to be $\delta$-PAC (probably approximately correct) on $\mathcal{E}$. We write $\Pi(\delta, \mathcal{E})$ for the family of such $\delta$-PAC strategies for the CBP on $\mathcal{E}$. Our aim is to design a $\delta$-PAC algorithm, whose budget $\tau$ is as small as possible. For a family of environments $\mathcal{E}$, the optimal worst case (average) budget $T^*(\delta, \mathcal{E})$ is defined as

$$T^*(\delta, \mathcal{E}) = \inf_{\pi \in \Pi(\delta, \mathcal{E})} \sup_{\nu \in \mathcal{E}} \mathbb{E}_{\pi,\nu}[\tau] \ . \tag{5}$$

In order to introduce relevant sets of environments $\mathcal{E}$, we introduce two quantities that characterize the difficulty of a clustering problem, let it be batch or active. First, we consider the minimal Euclidean distance between two distinct group centers

$$\Delta_* = \Delta_*(\nu) = \min_{k \neq k'} \|\mu(k) - \mu(k')\| > 0 \ . \tag{6}$$

Intuitively, the smaller $\Delta_*$, the more difficult it is to distinguish the groups and to recover the partition $G^*$. This quantity naturally appears in most clustering works in the batch setting (Dasgupta, 1999; Vempala and Wang, 2004; Giraud and Verzelen, 2019). Besides, we denote $\theta_*$ the balancedness of $G^*$, that is the proportion of arms in the smallest cluster

$$\theta_* = \min_{k \in [K]} \frac{|G_k^*|}{N} \; \in \; \left[ \frac{1}{N}, \frac{1}{K} \right]. \tag{7}$$

When $\theta_* = 1/K$, all the groups $G_k^*$ share the same size, and the partition is balanced.

Consider $\Delta > 0$, and $\theta > 0$, we define the set $\mathcal{E}(\Delta, \theta, \sigma, N, K, d)$ as the family of environments with $N$ arms, divided into $K$ groups as in Assumption 3, with a minimal gap $\Delta_*$ at least $\Delta$, a balancedness $\theta_*$ at least $\theta$, and with $d$-dimensional observations that are $\sigma$-subGaussian – see Assumption 1. Our main aim is to craft polynomial-time algorithms that attain the optimal worst case budget $T^*(\delta, \mathcal{E}(\Delta, \theta, \sigma, N, K, d))$, and to characterize this optimal worst-case budget.

## 3. Lower bound on the budget

We start by establishing a lower bound for the expected budget of any $\delta$-PAC algorithm over $\mathcal{E}(\Delta, \theta, \sigma, N, K, d)$.

**Theorem 4** *There exists a numerical constant $c > 0$, such that we have for any $\sigma > 0$, any $\Delta > 0$, any $d \geq 1$, any $\theta > 0$, any $\delta \in (0, 1/12)$, and any $N \geqslant 2K \geq 4$ such that $\mathcal{E}(\Delta, \theta, \sigma, N, K, d) \neq \emptyset$*

$$T^*(\delta, \mathcal{E}(\Delta, \theta, \sigma, N, K, d)) \geqslant cN + c\frac{\sigma^2}{\Delta^2} \left[ N \log\left( \frac{N}{\delta} \right) + \sqrt{dKN \log\left( \frac{N}{\delta} \right)} \right]. \tag{8}$$

The lower bound in (8) involves three different terms. As in any pure exploration problem, the first term $N$ is necessary because, when $\tau \leqslant N/2$, then the label of at least one arm has to be guessed randomly inducing a constant probability of error for the exact clustering. This term is only relevant for very large $\Delta$ and is not discussed further. The second term is the largest in the low-dimensional regime where $d \leq N \log(N/\delta)/K$, whereas the third one is the largest in the high-dimensional regime where $d \geq N \log(N/\delta)/K$. This dichotomy between low-dimensional and high-dimensional clustering problems also occurs in the batch problem. Together with the results of the next section, we will establish that it is intrinsic here –see the discussion and the proof sketch for further details. Note that (8) does not depend on $\theta$: we establish (8) for environments where $\theta_*$ is close to $1/K$, that is for balanced partitions. In fact, the total budget of our procedures $ACB$ and $ACB^*$ - see below - do not depend on $\theta_*$ except for extremely unbalanced partitions (very small $\theta_*$) so that the lower bound is tight even for mildly unbalanced partitions.

**Proof** [Sketch of proof of Theorem 4] The first two terms in the lower bound (8) - resp. $\frac{\sigma^2}{\Delta^2} N \log\left( \frac{N}{\delta} \right)$ and $\frac{\sigma^2}{\Delta^2} \sqrt{dKN \log\left( \frac{N}{\delta} \right)}$- are proved separately in Lemmas 6 and 7. Regarding the first term, we first observe that it depends neither on $d$, nor on $K$, nor on $\theta$. For the sake of this sketch, we can therefore restrict ourselves to a one-dimensional ($d = 1$) multi-armed bandit setting where each arm has $a \in [N]$ has either mean $\mu_a = 0$ or $\mu_a = \Delta$, so that $K = 2$. For this simplified toy problem, recovering the partition $G^*$ is equivalent to a Thresholding Bandit Problem (TBP), where the goal is to find the set of arms whose mean is higher or equal to $\Delta$. By building upon some ideas introduced in (Cheshire et al., 2020), we establish the lower bound $\frac{\sigma^2}{\Delta^2} N \log\left( \frac{N}{\delta} \right)$. Note that one may easily interpret this quantity using the fact that, for a specific arm, deciphering whether the mean of a specific arm is 0 or $\Delta$ with probability $1 - \delta/N$, one needs to sample it at least $\frac{\sigma^2}{\Delta^2} \log\left( \frac{N}{\delta} \right)$ times.

The proof of the second term is both more challenging and more innovative. Again, for the purpose of this sketch, let us assume that $K = 2$ and $\theta_* = 1/2$. We use a Bayesian approach by putting a Gaussian prior distribution on $\mu(1)$ with variance $d^{-1/2}\Delta I_d$ and by fixing $\mu(2) = -\mu(1)$ so that, with high probability, $\|\mu(2) - \mu(1)\| \geq \Delta$. Introducing this prior distribution on $\mathbb{R}^d$ is instrumental to recover the dependency of the budget on the dimension $d$ of the problem. First, we use the symmetry of the problem to show that the optimal budget is achieved by a strategy $\pi$ which, in expectation, samples all the arms uniformly. Then, we use a series of reduction by first noting that identifying the group of any node $a$ is, in some sense, at least as difficult, as the supervised problem where we would know the group of all the arms, except that of $a$. In turn, we show that tackling this active supervised problem with an uniform strategy $\pi$ is as difficult as tackling a batch supervised learning problem where each arm is sampled $\tau/N$ times. Finally, we craft an impossibility result for the latter problem. We emphasize that there is no computational restriction here, so that the lower bound for uniform sampling strategies is (8), and not the rate (2) which relates to polynomial-time algorithms (Even et al., 2024). $\blacksquare$

## 4. ACB and Upper bound on the budget

To introduce the main ideas underlying our clustering algorithm with bandit feedback, we first assume in the next subsection that $\Delta$, $\theta$, $\sigma$, $N$, $K$ and $d$ are known quantities, and we construct an algorithm, ACB, that is $\delta$-PAC for environments such that $\Delta_* \geqslant \Delta$ and $\theta_* \geqslant \theta$. We introduce our main algorithm, ACB*, adaptive to $\Delta_*$ and $\theta_*$ in Subsection 4.2.

### 4.1. Warm-up: optimal clustering with known $\Delta, \theta$

The main recipe of ACB is to first identify a set $\hat{S}$ of $K$ arms, which are representative of each group, and then, to classify all the arms based on a precise estimation of the means of the $K$ arms in $\hat{S}$. The ACB algorithm built then on two subroutines:

**1-** SRI (Sequential Representatives identification), which constructs a set $\widehat{S}$ that contains, with high probability, exactly one arm for each group, called the representatives of each group. To construct $\widehat{S}$, we use a sequential elimination technique, combined with high-dimensional two-sample tests.

**2-** ADC (Active Distance-based classification), which computes precise estimates of the means of the arms in $\widehat{S}$, and classifies the remaining arms based on minimum estimated distance to the representatives.

**Estimating distances.** In order to detect whether two arms $a$ and $b$ are in the same group, a key ingredient for both SRI and ADC is to get a good estimation of the square distance $\|\mu_a - \mu_b\|^2$ between the means. Computing the empirical means $\hat{\mu}_a$ and $\hat{\mu}_b$ of collected samples of $a$ and $b$, we can estimate $\|\mu_a - \mu_b\|^2$ by $\|\hat{\mu}_a - \hat{\mu}_b\|^2$. Yet, this simple estimator suffers from an unknown bias depending on the noise covariance matrix. This issue can be circumvented in active sampling, by:

**(i)** computing independent empirical means $\hat{\mu}_a$, $\hat{\mu}'_a$, and $\hat{\mu}_b$, $\hat{\mu}'_b$ for the arms $a$ and $b$, based on repeated measurements,

**(ii)** estimating $\|\mu_a - \mu_b\|^2$ with the unbiased estimator

$$\hat{d}_{ab}^2 = \langle \hat{\mu}_a - \hat{\mu}_b, \hat{\mu}'_a - \hat{\mu}'_b \rangle \ . \tag{9}$$

The construction of this estimator belongs to the statistical folklore for the problem of estimating the square norm of the mean of a random vector – see e.g. Carpentier (2015) for a previous occurrence. In the simpler case where the covariance structure would be known, one could instead use the simpler estimator from Collier and Dalalyan (2019).

**SRI subroutine (Sequential Representative Identification).** The core idea underlying the SRI subroutine is to start from a set $S = \{a_0\}$ made of a single arm, chosen uniformly at random, and then to successively sample new arms $a$, and to add them to $S$, if they passes a sequence of tests ensuring that $a$ is not represented in $S$ with high-probability. The sequence of tests checks if $a$ is already represented in $S$, i.e. if $\min_{b \in S} \|\mu_a - \mu_b\|^2 = 0$, by multiply checking if $\min_{b \in S} \hat{d}_{ab}^2 \leq \Delta^2/2$, with a sequence of estimators $\hat{d}_{ab}^2$ based on increasing sample sizes, ensuring increasing confidence. It is based on the call of the REPRESENTEDTEST subroutine described below, where empirical_mean$(a, n)$ refers to the action of sampling $n$ times the $a$-th arm, and computing the empirical mean of the collected samples. This action is performed twice to compute $\hat{\mu}_a$ and $\hat{\mu}_a'$.

---

**Function** RepresentedTest $(a, (\bar{\mu}_b, \bar{\mu}_b')_{b \in S}, \Delta, n)$**:**    ▷ Test if $a$ is represented in $S$
    $\hat{\mu}_a, \hat{\mu}_a' \leftarrow$ empirical_mean$(a, n)$
    **return** IS.TRUE$\left\{ \min_{b \in S} \langle \hat{\mu}_a - \hat{\mu}_b, \hat{\mu}_a' - \hat{\mu}_b' \rangle \leq \frac{\Delta^2}{2} \right\}$

---

More precisely, let us define

$$U := \lceil 8\theta^{-1} \log(8K/\delta) \rceil \quad ; \qquad r := \lceil \log_2(\log(4U/\delta)) \rceil ; \tag{10}$$

$$n_s := \left\lceil c_1 \frac{\sigma^2}{\Delta^2} (2^s + \log(12K)) \vee c_2 \frac{\sigma^2}{\Delta^2} \sqrt{d(2^s + \log(6))} \right\rceil \quad ; \tag{11}$$

$$s_0 := r \wedge \min\{s \geqslant 1; n_s \geqslant 2\} \quad ; \qquad n_{\max} := n_r \vee \left\lceil c_3 \frac{\sigma^2}{\Delta^2} \sqrt{d} \log(2K) \right\rceil \quad , \tag{12}$$

$$T_{\max} = 2K \left( n_{\max} + \sum_{s=s_0+1}^{r} n_s \right) + 2U n_{s_0} + 2U \sum_{s=s_0+1}^{r} \frac{n_s}{2^{s-4}} \quad , \tag{13}$$

with $c_1$, $c_2$, $c_3 > 0$ numerical constants, explicitly provided in the proof of Lemma 25. The SRI procedure successively samples candidate arms $a_u$ at random, and performs a sequence of REPRESENTEDTEST with (roughly) doubling sample size $n_s$ for $s = s_0, s_0 + 1, \ldots$, until either a REPRESENTEDTEST returns TRUE, in which case the arm $a_u$ is rejected (Line 7); or all tests up to $s = r$ have answered FALSE, in which case the arm $a_u$ is added to $S$ (Line 10). The procedure SRI stops when $|S| = K$, or when a maximal budget has been spent ($T_{\max}$ is defined in (13)) and it returns $\widehat{S} = S$. The minimal index $s_0$ ensures that the sample sizes $n_s$ are not smaller than 2.

The sequence of tests is designed in order to use few samples to reject arms already represented in $S$, while wrongly rejecting an unrepresented arm with probability less than 1/2. Indeed, the choice of the sample sizes $n_s$ and $n_{\max}$ ensures that the probability to take a wrong decision at the $s$-th step is smaller than $2^{-s-1}$. Hence, the probability that an arm already represented in $S$ is rightly rejected before step $s$ is at least $1 - 2^{-s}$, leading to a quick rejection with high-probability. In addition, the maximum sample size $n_r$ is chosen large enough, to ensure a vanishingly small probability of (wrongly) not rejecting such an arm. As for unrepresented arms, the probability to wrongly reject an arm $a_u$ not already represented in $S$ is smaller than $\sum_{s \geq 1} 2^{-s-1} = 1/2$, so that, with probability at least $1 - \delta/4$, we need less than $U$ candidate arms to identify one representative of each group.

We provide further guarantees on SRI subroutine in Appendix C, Lemma 25. In particular, if $S$ is the output of SRI applied with parameters $\Delta$ and $\theta$, then with probability larger than $1 - \delta$, (a) $S$ does not contains two arms from the same group. Moreover, if the true parameters $\Delta_*$ and $\theta_*$ are

**Procedure** SRI $(\delta, \Delta, \theta)$: /* Sequential Representative Identification */

  **Result:** $S$ a set of arms

1   Compute $U, r, s_0, n_s, n_{\max}, T_{\max}$ according to (10)–(13) and sample $a_0 \in [N]$

2   Set $S = \{a_0\}$ and $\hat{\mu}_{a_0}, \hat{\mu}'_{a_0} \leftarrow$ empirical_mean$(a_0, n_{\max})$

3   **for** $u = 1, \ldots, U$ **do**

4   $\quad$ Sample $a_u \in [N]$

5   $\quad$ **for** $s = s_0, \ldots, r$ **do**

6   $\quad\quad$ **if** REPRESENTEDTEST$(a_u, (\hat{\mu}_b, \hat{\mu}'_b)_{b \in S}, \Delta, n_s)$ **then**

7   $\quad\quad\quad$ BREAK ;                                        /* reject $a_u$ */

8   $\quad\quad$ **end**

9   $\quad\quad$ **if** $s = r$ **then**                     /* if $a_u$ has passed all tests */

10  $\quad\quad\quad$ $S \leftarrow S \cup \{a_u\}$                          /* Add $a_u$ to $S$ */

11  $\quad\quad\quad$ $\hat{\mu}_{a_u}, \hat{\mu}'_{a_u} \leftarrow$ empirical_mean$(a_u, n_{\max})$           /* Estimate $\mu_{a_u}$ */

12  $\quad\quad$ **end**

13  $\quad$ **end**

14  $\quad$ **if** $|S| = K$ *or budget* $> T_{\max}$ **then**

15  $\quad\quad$ BREAK                                        /* Terminate $u$ loop */

16  $\quad$ **end**

17  **end**

18  **return** $S$                    /* Return a representative for each group */

smaller than $\Delta$, $\theta$, then $S$ contains exactly $K$ arms, with one arm from each group. We also provide an upper bound on the budget used by SRI.

**ADC subroutine (Active Distance-based Classification).** Once a set $\widehat{S} = \{b_1, \ldots, b_K\}$ of representatives of each group has been successfully obtained with SRI, the mean of each group can be precisely estimated, and remaining arms can be classified based on distance estimation $\hat{d}^2_{ab}$ to these means. This classification is performed by the ADC subroutine.

Let us define

$$J := \left\lceil c_4 \frac{\sigma^2}{\Delta^2} L \vee c_5 \frac{\sigma^2}{\Delta^2} \sqrt{\frac{dN}{K} L} \right\rceil \ , \qquad I := \left\lceil c_4 \frac{\sigma^2}{\Delta^2} L \vee c_5 \frac{\sigma^2}{\Delta^2} \sqrt{\frac{dK}{N} L} \right\rceil \ , \qquad (14)$$

with $L = \log(6NK/\delta)$, and $c_4$, $c_5$ two universal constants defined in the proof of Lemma 26. Assume, without loss of generality, that $b_j \in G^*_j$ for all $j \in [K]$. Then, ADC first computes two precise estimations $\hat{\mu}(j), \hat{\mu}'(j)$ of the mean of arms in $G^*_j$ (Line 6), each based on $J$ samples of arm $b_j$. As these mean estimations are the references for the classification, the sample size $J$ is chosen large enough to ensure a small variance. Then, for each arm $a$, two mean estimations $\hat{\mu}_a, \hat{\mu}'_a$ are computed based on $I$ samples, and the arm $a$ is classified Line 11 according to the smallest estimated distance (15). The budget $I$ for individual mean estimation is much smaller than $J$ in high-dimension $d$, with $I = KJ/N$ for $d$ large. This budget ensures yet that the probability of misclassifying an arm is smaller than $\delta/N$. Overall, we prove that as long as the set $S$ obtained in the first step contains exactly one arm from each cluster, then subroutine ADC will provide with high probability a perfect clustering of the arms. We summarize our guarantees on ADC in Lemma 26.

**Procedure** ADC ($\delta, \Delta, S$)**:**          /* Active Distance-based Classification */

1  **if** $|S| \neq K$ **then**
2  |  **return** Null ;                    /* Return Null if size of $S$ is not $K$ */
3  **end**
4  Enumerate $S = \{b_1, \ldots, b_K\}$, and compute $I, J$ according to (14)
5  **for** $j \in [K]$ **do**
6  |  $\hat{\mu}(j), \hat{\mu}'(j) \leftarrow$ empirical_mean($b_j, J$) ;          /* Estimate the centers */
7  |  $\hat{G}_j \leftarrow \{b_j\}$
8  **end**
9  **for** $a \in [n] \setminus S$ **do**
10 |  $\hat{\mu}_a, \hat{\mu}'_a \leftarrow$ empirical_mean($a, I$)
11 |

   Add $a$ to the group $\hat{G}_k$ such that   $k \in \underset{j=1,\ldots,K}{\operatorname{argmin}} \left\langle \hat{\mu}_a - \hat{\mu}(j), \hat{\mu}'_a - \hat{\mu}'(j) \right\rangle$          (15)

12 **end** ;                             /* Classify arm $a$ */
13 **return** $\{\hat{G}_1, \ldots, \hat{G}_K\}$                    /* Return a clustering */

**ACB algorithm.** Combining the SRI and ADC subroutines, we get a simple clustering with bandit feedback algorithm ACB for the case where $\Delta_*$ and $\theta_*$ are known – see Algorithm 1.

| **Algorithm 2** ACB* ($\theta_*$ and $\Delta_*$ unknown) |
| --- |
| **Input:** $\delta$ |
| 1  **for** $l = 0, 1, \ldots$ **do** |
| 2       **for** $p = 0, \ldots, l$ **do** |
| 3           Compute $S_{p,l} \leftarrow \mathrm{SRI}(\delta_l, \Delta_p, \theta_{p,l} \vee \frac{1}{N})$ |
| 4           **if** $|S_{p,l}| = K$ **then** |
| 5               **for** $a \in S_{p,l}$ **do** |
| 6                   $\bar{\mu}_a, \bar{\mu}'_a \leftarrow$ empirical_mean($a, n'_p$) |
| 7               **end** |
| 8               $\hat{\Delta}^2 \leftarrow \inf_{a,b \in S_{p,l}} \langle \bar{\mu}_a - \bar{\mu}_b, \bar{\mu}'_a - \bar{\mu}'_b \rangle$ |
| 9               **return** $\hat{G} = \mathrm{ADC}(\delta/3, 2^{-1/2}\hat{\Delta}, S_{p,l})$ |
| 10          **end** |
| 11      **end** |
| 12 **end** |

| **Algorithm 1** ACB ($\theta_*$ and $\Delta_*$ known) |
| --- |
| **Input:** $\delta, \Delta, \theta$ |
| 1  $\hat{S} \leftarrow \mathrm{SRI}(\delta/2, \Delta, \theta)$ |
| 2  **return** $\hat{G} = \mathrm{ADC}(\delta/2, \Delta, \hat{S})$ |

### 4.2. Main algorithm ACB*

When the parameters $\Delta_*$ and $\theta_*$ are unknown, we cannot rely on a single call to SRI and ADC as in the ACB algorithm. Multiscale calls to SRI are required, for different candidate levels $\Delta_p$ and $\theta_{p,l}$ for $\Delta_*$ and $\theta_*$. These levels, related sample sizes $n'_p$, and confidence levels $\delta_l$ are defined by

$$\Delta_0^2 = \sigma^2[\log(K) + \sqrt{d} + \log\log(6N/\delta)], \qquad \delta_l = \frac{\delta}{6(l+1)^3} \tag{16}$$

$$\theta_{p,l} = \frac{1}{K2^{l-p}}, \quad \Delta_p = \Delta_0\sqrt{\frac{1}{2^p}}, \quad n'_p = \left\lceil c_6\frac{\sigma^2}{\Delta_p^2}\left(\log(3K^2/\delta) + \sqrt{d\log(3K^2/\delta)}\right)\right\rceil, \tag{17}$$

where $c_6$ is a numerical constant, whose value is given in (56).

The main recipe in ACB$^*$, is to scan decreasing candidate values $\Delta_p$ and $\theta_{p,l}$, until we find a scale where SRI returns a set $S_{p,l}$ of cardinality $K$, see Algorithm 2.

Below, we provide upper bounds on $T^*(\delta, \mathcal{E}(\Delta, \theta, \sigma, N, K, d))$ for both ACB and ACB$^*$. We write $\tau_{ACB}$ and $\tau_{ACB^*}$ for the budget of the non-adaptive procedure $ACB(\delta, \Delta, \theta)$, and of the adaptive one $ACB^*(\delta)$. Define the quantities

$$A = \frac{\sigma^2}{\Delta^2}\left[N\log(N/\delta) + \sqrt{dNK\log(N/\delta)} + \sqrt{d}\frac{\log(K)}{\theta}\right]$$

$$B = \frac{1}{\theta}\log(K/\delta) + \frac{\sigma^2}{\Delta^2}\frac{1}{\theta}\log\left(\frac{K}{\delta}\right)\left[\sqrt{d} + \log\log(N/\delta)\right]\ .$$

**Theorem 5** *Let $\delta > 0$. Let $\Delta > 0$, $\theta > 0$ be any two parameters such that $\mathcal{E}(\Delta, \theta, \sigma, N, K, d) \neq \emptyset$. Both the ACB (Algorithm 1) and its adaptive version ACB$^*$ (Algorithm 2) are $\delta$-PAC on $\mathcal{E}(\Delta, \theta, \sigma, N, K, d)$. There exist numerical constants $c$, $c'$, $c''$, independent of all the parameters $\Delta, \theta, \sigma, N, K, d$ such that the following holds. For any environment $\nu$ in $\mathcal{E}(\Delta, \theta, \sigma, N, K, d)$, such that $\theta \geq \log(K)/N$, we have*

$$\mathbb{E}_{ACB,\nu}[\tau_{ACB}] \leqslant cN + c'A\ ;\qquad \tau_{ACB} \leqslant cN + c'(A + B)\ \text{a.s.}$$

$$\mathbb{P}_{ACB^*,\nu}\left[\tau_{ACB^*} \leqslant cN + c''L\log^2(L)(A + B)\right] \geq 1 - \delta\ ,$$

*where $L := \log_2\left(\frac{1}{\theta K}\left(\frac{\Delta_0^2}{\Delta^2} \vee 1\right)\right)$.*

This theorem entails that the budget for both ACB and ACB$^*$ is optimal. We further comment on this in the discussion section.

## 5. Numerical experiments

In this section, we run experiments on synthetic data with standard Gaussian noise ($\sigma = 1$) to illustrate our theory. We consider environments with equidistant centers (with $\Delta_* = 1$), and balanced groups ($\theta_* \approx 1/K$). We choose a high-dimensional setting with $N = 200$, $d = 1000$, and $K \in \{10, 15, 20, 25\}$.

As the main competitor, we implement oracle-BOC, an oracle version of the BOC algorithm (Yang et al., 2024). As the setting is perfectly symmetric (balanced clusters, equidistant means), the Oracle-BOC policy is equivalent to the Uniform Sampling strategy, where Loyd algorithm initialized by maximin. We implement instead a kmeans++ initialization, as it is known to outperform maximin (Celebi et al., 2013). Besides, the total budget of oracle-BOC is chosen in such a way that the procedure is empirically $\delta$-PAC. As a consequence, oracle-BOC both corresponds to a state-of-the art batch clustering procedure and to an oracle version of Yang et al. (2024) where both the stopping time and the sampling strategy are provided by an oracle.

We run the non-adaptive procedure ACB and a variation of ACB$^*$ called ACB†, which is adaptive to the unknown parameter $\Delta_*$. Its structure is very similar to Algorithm 2 for ACB$^*$, with the difference of assuming that $\theta$ is known, we provide more explanations in Appendix A. In this experiment, we use the variant ACB† to allow for a more fair comparison to Oracle-BOC. Regarding ACB, we assume that $\Delta_*$ is known, and we implement the non-adaptive version of ACB with $\delta = 0.1$. In order to provide a tighter calibration of ACB, we slightly modify ACB algorithm in order to specialize it to the Gaussian distribution –see Appendix A.

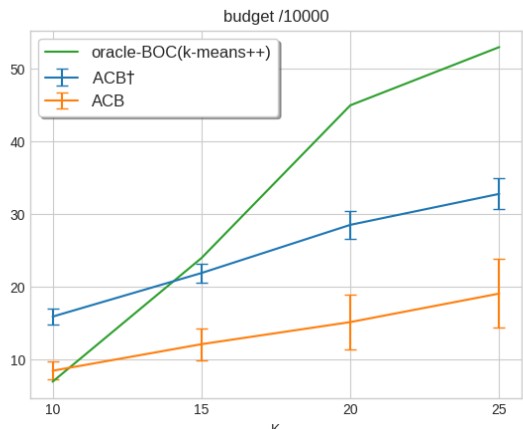

Figure 1: Comparison of the necessary budget for ACB and oracle-BOC with varying number of clusters.

We represent in blue (resp. orange) the (empirical) budget of ACB† (resp. ACB) computed with 100 simulations, for $K = 10, 15, 20, 25$. The error bars are equal to twice the standard deviation. In green, we provide the smallest budget for which oracle-BOC (initialized with kmeans++) makes less than $10\%$ of error out of 100 experiments. As this budget is a numerical constant, there are no error bars.

In Figure 1, we plot the estimated mean budget of ACB† and ACB as a function of $K$, as well as the budget of oracle-BOC, where the budget has been chosen by an oracle so that the procedure is exactly $\delta$-PAC with $\delta = 0.1$. This figure confirms our theoretical findings that, in a high-dimensional setting ($d \gg N/K$), ACB improves over oracle-BOC - which is here equivalent to a state of the art batch clustering algorithm - when the number $K$ of groups increases. Also, we have checked that ACB† and ACB are $\delta$-PAC. Fixing $\delta = 0.1$, we observe no more than 1 error out of 100 experiments. We detail further the experimental setup (including compute resources) in Appendix A. We provide also an experiment for which the number of clusters is fixed, and the dimension varies in Appendix A.

## 6. Discussion

**Optimality of ACB.** First, we discuss the budget of ACB, and we compare it to the information-theoretical lower bound of Theorem 4. To simplify the discussion, let us first consider the case where the partition $G^*$ is almost balanced, that is when $\theta$ is of the order of $1/K$, and assume that $\frac{\Delta^2}{\sigma^2} \lesssim \log(N/\delta)$. According to Theorem 5, the $\delta$-PAC algorithm ACB has an expected budget upper bounded by (3), as long as $K \leq N/\log(N)$. In light of Theorem 4, we see that the expected budget is optimal with respect to all the quantities of the problem: the number of arms $N$, the minimum separation $\Delta$, the number of groups $K$, the probability $\delta$, and the subGaussian norm $\sigma$. The only restriction is that the number of groups $K$ is smaller than $N/\log(N)$, but it is really mild as non-supervised learning problems are mostly relevant for dimension reduction, that is when $K$ is really small compared to $N$. In fact, for larger $K \in [\frac{N}{\log(N)}, N/2]$, the expected budget $\mathbb{E}_{ACB,\nu}[\tau_{ACB}]$ is optimal, up to a possible $\sqrt{\log(N)}$ multiplicative term. Theorem 5 also states high probability controls of the budget $\tau_{ACB}$ and $\tau_{ACB^*}$ which again, are optimal (up to log terms for the latter), in most regimeis optimal, up to a possibles.

When the true partition $G^*$ is extremely unbalanced, that is for $\theta_*$ as small as $1/N$ but the dimension $d$ is seen as a constant, our procedures turns out to still match the lower bound on the budget. When the true partition $G^*$ is extremely unbalanced, so that $\theta_* \leq \frac{\log(K)}{\sqrt{\log(N/\delta)KN}}$ and the dimension $d$ is really large, the bound $A$ on the expected budget may be larger than the lower bound of Theorem 4. Note that this regime is extremely atypical for clustering problems. We conjecture

that both the lower and the upper bounds could be improved in this extreme case, but we leave this for future work.

**Further comparison with (Yang et al., 2024).** When $\delta$ goes to zero while $\sigma$, $\Delta$, $K$, and $d$ are fixed, the average budget of ACB in (3) is at most of the order of $\frac{\sigma^2}{\Delta^2} N \log(1/\delta)$, and is consistent with the BOC algorithm of (Yang et al., 2024). Still, we mention that (Yang et al., 2024) manage to pinpoint the exact value of the asymptotic optimal budget, while our non-asymptotic bounds are only tight up to numerical constants. We however point out that this asymptotic expression hides dependencies on $K$, $d$, and $N$, which are not negligible unless $\delta$ is exponentially small with respect to $d, K$ - and in high dimension, such a high confidence regime is typically out of reach.

**Comparison to batch clustering.** We briefly come back to our fundamental questions on the comparison between the batch and clustering with bandit feedback problems. Contrary to the usual batch setting, we have established that the polynomial-time strategy ACB is information-theoretical optimal, thereby establishing the absence of computation-information gap. This is in contrast with the classical batch clustering problem, where strong evidence of a computation-information gap were proved in (Even et al., 2024) in high dimension, when there are many groups. We therefore illustrate here that clustering is an unsupervised learning problem, where repeated active sampling breaks a computational barrier, which is interesting and opens perspectives for other unsupervised clustering problems where computation-information gap are conjectured.

**Conclusion and limitations.** In our paper, we characterized the non-asymptotic minimal budget for recovering the groups in a collection of environment $\mathcal{E}(\Delta, \theta, \sigma, N, K, d)$, where the minimum distance between the groups is higher or equal to $\Delta$, and all groups have a size larger than $\theta$. We also crafted a strategy adaptive to both $\theta_*$ and $\Delta_*$. Unlike in batch clustering, our results prove that there is no computation-information gap.

Our work still has limitations, and raises several open questions:

First, it remains to explore how sequential and active learning can be leveraged for adapting to heterogeneous distances between groups and heterogeneous group sizes. This has been investigated in (Yang et al., 2024) in the asymptotic regime, but not in the non-asymptotic regime. It would be interesting to have an algorithm which fully adapts to all inter-groups gaps - and not just to the minimal one, or to some target distance. This is a relevant and interesting direction for future works, but it goes beyond the present work whose main aim was to disprove the existence of a statistical-computational gap in sequential clustering. In our work, we also assume that the mean vector means within each groups are exactly equal, as it allows us a simple comparison with batch clustering. An interesting way of relaxing this assumption would be to assume that the means within each group are not equal, but close within the groups. A recent paper Chandran et al. (2025) explores this question in the asymptotic regime where $\delta$ goes to $0$.

Second, when $\sigma$ is unknown, building a sampling strategy that is adaptive to it, would require to estimate the subGaussian norm of the noise, while at the same time estimating the distances between the means. We leave this question for a future work. Finally, as in most of the clustering literature, we assumed that the number $K$ of groups was known to the learner. Investigating the problem of estimating or testing the number of groups in an active setting is also an interesting research direction.

## Acknowledgments

The work of V. Thuot and N. Verzelen has been partially supported by grant ANR-21-CE23-0035 (ASCAI,ANR). The work of A. Carpentier has been partially supported by the DFG CRC 1294

'Data Assimilation', Project A03, by the DFG Forschungsgruppe FOR 5381 "Mathematical Statistics in the Information Age - Statistical Efficiency and Computational Tractability", Project TP 02, by the Agence Nationale de la Recherche (ANR) and the DFG on the French-German PRCI ANR ASCAI CA 1488/4-1 "Aktive und Batch-Segmentierung, Clustering und Seriation: Grundlagen der KI". The work of C. Giraud has been partially supported by grant ANR-19-CHIA-0021-01(BiSCottE, ANR) and ANR-21-CE23-0035 (ASCAI, ANR).

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

## Appendix A. Details on the numerical experiments

### Experimental setting

We consider artificial data, generated with standard Gaussian noise ($\sigma = 1$). We build environments with equidistant centers, and balanced groups. Precisely, we choose $\mu(k) = e_k/\sqrt{2}$, where $\{e_1, \ldots, e_K\}$ are the $K$ first vector of the canonical base of $\mathbb{R}^d$, so that the centers are equidistant, and $\Delta_* = 1$.

We choose a large number of arms $N = 200$. We fix a partition where each group has a size $\lfloor N/K \rfloor$ or $\lfloor N/K \rfloor + 1$, which makes the partition almost balanced, with $\theta_* = \frac{1}{N} \lfloor \frac{N}{K} \rfloor \sim \frac{1}{K}$.

We provide in this paper two figures. In Figure 1, the dimension is fixed to $d = 1000$, and the number of clusters varies in $\{10, 15, 20, 25\}$. In Figure 2, the number of clusters is fixed to $K = 15$, and the dimension varies in $\{500, 1000, 1500, 2000\}$.

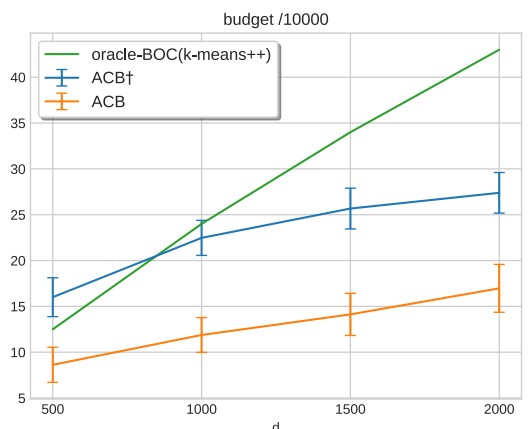

Figure 2: Comparison of the necessary budget for ACB and oracle-BOC, with varying dimension.

In blue (resp. orange) the (empirical) budget of ACB† (resp. ACB) computed with 100 simulations, for $d = 500, 1000, 1500, 2000$. The error bars are equal to twice the standard deviation. In green, we provide the smallest budget for which oracle-BOC (initialized with kmeans++) makes less than 10% of error out of 100 experiments.

### Variant of the procedure and parametrization of ACB

In the main paper, we introduced and calibrated ACB to allow for subGaussian noise. In particular, the quantities $n_s$, $n_{\max}$, $I$ and $J$, defined in eqs. (10) to (12) and (14) were calibrated by inverting concentration inequalities, at the cost of non-optimal numerical constants.

In order to study numerically our procedure, we implement a variant of the procedures whose tuning parameters is adjusted to the Gaussian setting. The test statistics and the classifier, so as the tuning parameters are adjusted to specifically work with Gaussian distribution.

The structure of SRI remains unchanged, however, we use the following calibration. In SRI, we avoid dual sampling when computing $\hat{d}_{ab}$ ((9)) estimate of $\|\mu_a - \mu_b\|^2$ in order to save a factor two in the budget. We modify indeed the test statistic in the function REPRESENTEDTEST used in SRI. We use IS.TRUE $\left\{ \min_{b \in S} \left[ \|\hat{\mu}_a - \bar{\mu}_b\|^2 - d\sigma^2 \left( \frac{1}{n_s} + \frac{1}{n_{\max}} \right) \right] \leq \Delta^2/2 \right\}$, so that there is no need to compute $\hat{\mu}_a'$ in REPRESENTEDTEST, and neither $\bar{\mu}_b'$ in SRI. Observe that $\|\hat{\mu}_a - \bar{\mu}_b\|^2$ is an estimator of $\|\mu_a - \mu_b\|^2$ which is biased. As in the experiment, the variance is known, we debias it, using the shift $d\sigma^2(\frac{1}{n_s} + \frac{1}{n_{\max}})$ in the statistics above.

In order to have a $\delta$-PAC algorithm, we choose $n_{\max} = 4\frac{\sigma^2}{\Delta^2}(x - d)$, where $x$ is the $1 - \delta/K$ quantile of a $\chi^2$ distribution with $d$ degrees of freedom. This quantile is obtained with the library scipy.stats. Now, we cet $n_0, \ldots, n_r$, by putting $n_s = \lceil 2^s n_0 \rceil$ for $s = 0, \ldots, r$, for all $s$. We choose $n_0$ so that the budget spent on the rejected candidates should be close to the budget spent on the

accepted representatives. We choose $n_0 = \lceil (K/U')n_{\max} \rceil$ where $U' = (1/\theta)\log(1/\delta)$. Finally, $r$ is chosen such that $n_r = 2^r n_0$ is equal to $n_{\max}$, up to a factor 2.

To simplify the procedure, the stopping condition from Line 12 in SRI is modified, and we only stop when $S$ contains $K$ representatives, and not sooner.

In the Active Distance-based Classification routine (ADC), we also modify the sampling size $I$ and $J$ (Equation (14)), and the classifier from (11). In the classification, we label each arm with $\mathrm{argmin}_{j=1,\ldots,K} \|\hat{\mu}_a - \hat{\mu}(j)\|$, which is a distance-based classifier as in Equation (15), but without dual sampling. By the analysis of the probability of error of this classifier, we choose

$$I = \left\lceil \frac{\sigma^2}{\Delta^2} \max(16\beta, 4\sqrt{2K/N}\alpha) \right\rceil \quad, \qquad J = \left\lceil \frac{\sigma^2}{\Delta^2} \max(16\beta, 4\sqrt{2N/K}\alpha) \right\rceil \quad,$$

where $\beta$ is the $1 - \delta/(4K(N - K))$ quantile of a standard normal distribution (obtained with scipy.stats), and $\alpha$ is the $1-\delta/(4K(N-K))$ quantile of a product of independant standard $\mathcal{N}(0, I_d)$, that we had to compute empirically with Monte Carlo. With this choice of tuning parameters, one can prove that the corresponding variant of ACB is $\delta$-PAC for Gaussian data. The proof is analogous to the one in Section C. Our numerical experiments confirm that, with these tuning parameters, the modified procedure is still $\delta$-PAC.

**Description of the variant ACB† and implementation**

We now describe the variant $ACB\dagger$ of $ACB_*$ used in the experiments. This version is calibrated to work well with balanced groups, or with a known balancedness $\theta$. In ACB†, we assume no knowledge of $\Delta_*$, and we perform SRI with growing values $(\Delta_k)_k$. We start with $\Delta_0 = \sqrt{4\sigma^2(x - d)}$, with $x$ the $1 - \delta/K$ quantile of a chi-square distribution with $d$ degrees of freedom. We then use $\Delta_k = \Delta_0/2^k$ and $\delta_k = \delta/(6(k + 1)^2)$.

First, for each call of SRI, we use the calibration of SRI described in the paragraph above, and we put a limit on its budget. We limit the budget of $\mathrm{SRI}(\delta_k, \Delta_k, \theta)$ by $T'_{\max} = \tau_{ADC}(\delta_k, \Delta_k) = (N - K)I + KJ$, which is the budget that we use to classify with $\mathrm{ADC}(\delta_k, \Delta_k)$, and where $I$, $J$ are calibrated as in the paragraph above.

When we reach $k$ such that $\mathrm{SRI}(\delta, \Delta_k, \theta)$ contains $K$ arms, we estimate $\Delta_*$, based on the data collected on this call of SRI. If $S$ is the set of representatives identified with SRI, and $(\hat{\mu}_a)_{a \in S}$ are the estimates of the centers computed by SRI, we compute

$$\hat{\Delta}^2 = \underset{a \neq b \in S}{\mathrm{argmin}} \left\{ \|\hat{\mu}_a - \hat{\mu}_b\|^2 - 2d\frac{\sigma^2}{n_{\max}} \right\} \quad.$$

Then, ADC is applied with the parameter $\hat{\Delta}$.

---

**Algorithm 3** ACB† ($\Delta_*$ unknown)

**Input:** $\delta$, and $\theta$

13 **for** $k = 0, 1, \ldots$ **do**
14      Compute $S_k \leftarrow \mathrm{SRI}(\delta_k, \Delta_k, \theta \vee \frac{1}{N})$
15      **if** $|S_k| = K$ **then**
16          $\hat{\Delta}^2 \leftarrow \inf_{a,b \in S_k} \left\{ \|\hat{\mu}_a - \hat{\mu}_b\|^2 - 2d\frac{\sigma^2}{n_{\max}} \right\}$
17          **return** $\hat{G} = \mathrm{ADC}(\delta/3, \hat{\Delta}, S_k)$
18      **end**
19 **end**

---

**Experiments Compute Resource**

We used for the experiments python/Anaconda/3-5.1.0 and the scikit-learn/1.02 package. The experiment were ran in the cluster MESO@LR, working with CPUs of 4Gb and 8Gb. To give an idea on the computation cost, for $N, d, K = 200, 1000, 10$, each call for ACB takes approximately 6 minutes, while each call for ACB† took approximately 18 minutes. In total, the curve for ACB from Figure 1 took around 10 for each value of $K$ and 30 hours for ACB†.

## Appendix B.  Proof of the Lower Bound

**Sketch of the proof**

Throughout Appendix B, we fix $\Delta > 0$, $\sigma$ and $d$. In this section, we bound the worst case budget for any $\delta$-PAC algorithm on the collection of environments $\mathcal{E}(\Delta, \theta, \sigma, N, K, d)$ –see (5), and we prove Theorem 4.

We start in Appendix B.1 by reducing the clustering with bandit feedback problem to a binary classification problem. For that purpose, we construct a family of environments, for which, the problem of clustering with bandit feedback essentially reduces to $\lfloor K/2 \rfloor$ independent and identical sub-problems of binary classification. The environments that we construct are symmetrical, (in some sense defined in the proof) and we will explain in Lemma 12 that we can find an optimal algorithm (as defined in Definition 11) that samples (in expectation) the same number of time each arm. This construction jointly deals with the low-dimensional (Lemma 6) and the high-dimensional (Lemma 7) regimes. For the construction, we will need to assume that $N \geqslant 2K$, that $K$ is even, and that $K$ divides $N$, and we explain in Lemma 9 how to reduce to this hypothesis.

We divide then the proof in two main lemmas, dealing with the low-dimensional and high-dimensional regimes. We recall that $T^*$ – see (5) – is the optimal worst case budget.

**Lemma 6**  *If $N \geqslant 2K$, $K$ is even, $K$ divides $N$, and $\theta = 1/K$, then for any $\delta \in (0, 1)$,*

$$T^*(\delta, \mathcal{E}(\Delta, \theta, \sigma, N, K, d)) \geqslant \frac{\sigma^2}{\Delta^2} N \, \mathrm{kl}\left(1 - \delta, \frac{\delta}{N}\right) \ ,$$

*where* $\mathrm{kl}$ *is the relative entropy defined as* $\mathrm{kl} : x, y \mapsto x \log(x/y) + (1-x) \log((1-x)/(1-\delta))$.

**Lemma 7**  *If $N \geqslant 2K$, $K$ is even, $K$ divides $N$, and $\theta = 1/K$, then for all $\delta \in (0, 1/6)$,*

$$T^*(\delta, \mathcal{E}(\Delta, \theta, \sigma, N, K, d)) \geqslant \frac{\sigma^2}{\Delta^2} \sqrt{\frac{dKN}{72} \, \mathrm{kl}\left(\frac{1}{3} - 2\delta, \frac{4\delta}{N}\right)} \ .$$

In Appendix B.2, we prove Lemma 6, the dimension-free lower bound. It is enough for this term to assume that the centers of the groups are known, and we use an information-theoretic method with the KL-divergence, which is somewhat related to previous works for the thresholding bandit problem derived by (Cheshire et al., 2020).

In Appendix B.3, we prove Lemma 7 in the high-dimensional regime. For this purpose, we will consider a Bayesian setting and assume a Gaussian prior on the centers of the groups. The KL-divergence is hard to compute for the probability induced by the interaction between an algorithm and a Bayesian bandit environment. To overcome this technical problem, we formalize the intuition that the problem of clustering with bandit feedback is in some sense "harder" than a problem of

supervised learning where the player knows the labels of every arm except one arm that has to be classified. It will reduce the problem into a two-sample (batch) testing problem (see Definition 18), and the conclusion will follow from some explicit computation and an impossibility result for this latter batch problem.

We postpone the proofs of some technical lemmas in Appendix B.4

**Remark 8** *We explain quickly the term $cN$ in the lower bound from Theorem 4. Assume that, for any $\nu \in \mathcal{E}(\Delta, \theta, \sigma, N, K, d)$, it holds that $\mathbb{E}_{\pi,\sigma}[\tau] \leqslant cN$ with $c < 1/2$. Then, for any environment $\nu$, there is a fixed probability that two arms from two different groups are not sampled at all during the procedure. The best to do for the learner is then to estimate randomly the groups of the arms, inducing a fixed probability of making at least one error in the clustering. We do not discuss further this term $cN$ in the lower bound in the remainder of the proof. Still, note that it is only relevant in an artificial regime where $\Delta$ is arbitrary large.*

Now, we explain how the remark above, Lemmas 6 and 7 imply Theorem 4.

**Proof** [Proof of Theorem 4]

Let $N, K$ such that $N \geqslant 2K$. Let $\theta > 0$ such that $\mathcal{E}(\Delta, \theta, \sigma, N, K, d) \neq \emptyset$.

We first reduce the problem into a problem where $K$ is even, $N$ is a multiple of $K$, and the groups have the same size $N/K$. With this technical condition fulfilled, we will be able to Lemmas 6 and 7. We define $N'$, $K'$, and $\theta'$:

- if $K$ is even, $K' := K$ and $N' := K\lfloor N/K \rfloor$ ;

- if $K$ is odd, $K' := K - 1$ and $N' = K' \left\lfloor \frac{N - \lceil \theta N \rceil}{K'} \right\rfloor$ ;

- in both cases, $\theta' := 1/K'$ .

We now use the following natural reduction result, whose proof is in Appendix B.4.

**Lemma 9** *The optimal worst case budget over $\mathcal{E}(\Delta, \theta, \sigma, N, K, d)$ is larger than the one over $\mathcal{E}(\Delta, \theta', \sigma, N', K', d)$,*

$$T^*(\delta, \mathcal{E}(\Delta, \theta, \sigma, N, K, d)) \geqslant T^*(\delta, \mathcal{E}(\Delta, \theta', \sigma, N', K', d)) \ .$$

It holds immediately that that $K'$ is even, and that $K'$ divides $N'$. Moreover, as $\mathcal{E}(\Delta, \theta, \sigma, N, K, d) \neq \emptyset$, then $\lceil \theta N \rceil \leqslant N/K$. This inequality and the assumption $N \geqslant 2K$, implies that $N' \geqslant 2K'$. We can then use Lemma 6 and Lemma 7 with $N'$ and $K'$ in order to bound $T^*(\delta, \mathcal{E}(\Delta, \theta', N', K', d))$.

We have for any $\delta \in (0, 1/6)$,

$$T^*(\delta, \mathcal{E}(\Delta, \theta', \sigma, N', K', d)) \geqslant \frac{\sigma^2}{\Delta^2} N' \, \mathrm{kl}\left(1 - \delta, \frac{\delta}{N'}\right) \vee \frac{\sigma^2}{\Delta^2} \sqrt{\frac{dK'N'}{72} \, \mathrm{kl}\left(\frac{1}{3} - 2\delta, \frac{4\delta}{N'}\right)} \ .$$

We can also easily deduce from the expression of $N'$ that $N' \geqslant N/6$.

Finally, we study $\delta \mapsto \mathrm{kl}(1 - \delta, 2\delta/N')$ to obtain the bound valid for all $\delta \in (0, 1)$ and for all $N' \geqslant 1$,

$$\mathrm{kl}\left(1 - \delta, \frac{2\delta}{N'}\right) \geqslant \log\left(\frac{1}{\delta}\right) + \log(N')(1 - \delta) - 1.5 \ .$$

In particular, we have the bound $\mathrm{kl}\left(1 - \delta, \frac{2\delta}{N'}\right) \geqslant \frac{1}{2}\log(N'/\delta)$ for $\delta \in (0, 1/4)$.

By studying the variation of $\delta \mapsto \mathrm{kl}(1/3 - 2\delta, 4\delta/N')$, we obtain the bound valid for all $\delta \in (0, 1/6)$ and for all $N'$,

$$\mathrm{kl}\left(\frac{1}{3} - 2\delta, \frac{4\delta}{N'}\right) \geqslant \frac{1}{3}\left[\log\left(\frac{1}{4\delta}\right) + \log(N')(1 - 6\delta)\right] - 0.7 \ .$$

Combining all these inequalities and Remark 8, we obtain Theorem 4. ∎

## B.1. From clustering with bandit feedback to binary classification

### B.1.1. CONSTRUCTION OF A FAMILY OF ENVIRONMENTS

From now on, we assume that $K$ is even, and $N/K$ is an integer. In all the proof, we only consider perfectly balanced environments such that $\theta = 1/K$. We also assume that $N \geqslant 2K$. Define $L := \lfloor K/2 \rfloor$. In this subsection, we construct a family of environments defined with a prior on the centers of the groups.

We assume that the noises are Gaussian with covariance matrix $\sigma^2 I_d$. This fulfills the subGaussian noise hypothesis from Assumption 1. In this Gaussian model, an environment is characterized by the hidden partition $G^*$ and the (distinct) centers of the groups.

We use a Bayesian approach, and we define the $K = 2L$ centers of the groups, that we order as $\mu_{1,1}, \mu_{1,-1}, \ldots, \mu_{L,1}, \mu_{L,-1}$. For all $l \in [L]$, we construct the centers $\mu_{l,1}$ and $\mu_{l,-1}$ as symmetrical with respect to some offset. More specifically, for all $(l, g) \in [L] \times \{-1, 1\}$, we define

$$\mu_{l,g} := g\bar{\mu}(l) + C(l) \ , \tag{18}$$

where

- for all $l \in [L]$, $C(l) \in \mathbb{R}^d$ is a fixed offset defined as $C(l) = \beta(l\Delta, 0, \ldots, 0) \in \mathbb{R}^d$;

- $\beta > 1$ will be fixed later and is arbitrary large;

- $\bar{\mu} := \bar{\mu}(1), \ldots, \bar{\mu}(L)$ are i.i.d and $\bar{\mu}(l) \sim \gamma$. The prior distribution $\gamma$ over $\mathbb{R}^d$ will be set differently if we consider the low or high-dimensional regime. We will specify later this prior.

Through the proof, we fix a partition $G^*$ of $[N]$ into $K$ groups. The partition $G^*$ is composed of $K = 2L$ nonempty groups $G^*_{1,1}, G^*_{1,-1}, \ldots, G^*_{L,1}, G^*_{L,-1}$ associated to the means $\mu_{1,1}, \mu_{1,-1}, \ldots, \mu_{L,1}, \mu_{L,-1}$. For each arm $a \in [N]$, we denote as $(l^*_a, g^*_a) \in [L] \times \{-1, 1\}$ for the labels such that $a \in G^*_{l^*_a, g^*_a}$ and $\mu_a = \mu_{l^*_a, g^*_a}$. Also, we will always restrict ourselves to balanced partitions $G^*$ so that each group $G^*_{l,g}$ has the same size $N/K$ and thus $\theta_* = 1/K$.

In summary, we have

$$[N] = \bigsqcup_{(l,g)\in[L]\times\{-1,1\}} G^*_{l,g} \ ,$$

where the groups $(G^*_{l,g})$ are nonempty and share the same size $N/K$.

We also define the so-called $L$ "blocks". For $l \in [L]$, we define $G^*_l := \{a \in [N] \ ; \ l^*_a = l\} = G^*_{l,1} \sqcup G^*_{l,-1}$. For each arm $a \in [N]$, $l^*_a$ corresponds to the label of the pair of groups (block) $G^*_l$

that contains $a$. If $l_a^* = l$, then the arm $a$ belongs either to $G_{l,1}^*$ or $G_{l,-1}^*$ depending on the value of $g_a^* \in \{-1, 1\}$. We also denote as $G_+^* := \{a \in [L]; g_a^* = +1\}$.

We now construct a set of partitions obtained from $G^*$ by switching two arms from the two different groups of the same block. Arbitrarily define a set $\{s(1), \dots, s(L)\}$ of arms such that for all $l \in [L], s(l) \in G_{l,-1}^*$. For any arm $a \in [N]$, we write $b_a := s(l_a^*)$. For an arm $a$ in $G_+^* = \{a \in [L]; g_a^* = +1\}$, we define $G_{(a)}^*$ as the partition equal to $G^*$ except that the arm $a$ is switched from $G_{l_a^*, 1}$ to $G_{l_a^*, -1}$, and the arm $b_a$ is switched from $G_{l_a^*, -1}$ to $G_{l_a^*, +1}$. This is a valid partition with $K$ nonempty and perfectly balanced groups. As we took $N \geqslant 2K$, it holds that, if any two distinct partition $G$ and $G'$ belong to $\{G^*\} \cup \{G_{(a)}^*\}_{a \in G_+^*}$, we have $G \not\sim G'$. As a consequence, any $\delta$-PAC algorithm distinguishes, with probability higher than $1 - \delta$, whether the environments is characterized by a partition $G^*$ or by some $(G_a^*)_{a \in G_+^*}$.

For any partition $G'$ such that $[N] = \sqcup_{l,g} G'_{l,g}$, we denote as $\nu(G', \bar{\mu})$ for the environment constructed in this paragraph with the means $(\mu_{l,g})_{l,g} = (C(l) + g\bar{\mu}(l))$ and $\bar{\mu} \in \mathbb{R}^d$. We will use $\mathbb{P}_{\pi, G', \bar{\mu}}$ [resp. $\mathbb{E}_{\pi, G', \bar{\mu}}$] for the probability distribution [resp. expectation] induced by the interaction between an algorithm $\pi$ and the environment $\nu(G', \bar{\mu})$ for a fixed realization of $\bar{\mu}$. We also denote as $\mathbb{P}_{\pi, G'} = \int_{\bar{\mu}} \mathbb{P}_{\pi, G', \bar{\mu}} \, d\gamma^{\otimes L}(\bar{\mu})$ [resp. $\mathbb{E}_{\pi, G'}$] as the integrated probability with respect to the prior $\gamma^{\otimes L}$ on $\bar{\mu}$ [resp. expectation].

There is a technical detail that has to be handled with this Bayesian prior, if $\bar{\mu}_l$ is too small or too large, the environment $\nu(G', \bar{\mu})$ is not necessary in $\mathcal{E}(\Delta, \theta, \sigma, N, K, d)$. We define therefore $\mathcal{Y} := \bigcap_{l \in [L]} \{\Delta/2 \leqslant \|\bar{\mu}(l)\| \leqslant \Delta(\beta - 1)/2\}$. On $\mathcal{Y}$, the centers are distinct, the minimal gap is larger than $\Delta$, and the set of possible values for $(\bar{\mu}(l))_l$ are disjoint.

We denote $\mathcal{E}_{Sym}(G^*, \gamma)$ as the Bayesian family of environments of the form $\nu(G', \bar{\mu})$, where $\bar{\mu} \sim \gamma^{\otimes L}$ and the partitions $G' \in \{G^*\} \cup \bigcup_{a \in G_+^*} \{G_{(a)}^*\}$.

We explain a bit more the construction.

**Remark 10**

1. *The parameter $\beta$ will be arbitrary large so that it is very easy to decide if two arms belong to different blocks or not. In this case, it is intuitively easy to first separate the arms into $L$ blocks (that means to estimate $l_1^*, \dots, l_N^*$). Then the difficulty of the problem mostly lies in the $L$ sub-problems of binary classification, where each block has to be partition into two groups.*

2. *In the low-dimensional regime, we will take $\bar{\mu}(l) = (\Delta/2, 0 \dots, 0)$ ($\gamma$ is deterministic). It means that we will derive the lower bound from Lemma 6 for fixed centers of the groups $\mu(1), \dots, \mu(K)$ which basically amounts to the simpler setting where the learner knows the centers in advance.*

3. *In the high-dimensional regime, we will use a Gaussian prior on $(\bar{\mu}(l))_{l \in [L]}$. With this prior, we will be able to quantify to what extent we have to estimate the unknown means $(\bar{\mu}(l))_{l \in [L]}$ to be able to group the arms.*

### B.1.2. SYMMETRIZATION

Now, we exploit the different symmetries of the environments of the shape $\nu(G', \bar{\mu})$, and the symmetries of the distribution of the centers when $\bar{\mu} \sim \gamma$, in order to restrict our study to algorithms that are $\delta$-PAC on $\mathcal{E}_{Sym}(G^*, \gamma)$ and that satisfies a symmetry property defined below.

**Definition 11** *We say that $\pi$ is $\delta$-PAC on $\mathcal{E}_{Sym}(G^*, \gamma)$, if, conditionally on the event $\mathcal{Y}$, we have*

$$\mathbb{P}_{\pi, G'}(\hat{G} \sim G' | \mathcal{Y}) \geqslant 1 - \delta \ ,$$

*for any $G' \in \{G^*\} \cup \{G^*_{(a)}\}_{a \in G^*_+}$. We say that an algorithm $\pi$ is symmetric on $\mathcal{E}_{Sym}(G^*, \gamma)$, if for any $b \in [N]$ and $G' \in \{G^*\} \cup \{G^*_{(a)}\}_{a \in G^*_+}$, then*

$$\mathbb{E}_{\pi, G'}[N_b(\tau) | \mathcal{Y}] = \frac{1}{N} \mathbb{E}_{\pi, G'}[\tau | \mathcal{Y}] = \frac{1}{N} \mathbb{E}_{\pi, G^*}[\tau | \mathcal{Y}] \ .$$

*We denote as $\Pi_{Sym}(\delta, \mathcal{E}_{Sym}(G^*, \gamma))$ for the family of symmetric and $\delta$-PAC algorithms on $\mathcal{E}_{Sym}(G^*, \gamma)$.*

Finally, we define the optimal Bayesian budget for an algorithm in $\Pi_{Sym}(\delta, \mathcal{E}_{Sym}(G^*, \gamma))$ as

$$T^*(\delta, \mathcal{E}_{Sym}(G^*, \gamma)) := \inf_{\pi \in \Pi_{Sym}} \mathbb{E}_{\pi, G^*}[\tau | \mathcal{Y}] \ ,$$

where the inf is taken over $\Pi_{Sym}(\delta, \mathcal{E}_{Sym}(G^*, \gamma))$, recalling that $\mathbb{E}_{\pi, G^*}$ is the integrated budget with respect to the prior $\gamma$.

The next lemma implies that we only need to lower bound the quantity $T^*_{Sym}(\delta, \mathcal{E}_{Sym}(G^*, \gamma))$.

**Lemma 12** *If $K$ is even, $K$ divides $N$, $\theta = 1/K$, and $N \geq 2K$, it holds that*

$$T^*(\delta, \mathcal{E}(\Delta, \theta, \sigma, N, K, d)) \geqslant T^*_{Sym}(\delta, \mathcal{E}_{Sym}(G^*, \gamma)) \ .$$

**Remark 13**

*We highlight that this construction essentially reduces the problem into $L$ sub-problems of active binary classification. On the family of environments $\mathcal{E}_{Sym}(G^*, \gamma)$, the offsets $C_1, \ldots, C_L$ and the labels of the blocks $l^*_1, \ldots, l^*_N$ are fixed and common to all the environments $\nu(G^*, \bar{\mu})$ and $\nu(G^*_{(a)}, \bar{\mu})$, it is equivalent to say that this is known by the learner. Then, the problem consists on estimating the partition into two groups $G^*_l = G^*_{l,1} \sqcup G^*_{l,-1}$ (up to switching of the two groups) for any of the $L$ blocks. If an algorithm is symmetric, it will have access in expectation to the same budget to solve each sub-problem.*

*The proof of this Lemma, technical but standard is provided in Appendix B.4. In the proof, we explain how to use the knowledge of the blocks $G^*_1, \ldots, G^*_L$ and the offsets $C(1), \ldots, C(l)$ in order to transform any algorithm into a symmetric algorithm – see Definition 11. The rough idea is to permute the arms, and then to apply the algorithm to the permuted arms.*

## B.2. First Lower bound : proof of Lemma 6

In this section, we prove the Lower Bound from Lemma 6. We highlight that the lower bound from Lemma 6 does not depend on the dimension $d$. Thus, we will derive lower bound for fixed centers of the groups which basically amounts to the simpler setting where the learner knows them in advance.

We use the construction of Appendix B.1, and we choose the prior distribution $\gamma_1 := \delta_\mu$ to be a Dirac, i.e, $\bar{\mu}(l) = \mu$ for all $l$ and the centers are deterministic and fixed. We choose $\mu = (\Delta/2, 0, \ldots, 0) \in \mathbb{R}^d$ and $\beta = 2$. The environment $\nu(G^*, \bar{\mu})$ is in $\mathcal{E}(\Delta, \theta, \sigma, N, K, d)$, so the event $\mathcal{Y}$ from Definition 11 holds almost surely.

**Remark 14** *The clustering with bandit feedback problem on $\mathcal{E}_{Sym}(G^*, \gamma)$ is highly connected to a specific instance of the Thresholding Bandit Problem (TBP), another pure exploration problem studied in (Cheshire et al., 2020). In this problem, a player interacts with a multi-armed bandit environment with one-dimensional rewards, and she has to recover the set of arms with a mean larger or equal to a certain threshold (for us, this threshold is $\Delta$). The proof of Lemma 6 is inspired by the proof of Theorem 1 in (Cheshire et al., 2020). For the thresholding bandit problem, the authors derive a lower bound in the fixed budget setting. In this setting, the player has to minimize the simple regret (related to the probability of error), using a fixed budget. From their result, we could deduce a lower bound of the form $\frac{\sigma^2}{\Delta^2}(N-K)\log(\frac{N-K}{\delta})$. Here, we use a workaround to establish a slightly tighter lower bound of the form $\frac{\sigma^2}{\Delta^2}N\log(\frac{N}{\delta})$.*

We consider $T^*_{Sym}(\delta, \mathcal{E}_{Sym}(G^*, \gamma_1)) = \inf_{\pi \in \Pi_{Sym}} \mathbb{E}_{\pi, G^*}[\tau]$ where the $\inf$ is taken over all symmetric and $\delta$-PAC algorithm on $\mathcal{E}_{Sym}(G^*, \gamma_1)$ –see Definition 11.

**Lemma 15** *If $K$ is even, $K$ divides $N$ and $N \geqslant K$, then,*

$$T^*_{Sym}(\delta, \mathcal{E}_{Sym}(G^*, \gamma_1)) \geqslant N \frac{\sigma^2}{\Delta^2} \mathrm{kl}\left(1 - \delta, \frac{2\delta}{N}\right).$$

Then, Lemma 6 simply follows from the reduction arguments of Lemma 12 and 15 that we prove now.

**Proof** [Proof of Lemma 15] Let $\pi$ be a symmetric and $\delta$-PAC algorithm for the clustering with bandit feedback problem on $\mathcal{E}_{Sym}(G^*, \gamma_1)$. It outputs a partition $\hat{G}$ of $[N]$ such that for any $a \in G^*_+$,

$$\mathbb{P}_{\pi, G^*_{(a)}}(\hat{G} \sim G^*_{(a)}) \geqslant 1 - \delta \quad , \text{and}$$

$$\mathbb{P}_{\pi, G^*}(\hat{G} \sim G^*) \geqslant 1 - \delta \ .$$

The main tool that we use is a data-processing inequality– see e.g.(Gerchinovitz et al., 2020). We will use the KL-divergence which, in our setting, turns out to be explicitly computed. The difficulty of the proof is to recover the term $\log(N/\delta)$ in the lower bound of the budget. For that, we adapt the proof page 15 of (Cheshire et al., 2020) to the fixed confidence setting. The idea is that, instead of constructing one partition, different from $G^*$, we constructed a collection of $\{G^*_{(a)}\}_{a \in G^*_+}$, where any algorithm has to distinguish $G^*$ from any of these environments (up to relabelling).

First, we use lemma 1 from (Kaufmann et al., 2016) which relies on the data-processing inequality and the decomposition of the KL-divergence in the multi-armed bandit model. It holds that, for any $a \in G^*_+$,

$$\mathrm{kl}\left(\mathbb{P}_{\pi, G^*_{(a)}}(\hat{G} \sim G^*_{(a)}), \mathbb{P}_{\pi, G^*}(\hat{G} \sim G^*_{(a)})\right) \leqslant \mathrm{KL}\left(\mathbb{P}_{\pi, G^*_{(a)}}, \mathbb{P}_{\pi, G^*}\right) \tag{19}$$

$$= \mathbb{E}_{\pi, G^*_{(a)}}[N_a(\tau) + N_{b_{l^*(a)}}(\tau)]\frac{\Delta^2}{2\sigma^2} \ ,$$

the last equality follows from the fact that the environments $\nu(G^*, \bar{\mu})$ and $\nu(G^*_{(a)}, \bar{\mu})$ only differ on arm $a$ and $b_a$ and $\mathrm{KL}(\mathcal{N}(-\Delta/2, \sigma^2), \mathcal{N}(\Delta/2, \sigma^2)) = \Delta^2/2\sigma^2$. We recall that for any $b \in [N]$, $N_b(\tau)$ is the number of times that the arm $b$ is sampled.

Thanks to the joint convexity of the kl function (see Gerchinovitz et al., 2020, Corollary 3), we have

$$
\mathrm{kl}\left(\frac{1}{N/2}\sum_{a\in G_+^*}\mathbb{P}_{\pi,G_{(a)}^*}(\hat{G}\sim G_{(a)}^*), \frac{1}{N/2}\sum_{a\in G_+^*}\mathbb{P}_{\pi,G^*}(\hat{G}\sim G_{(a)}^*)\right) \tag{20}
$$
$$
\leqslant \frac{1}{N/2}\sum_{a\in G_+^*}\mathrm{kl}\left(\mathbb{P}_{\pi,G_{(a)}^*}(\hat{G}\sim G_{(a)}^*), \mathbb{P}_{\pi,G^*}(\hat{G}\sim G_{(a)}^*)\right) \ .
$$

By construction, the partition $G^*$ and all the different partitions $(G_a^*)_{a\in G_+^*}$ belong to different equivalence classes with respect to the relation $\sim$. As $\pi$ is $\delta$-PAC – see Definition 11 , we deduce that

$$
\forall a \in G_+^*, \ \mathbb{P}_{\pi,G_{(a)}^*}(\hat{G}\sim G_{(a)}^*) \geqslant 1-\delta \ ;
$$
$$
\sum_{a\in G_+^*}\mathbb{P}_{\pi,G^*}(\hat{G}\sim G_{(a)}^*) = \mathbb{P}_{\pi,G^*}(\sqcup_{a\in G_+^*}\{\hat{G}\sim G_{(a)}^*\}) \leqslant \mathbb{P}_{\pi,G^*}(\hat{G}\not\sim G^*) \leqslant \delta \ .
$$

With the monotony properties of the kl function, we obtain

$$
\mathrm{kl}\left(1-\delta, \frac{\delta}{N/2}\right) \leqslant \mathrm{kl}\left(\frac{1}{N/2}\sum_{a\in G_+^*}\mathbb{P}_{\pi,G_{(a)}^*}(\hat{G}\sim G_{(a)}^*), \frac{1}{N/2}\sum_{a\in G_+^*}\mathbb{P}_{\pi,G^*}(\hat{G}\sim G_{(a)}^*)\right) \ . \tag{21}
$$

Gathering Equations (19), (20) and (21), we obtain

$$
\mathrm{kl}\left(1-\delta, \frac{2\delta}{N}\right) \leqslant \frac{1}{N/2}\sum_{a\in G_+^*}\mathbb{E}_{\pi,G_{(a)}^*}[N_a(\tau)+N_{b_a}(\tau)]\frac{\Delta^2}{2\sigma^2} \tag{22}
$$

We recall that $\pi$ is symmetric. Hence, For any $a \in G_+^*$, we have

$$
\mathbb{E}_{\pi,G_{(a)}^*}[N_a(\tau)+N_{b_a}(\tau)] = \frac{2}{N}\mathbb{E}_{\pi,G_{(a)}^*}[\tau] = \frac{2}{N}\mathbb{E}_{\pi,G^*}[\tau] \ .
$$

Finally, with Equation (22), we conclude that

$$
\mathbb{E}_{\pi,G^*}[\tau] \geqslant \frac{\sigma^2}{\Delta^2}N\,\mathrm{kl}\left(1-\delta, \frac{2\delta}{N}\right) \ . \tag{23}
$$

We take now the inf over all algorithms $\pi$, which are $\delta$-PAC and symmetric, this proves Lemma 15.
∎

## B.3. Second Lower Bound: proof of Lemma 7

In this section, we prove the lower bound from Lemma 7. If $d \leqslant (8/3)^2\log(K/\delta)$, the lower bound from Lemma 7 is smaller than the dimension-free lower bound from Lemma 6, which is already proved. We may then assume that $d \geqslant (8/3)^2\log(K/\delta)$. For the sake of the presentation, we the proofs of some technical lemmas to the end of the next subsection.

STEP 1: INTRODUCTION OF THE GAUSSIAN PRIOR

In this regime, we choose the prior distribution $\gamma$ to be Gaussian. Indeed, we introduce $\gamma_2 = \mathcal{N}(0, \rho^2 I_d)$ with $\rho^2 = \frac{\Delta^2}{d}$ and $\bar{\mu}(1), \ldots, \bar{\mu}(L)$ are i.i.d of law $\mathcal{N}(0, \rho^2 I_d)$. Also, we choose $\beta = 4$. We consider the Bayesian family of environments constructed in Appendix B.1 $\mathcal{E}_{Sym}(G^*, \gamma_2)$.

Because of this Bayesian prior, we have some additional technical challenge in comparison to the low-dimensional case.

1. We can not use the decomposition of the KL-divergence for bandit in order to compute $\mathrm{KL}(\mathbb{P}_{\pi, G^*_{(a)}}, \mathbb{P}_{\pi, G^*})$ because the integral over the prior $\gamma_1$ is inside the KL-divergence. Most of the work consists on upper bounding this divergence with a divergence that can be computed.

2. We can not compare the maximum budget over $\mathcal{E}(\Delta, \theta, \sigma, N, K, d)$ (i.e. $\sup_{\nu \in \mathcal{E}(\Delta)} \mathbb{E}_{\pi, \nu}[\tau]$) to the Bayesian budget $\mathbb{E}_{\pi, G^*}[\tau]$ because the minimal gap of $\nu(G^*, \bar{\mu})$ is not always larger than $\Delta$. This is why we condition on the event $\mathcal{Y} = \bigcap_{l \in [L]} \{\Delta/2 \leqslant \|\bar{\mu}(l)\| \leqslant \Delta(\beta - 1)/2\} \subset \{\nu(G^*, \bar{\mu}) \in \mathcal{E}(\Delta, \theta, \sigma, N, K, d)\}$.

We compute $\mathbb{P}_{\gamma^{\otimes L}}(\mathcal{Y}^c)$ for the Gaussian prior. This is the only time we will use the hypothesis $d \geqslant (8/3)^2 \log(K/\delta)$.

**Lemma 16** *If we assume that $d \geqslant (8/3)^2 \log(K/\delta)$ and $\gamma_2 = \mathcal{N}(0, \rho^2)$, we have*

$$\mathbb{P}_{\gamma_2^{\otimes L}}(\mathcal{Y}) = \mathbb{P}_{\gamma_2^{\otimes L}}\left(\bigcap_{l \in [L]}\{\Delta/2 \leqslant \|\bar{\mu}(l)\| \leqslant 3\Delta/4\}\right) \geqslant 1 - \delta \ .$$

STEP 2: FROM ACTIVE BINARY CLASSIFICATION TO (BATCH) TWO-SAMPLE TESTING

Let $\pi \in \Pi_{Sym}$ be a $\delta$-PAC and symmetric algorithm for the clustering with bandit feedback problem on $\mathcal{E}_{Sym}(G^*, \gamma_2)$ –see Definition 11. We define $t = 6\mathbb{E}_{\pi, G^*}[\tau|\mathcal{Y}]/N$ and $T = 6\mathbb{E}_{\pi, G^*}[\tau|\mathcal{Y}]/K$.

We recall that, for any $a \in G^*_+ = \{a; g^*_a = 1\}$, $G^*_{(a)}$ is obtained by switching one arm $a$ with another arm $b_a \in G^*_{l^*_a, -1}$. We recall that $N_a(\tau) := \sum_{t=1}^{\tau} \mathbb{1}_{\{A_s = a\}}$ is the number of times the arm $a$ is sampled. We also denote, $M_l(\tau) = \sum_{b: l^*_b = l} N_b(\tau)$ as the number of times the arms in the block $G^*_l$ are sampled. As $\pi$ is symmetric and as the blocks have the same size $2N/K$, we have, for any $a \in G^*_+$,

$$t_a := 3\mathbb{E}_{\pi, G^*_{(a)}}[N_a(\tau) + N_{b_a}(\tau)|\mathcal{Y}] = t \ , \text{ and } \qquad T_a := 3\mathbb{E}_{\pi, G^*_{(a)}}[M_{l^*_a}(\tau)|\mathcal{Y}] = T \ .$$

**Remark 17** *We now give some heuristic in order to explain the rest of the proof. Imagine that, at time $\tau$, the learner receives an oracle that gives the labels of all the arms except the arm $a$, assume also that the learner knows that $a \in G^*_l$. As in a supervised classification setting, the player has to find the label $g_a$ of the unlabeled data sampled from $a$, using the labelled data available. It has access to $N_a(\tau)$ observations from $a$ distributed as $\mathcal{N}(g_a\bar{\mu}(l), \sigma^2 I_d)$, and $M_{l^*_a}(\tau) - N_a(\tau)$ labelled data distributed as $\mathcal{N}(\bar{\mu}(l), \sigma^2 I_d)$. It also has access to data from the other blocks, but those data are not useful to find $g_a$. Moreover, $N_a(\tau)$ is of the order of $\mathbb{E}_{\pi, G^*}[\tau]/N$ and $M_{l^*_a}(\tau) - N_a(\tau)$ is of the order of $\mathbb{E}_{\pi, G^*}[\tau]/L$. As a consequence, with this amount of data, a learner should be able to correctly recover the labels in this simplified setting.*

With this heuristic in mind, we introduce the following (batch) two-sample testing problem.

**Definition 18** *Let $t, T$ be two integers, we consider data $Y_1, \ldots, Y_t, Z_1, \ldots, Z_T$ and two symmetric hypotheses $\mathcal{H}_1$ and $\mathcal{H}_{-1}$ such that, for $g \in \{-1, 1\}$, under $\mathcal{H}_g$, the data follows the law $\mathbb{P}_g$ defined as follows:*

- *$\mu \sim \gamma$ and conditionally on $\mu$ :*

- *$Y_1, \ldots, Y_t, Z_1, \ldots, Z_T$ are independent;*

- *$\forall r \in [t]$, $Y_r \sim \mathcal{N}(g\mu, \sigma^2 I_d)$*

- *$\forall s \in [T]$, $Z_r \sim \mathcal{N}(\mu, \sigma^2 I_d)$.*

This problem is interesting because we can explicitly compute the KL-divergence.

**Lemma 19** *Let $g \in \{-1, 1\}$ and $\mathbb{P}_g$ defined in Definition 18. It holds that*

$$\mathrm{KL}(\mathbb{P}_{-g}, \mathbb{P}_g) = \mathrm{KL}(\mathbb{P}_g, \mathbb{P}_{-g}) = \frac{2tT\rho^4 d}{\sigma^4 + \sigma^2 \rho^2 (t + T)} \leqslant \frac{2tT\rho^4 d}{\sigma^4} \wedge \frac{2\rho^2 d}{\sigma^2} \frac{tT}{t + T} \quad .$$

Now, we explain properly the ideas introduced in the previous remark. We define the event $B_a = \{N_a(\tau) + N_{b_a}(\tau) \leqslant t\} \cap \{M_{l_a^*}(\tau) \leqslant T\}$. Thanks to Markov inequality, the event $B_a$ has a probability higher than a constant and conditionally on $B_a$, the algorithm $\pi$ has access to strictly less information than in (batch) two-sample testing problem defined above with $t_a$ and $T_a$. We formalize this in the following coupling lemma.

**Lemma 20** *Let $a \in G_+^*$ be an arm and fix $A_a$ an event. Consider the family of random variables $(Y_1, \ldots, Y_t)$, $(Z_1, \ldots, Z_T)$ that follows a distribution $\mathbb{P}_{-1}$ – see Definition 18. Consider also an independent sequence $(\epsilon_s, U_s)_{s \geqslant 1}$ of random variables such that for all $s \geqslant 1$, $\epsilon_s \sim \mathcal{N}(0, I_d)$ and $U_s \sim \mathcal{U}([0, 1])$. Then, there exists a function $f_a$ that is measurable according to the random variables $Y, Z, \epsilon, U$ and such that $A_a \cap B_a = f_a(Y, Z, \epsilon, U)$, where the equality holds with respect to the probability distribution $\mathbb{P}_{\pi, G_{(a)}^*} = \int_{\bar{\mu}} \mathbb{P}_{\pi, G_{(a)}^*, \bar{\mu}} \, \mathrm{d}\gamma^{\otimes L}(\bar{\mu})$.*
*Similarly, if $(Y, Z) \sim \mathbb{P}_1$, with the same function $f_a$, $A_a \cap B_a = f_a(Y, Z, \epsilon, U)$, under the probability distribution $\mathbb{P}_{\pi, G^*}$.*

In the previous lemma, we will consider $A_a := \{\hat{G} \sim G_{(a)}^*\}$ for $a \in G_+^*$. By construction of $G_{(a)}^*$ (because $N \geqslant 2K$), the events $A_a$ are disjoint. By using the fact that $\pi$ is $\delta$-PAC on $\mathcal{E}(\gamma, \Delta)$, we have the following property for $A_a$,

**Lemma 21** *The family $(A_a \cap B_a)_{a \in G_+^*}$ is such that*

1. *$\sum_{a \in G_+^*} \mathbb{P}_{\pi, G^*}(A_a \cap B_a) \leqslant \delta + \mathbb{P}_{\gamma^{\otimes L}}(\mathcal{Y}^c) \leqslant 2\delta;$*

2. *$\mathbb{P}_{\pi, G_{(a)}^*}(A_a \cap B_a) \geqslant 1/3 - \delta - \mathbb{P}_{\gamma^{\otimes L}}(\mathcal{Y}^c) \geqslant 1/3 - 2\delta.$*

We delay the technical proofs of Lemma 20 and Lemma 21. From there, we have all the tools that we need. We now use data-processing inequalities similar to the proof of Lemma 6 to conclude.

STEP 3: CONCLUSION TO THE PROOF OF LEMMA 7

We assume that $\delta \in (0, 1/6)$, so that $\mathrm{kl}\left(1/3 - 2\delta, \frac{2\delta}{N/2}\right)$ is defined.

We use the first point of Lemma 21. We notice that the events $(A_a)_a$ are disjoint by construction of $G^*_{(a)}$ and because we took at least two arms by groups ($N \geqslant 2K$), it holds that

$$\sum_{a \in G^*_+} \mathbb{P}_{\pi, G^*}(A_a \cap B_a) = \mathbb{P}_{\pi, G^*}(\sqcup_{a \in G^*_+} A_a \cap B_a) \leqslant 2\delta .$$

With the second point of Lemma 21, for any $a \in G^*_+$, we have

$$\mathbb{P}_{G^*_{(a)}}(A_a \cap B_a) \geqslant 1/3 - 2\delta \geqslant 0 .$$

We use the monotony properties of the $\mathrm{kl}$ function, it holds that

$$\mathrm{kl}\left(1/3 - 2\delta, \frac{2\delta}{N/2}\right) \leqslant \mathrm{kl}\left(\frac{1}{N/2}\sum_{a \in G^*_+} \mathbb{P}_{\pi, G^*_{(a)}}(A_a \cap B_a), \frac{1}{N/2}\sum_{a \in G^*_+} \mathbb{P}_{G^*}(A_a \cap B_a)\right) .$$

Thanks to the joint convexity of the $\mathrm{kl}$ function (see Gerchinovitz et al., 2020, corollary 3), we deduce that

$$\mathrm{kl}\left(\frac{1}{N/2}\sum_{a \in G^*_+} \mathbb{P}_{\pi, G^*_{(a)}}(A_a \cap B_a), \frac{1}{N/2}\sum_{a \in G^*_+} \mathbb{P}_{\pi, G^*}(A_a \cap B_a)\right)$$
$$\leqslant \frac{1}{N/2}\sum_{a \in G^*_+} \mathrm{kl}\left(\mathbb{P}_{\pi, G^*_{(a)}}(A_a \cap B_a), \mathbb{P}_{\pi, G^*}(A_a \cap B_a)\right) .$$

Now, we use the coupling lemma 20,

$$\mathbb{P}_{\pi, G^*_{(a)}}(A_a \cap B_a) = \mathbb{P}_1 \times \mathbb{P}_{\epsilon, U}(f_a(Y, Z, \epsilon, U))$$
$$\mathbb{P}_{\pi, G^*}(A_a \cap B_a) = \mathbb{P}_{-1} \times \mathbb{P}_{\epsilon, U}(f_a(Y, Z, \epsilon, U)) .$$

We use the data-processing inequality (see Gerchinovitz et al., 2020, corollary 2), for all $a \in G^*_+$,

$$\mathrm{kl}\left(\mathbb{P}_{\pi, G^*_{(a)}}(A_a \cap B_a), \mathbb{P}_{\pi, G^*}(A_a \cap B_a)\right) = \mathrm{kl}\left(\mathbb{P}_1 \times \mathbb{P}_{\epsilon, U}(f_a(Y, Z, \epsilon, U)), \mathbb{P}_{-1} \times \mathbb{P}_{\epsilon, U}(f_a(Y, Z, \epsilon, U))\right)$$
$$\leqslant \mathrm{KL}\left(\mathbb{P}_{-1} \otimes \mathbb{P}_{\epsilon, U}, \mathbb{P}_1 \otimes \mathbb{P}_{\epsilon, U}\right)$$
$$= \mathrm{KL}\left(\mathbb{P}_{-1}, \mathbb{P}_1\right) .$$

Gathering the previous inequalities, we obtain

$$\mathrm{kl}\left(\frac{1}{3} - 2\delta, \frac{4\delta}{N}\right) \leqslant \frac{1}{N/2}\sum_{a \in G^*_{(a)}} \mathrm{KL}(\mathbb{P}_{-1}, \mathbb{P}_1) .$$

We recall that $\rho^2 = \Delta^2/d$. With the explicit computation from Lemma 19, we have

$$\frac{d\sigma^4}{2\Delta^4}\mathrm{kl}\left(\frac{1}{3} - 2\delta, \frac{4\delta}{N}\right) \leqslant \frac{1}{N/2}\sum_{a \in G^*_+} tT = tT .$$

Finally, we have, using the definition of $t$ and $T$,

$$\mathbb{E}_{\pi,G^*}[\tau|\mathcal{Y}]^2 \geqslant \frac{d\sigma^4 KN}{72\Delta^4} \, \mathrm{kl}\left(\frac{1}{3} - 2\delta, \frac{4\delta}{N}\right) \ .$$

As it is true for any $\pi \in \Pi_{\mathcal{O}}$, take the inf in the last inequality over $\pi \in \Pi_{\mathcal{O}}$ and use Lemma 12 to get

$$T^*(\delta, \mathcal{E}(\Delta, \theta, \sigma, N, K, d)) \geqslant \frac{\sigma^2}{\Delta^2}\sqrt{\frac{dKN}{72} \, \mathrm{kl}\left(\frac{1}{3} - 2\delta, \frac{4\delta}{N}\right)} \ ,$$

this is exactly the inequality of Lemma 7.

## B.4. Proof of technical lemmas

### B.4.1. PROOF OF LEMMA 9

Let $N, K$ such that $N \geqslant 2K$. Let $\theta > 0$ such that $\mathcal{E}(\Delta, \theta, \sigma, N, K, d) \neq \emptyset$. We prove Lemma 9 assuming that $K$ is odd, the other case is simpler and can be proved with the same construction up to minor details.

Recall the expressions introduced before Lemma 9, $K' = K - 1$, $N' = K'\left\lfloor\frac{N - \lceil\theta N\rceil}{K'}\right\rfloor$ and $\theta' = 1/K'$.

Let $\pi$ being $\delta$-PAC on $\mathcal{E}(\Delta, \theta, \sigma, N, K, d)$, we will use $\pi$ to construct $\pi'$, an algorithm which is $\delta$-PAC on $\mathcal{E}(\Delta, \theta', \sigma, N', K', d)$.

Let $\nu' \in \mathcal{E}(\Delta, \theta', \sigma, N', K', d)$ be an environment with $K - 1$ perfectly balanced groups. We run the algorithm $\pi$ where we create the data $X_1, \ldots, X_{\tau^\pi}$ with the following coupling.

- If $A_t^\pi \in [N']$, we sample $X_t$ with the arm $A_t^\pi$ from $\nu'$.

- If $A_t^\pi \in [N' + 1; N - \lceil\theta N\rceil]$, we sample $X_t$ with $a_1$, the first arm from $\nu'$.

- If $A_t^\pi \in [N - \lceil\theta N\rceil + 1, N]$, we create $X_t = c$ where $c$ is an arbitrary large constant.

Equivalently, we have created the environment $\nu$ where the $N'$ first arms are the arms of $\nu$; the $\lceil\theta N\rceil$ last arms are in an artificial group associated to a Dirac in $c$, and the remaining arms are in the same group as $a_1$. The environment $\nu$ has a hidden partition $G_1^*, \ldots, G_K^*$ where $G_1^* = G_1' \cup [N' + 1; N - \lceil\theta N\rceil]$, $G_2^*, \ldots, G_{K-1}^* = G_2', \ldots, G_{K-1}'$, and $G_K^* = [N - \lceil\theta N\rceil + 1, N]$. By construction, this environment is in $\mathcal{E}(\Delta, \theta, \sigma, N, K, d)$. In particular, the balancedness is larger than $\theta$, and the minimal gap is larger than $\Delta$ if $c$ is large enough.

When $\pi$ reaches $\tau^\pi$, it outputs a partition of $[N]$, $\hat{G}_1^\pi, \ldots, \hat{G}_K^\pi$ , and we output $\hat{G}^{\pi'}$ as the partition defined by the restriction to $[N']$ of the partition $\hat{G}^\pi$. This is what we call the algorithm $\pi'$.

As $\pi$ is $\delta$-PAC on $\mathcal{E}(\Delta, \theta, \sigma, N, K, d)$, it holds that, with a probability $\mathbb{P}_{\pi,\nu}$ higher than $1 - \delta$, $\hat{G} \sim G^*$, and this implies that $\hat{G}^{\pi'} \sim G'$. Finally, we have $\mathbb{P}_{\pi',\nu'}(\hat{G}^{\pi'} \sim G') \geqslant \mathbb{P}_{\pi,\nu}(\hat{G}^\pi \sim G^*) \geqslant 1 - \delta$. This means that $\pi'$ is indeed $\delta$-PAC on $\mathcal{E}(\Delta, \theta', \sigma, N', K', d)$.

In terms of budget, we have $\tau^{\pi'} \leqslant \tau^\pi$, because the data provided from the last group are artificially created by the algorithm. We deduce that

$$\mathbb{E}_{\pi',\nu'}[\tau^{\pi'}] \leqslant \mathbb{E}_{\pi,\nu}[\tau^\pi] \leqslant \sup_{\nu \in \mathcal{E}(\Delta, \theta, \sigma, N, K, d)} \mathbb{E}_{\pi,\nu}[\tau] \ .$$

Then, we take the sup over $\nu' \in \mathcal{E}(\Delta, \theta', \sigma, N', K', d)$, and we have

$$T^*(\delta, \mathcal{E}(\Delta, \theta', \sigma, N', K', d)) \leqslant \sup_{\nu' \in \mathcal{E}(\Delta, \theta', \sigma, N', K', d)} \mathbb{E}_{\pi', \nu'}[\tau] \leqslant \sup_{\nu \in \mathcal{E}(\Delta, \theta, \sigma, N, K, d)} \mathbb{E}_{\pi, \nu}[\tau] \ .$$

Finally, we consider the inf over $\pi$ $\delta$-PAC on $\mathcal{E}(\Delta, \theta, \sigma, N, K, d)$, which concludes the proof of Lemma 9.

### B.4.2. PROOF OF LEMMA 12

Let $\pi'$ be a $\delta$-PAC algorithm on $\mathcal{E}(\Delta, \theta, \sigma, N, K, d)$.

We will use the algorithm $\pi'$ to construct an algorithm $\pi$, which is symmetric and $\delta$-PAC on the class $\mathcal{E}_{Sym}(G^*, \gamma)$ – see Definition 11. We will use the symmetries in the structure of the environment $\nu(G', \bar{\mu})$ when $\bar{\mu}$ is distributed with the prior $\gamma^{\otimes L}$ as the main argument to prove that $\pi$ will have the wanted properties. To avoid confusion, we index $(A_s^{\pi'})_s$, $\tau^{\pi'}$ and $\hat{G}^{\pi'}$ for the algorithm $\pi'$ and without $'$ for the algorithm $\pi$. As explained in the previous remark, the algorithm $\pi$ just need to perform well (i.e., being $\delta$-PAC) on the family $\mathcal{E}_{Sym}(G^*, \gamma)$, so we can use the offsets and the labels $l_1^*, \ldots, l_N^*$ to construct the algorithm $\pi$.

**Construction of $\pi$**

In this paragraph, we describe how we symmetrize a strategy $\pi'$ –see Algorithm 4. Let $G' \in \{G^*\} \cup \{G_{(a)}^*\}_{a \in G_+^*}$ being a partition. In order to make the reading easier, we use the notation $l_a^* = l^*(a)$ for all $a \in [N]$. For any arm $a$, we denote as $g'(a) \in \{-1, 1\}$ as the label such that the mean of $a$ is $\mu_a = g'(a)\bar{\mu}(l^*(a)) + C(l^*(a))$, in the environment $\nu(G', \bar{\mu})$, for any $\bar{\mu} \in \mathbb{R}^d$.

We need to define the behavior of $\pi$ when facing the environment $\nu(G', \bar{\mu})$ for any $\bar{\mu}$.

Define $\mathcal{S}$ as the set of permutations of $[N]$ that switch the blocks in $G^*$, that is to say if $\kappa \in \mathcal{S}$ then for all $l \in [L]$, $\exists l' \in [L]$, such that $\kappa(G_l^*) = G_{l'}^*$. For any $\kappa \in \mathcal{S}$, $\kappa$ naturally induces a permutation of $[L]$ denoted as $\tilde{\kappa}$ such that for all $a \in [N]$, $l^*(\kappa(a)) = \tilde{\kappa}(l^*(a))$.

First, the strategy $\pi$ uniformly samples a permutation $\kappa$ in $\mathcal{S}$ and a vector $\chi \in \{-1, 1\}^L$. From a rough perspective, the strategy $\pi$ will then apply the strategy $\pi'$ by permuting the blocks using $\kappa$ and reversing the means of each block using $\chi$.

---

**Algorithm 4** Symmetrization of $\pi'$.

**Input:** $\nu(G', \bar{\mu})$ an environment in $\mathcal{E}_{Sym}(G^*, \gamma)$
**Result:** $\hat{G}^\pi$, partition of $[N]$

1  $t = 1$
2  Take $\kappa \sim \mathcal{U}(\mathcal{S})$
3  Take $\chi \sim \mathcal{U}(\{-1, 1\}^L)$
4  **while** $t \leqslant \tau^{\pi'}(A_1^{\pi'}, X_1^{\pi'}, \ldots, A_{t-1}^{\pi'}, X_{t-1}^{\pi'})$ **do**
5    $\quad$ Choose an arm with $\pi'$ and get $A_t^{\pi'}(A_1^{\pi'}, X_1^{\pi'}, \ldots, A_{t-1}^{\pi'}, X_{t-1}^{\pi'}) \in [N]$.
6    $\quad$ Sample $X_t^\pi$ from $A_t^\pi := \kappa(A_t^{\pi'})$
7    $\quad$ Create the data $X_t^{\pi'} := \chi(\tilde{\kappa}(l^*(A_t^{\pi'}))) [X_t^\pi - C(\tilde{\kappa}(l^*(A_t^{\pi'})))] + C(l^*(A_t^{\pi'}))$
8    $\quad$ t=t+1
9  **end**
10  Compute $\hat{G}^{\pi'}(A_1^{\pi'}, X_1^{\pi'}, \ldots, A_\tau^{\pi'}, X_\tau^{\pi'}) := \hat{G}_1^{\pi'}, \ldots, \hat{G}_K^{\pi'}$
11  **return** $\hat{G}_1^\pi, \ldots, \hat{G}_K^\pi := \kappa(\hat{G}_1^{\pi'}), \ldots, \kappa(\hat{G}_K^{\pi'})$

---

Within the procedure $\pi$, we run algorithm $\pi'$ with modified data $X_1^{\pi'}, \ldots, X_\tau^{\pi'}$. At time $t$, the algorithm $\pi'$ chooses to sample the arm $A_t^{\pi'}$, where the decision is based on the data $(X_s^{\pi'}, A_s^{\pi'})_{s \leqslant t-1}$. Instead of sampling the arm chosen by $\pi'$, the algorithm $\pi$ samples $X_t^\pi$ from the arm $A_t^\pi := \kappa(A_t^{\pi'})$ and sends the data $X_t^{\pi'}$ to $\pi'$, according to the formula

$$X_t^{\pi'} = \chi(\tilde{\kappa}(l^*(A_t^{\pi'})))[X_t^\pi - C(\tilde{\kappa}(l^*(A_t^{\pi'})))] + C(l^*(A_t^{\pi'})) ,$$

where we recall that $C(l)$ is the offset associated to block $l$. When $\pi'$ decides to stop, $\pi$ also stops; i.e., $\tau^\pi(X_1^\pi, A_1^\pi, \ldots, X_\tau^\pi, A_\tau^\pi) = \tau^{\pi'}(X_1^{\pi'}, A_1^{\pi'}, \ldots, X_{\tau'}^{\pi'}, A_{\tau'}^{\pi'})$. Then, $\pi'$ outputs a partition $\hat{G}^{\pi'} = \hat{G}_1^{\pi'}, \ldots, \hat{G}_K^{\pi'}$ based on the modified data, and $\pi$ outputs $\hat{G}_1^\pi, \ldots, \hat{G}_K^\pi := \kappa(\hat{G}_1^{\pi'}), \ldots, \kappa(\hat{G}_K^{\pi'})$.

**Lemma 22** *Take $\kappa \in \mathcal{S}$, and $\chi \in \{-1,1\}^L$. For all $l \in [L]$, define $\bar{\mu}_\kappa(l) := \bar{\mu}(\tilde{\kappa}(l))$. As $\bar{\mu}$ is sampled according to $\gamma^{\otimes L}$, then $\bar{\mu}_\kappa$ follows the same prior $\gamma^{\otimes L}$. Define $G'(\kappa, \chi)$ as a partition of $[N]$ into $2L$ groups such that for all $(l,g) \in [L] \times \{-1; 1\}$, then*

$$G'(\kappa, \chi)_{l,g} = \{a \in [N]; l^*(a) = l, \text{ and } g'(\kappa(a))\chi(\tilde{\kappa}(l^*(a))) = g\} = \kappa^{-1}\left(G'_{\tilde{\kappa}(l), g\chi(\tilde{\kappa}(l))}\right) .$$

*Conditionally on $\kappa, \chi, \bar{\mu}$, the modified data $X_s^{\tau'}$ are distributed according to the probability induced by the interaction between $\pi'$ and the environment $\nu(G'(\kappa, \chi), \bar{\mu}_\kappa)$, after integration on the prior $\gamma$, we have*

$$\mathbb{P}_{\pi, G'}(\cdot | \mathcal{Y}, \kappa, \chi) = \mathbb{P}_{\pi', G'(\kappa, \chi)}(\cdot | \mathcal{Y}) .$$

**Remark 23** *It is very important to note that, as $G'$ is a partition with $K = 2L$ groups of the same size, the partition $G'(\kappa, \chi)$ is also balanced.*

**Proof** [Proof of Lemma 22] Let $\bar{\mu} \in (\mathbb{R}^d)^L$ be a realization of the prior $\gamma^{\otimes L}$.

When $\pi'$ tries to sample the arm $a = A_t^{\pi'}$, we sample in fact $\kappa(a)$. Using the Gaussian assumption on the data, and the expression of the centers of the environment $\nu(G', \bar{\mu})$, it holds that

$$X_t^\pi = g'(\kappa(a))\bar{\mu}(l^*(\kappa(a))) + C(l^*(\kappa(a))) + \epsilon_s = g'(\kappa(a))\bar{\mu}(\tilde{\kappa}(l^*(a))) + C(\tilde{\kappa}(l^*(a))) + \epsilon_t ,$$

where $\epsilon_t \sim \mathcal{N}(0, \sigma^2 I_d)$. We used also in the second equality that $\kappa$ induces a permutation of the blocks, so that $l^*(\kappa(a)) = \tilde{\kappa}(l^*(a))$.

We now decompose $X_t^{\pi'}$, using the expression defined Line 7 of Algorithm 4. Assuming that $\tilde{\kappa}(l^*(a)) = m \in [L]$, we have

$$\begin{aligned} X_t^{\pi'} &= \chi(m)[X_t^\pi - C(m)] + C(l^*(a)) \\ &= \chi(m)[g'(\kappa(a))\bar{\mu}(m) + \epsilon_t] + C(l^*(a)) . \end{aligned}$$

We develop and reorganise the terms, and we use the expression $\bar{\mu}_\kappa(l) = \bar{\mu}(\tilde{\kappa}(l))$,

$$\begin{aligned} X_t^{\pi'} &= g'(\kappa(a))\chi(m)\bar{\mu}(m) + \chi(m)\epsilon_t + C(l^*(a)) \\ &= g'(\kappa(a))\chi(m)\bar{\mu}_\kappa(l^*(a)) + \chi(m)\epsilon_t + C(l^*(a)) . \end{aligned}$$

As $\epsilon_t$ is symmetric with respect to 0, then $\epsilon_t' := \chi(m)\epsilon_t$ is distributed as a normal distribution $\mathcal{N}(0, \sigma^2 I_d)$. Besides, the $(\epsilon_t')_t$ are independent. The arm $a$ appears to $\pi'$ to have a mean $C(l^*(a)) +$

$\tilde{g}\bar{\mu}_\kappa(l^*(a))$, where $\tilde{g} = g'(\kappa(a))\chi(\tilde{\kappa}(l^*(a))) \in \{-1, 1\}$. It appears then that the data received by $\pi'$ are distributed as $\nu(G'(\kappa, \chi), \bar{\mu}_\kappa)$, where, for all $(g, l) \in [L] \times \{-1, 1\}$,

$$G'(\kappa, \chi)_{l,g} = \{a \in [N]; l^*(a) = l, \text{ and } g'(\kappa(a))\chi(\tilde{\kappa}(l)) = g\} ,$$

which proves the first part of the lemma.

The second expression for $G'(\kappa, \chi)_{l,g}$ is now obtained using the fact that $\kappa$ permutes the blocks , so that $l^*(\kappa(a)) = \tilde{\kappa}(l)$ and also that $\chi(\tilde{\kappa}(l)) \in \{-1, 1\}$.

$$\{a \in [N]; l^*(a) = l, \text{ and } g'(\kappa(a))\chi(\tilde{\kappa}(l)) = g\} = \{a \in [N]; l^*(\kappa(a)) = \tilde{\kappa}(l), \text{ and } g'(\kappa(a)) = g\chi(\tilde{\kappa}(l))\}$$
$$= \kappa^{-1}\left(G'_{\tilde{\kappa}(l), g\chi(\tilde{\kappa}(l))}\right) .$$

Finally, if $\bar{\mu} \sim \gamma^{\otimes L}$, by exchangeability of the law of $\gamma^{\otimes L}$, and as $\tilde{\kappa}$ is a permutation of $[L]$, the vector $(\bar{\mu}(\tilde{\kappa}(l)))_{l \in [L]}$ is distributed as $(\bar{\mu}(l))_{l \in [L]}$. We also highlight that the event $\mathcal{Y} = \bigcap_{l \in [L]}\{\Delta/2 \leqslant \|\bar{\mu}(l)\| \leqslant \Delta(\beta - 1)/2\} = \bigcap_{l \in [L]}\{\Delta/2 \leqslant \|\bar{\mu}_\kappa(l)\| \leqslant \Delta(\beta - 1)/2\}$ remains the same, so that we have the equality of the laws

$$\mathbb{P}_{\pi, G'}(\cdot|\mathcal{Y}, \kappa, \chi) = \mathbb{P}_{\pi', G'(\kappa, \chi)}(\cdot|\mathcal{Y}) .$$

∎

### Correction of $\pi$

We now deduce that $\pi$ is $\delta$-PAC on $\mathcal{E}_{Sym}(G^*, \gamma)$ –see Definition 11.

By construction of the algorithm, and with the definition of $G'(\kappa, \chi)$ given in Lemma 22, we have conditionally on $\kappa, \chi$, and $\bar{\mu}$,

$$\mathbb{P}_{\pi, G', \bar{\mu}}(\hat{G}^\pi \sim G'|\kappa, \chi) = \mathbb{P}_{\pi', G'(\kappa, \chi), \bar{\mu}_\kappa}\left(\hat{G}^{\pi'} \sim G'(\kappa, \chi)\right) .$$

If $\bar{\mu} \in \mathcal{Y}$, then we have also $\bar{\mu}_\kappa \in \mathcal{Y}$ and the environment $\nu(G'(\kappa, \chi), \bar{\mu}_\kappa)$ is in $\mathcal{E}(\Delta, \theta, \sigma, N, K, d)$. We recall that $\pi$ is $\delta$-PAC on $\mathcal{E}(\Delta, \theta, \sigma, N, K, d)$, we then have

$$\mathbb{P}_{\pi', G'(\kappa, \chi), \bar{\mu}_\kappa}\left(\hat{G}^{\pi'} \sim G'(\kappa, \chi)\right)\mathbb{1}_\mathcal{Y} \geqslant (1 - \delta)\mathbb{1}_\mathcal{Y} .$$

The conclusion then follow by integrating over the law of $\kappa, \chi$, and $\bar{\mu}$ to obtain $\mathbb{P}_{\pi, G'}(\hat{G}^\pi \sim G'|\mathcal{Y}) \geqslant 1 - \delta$, and $\pi$ is indeed $\delta$-PAC on $\mathcal{E}_{Sym}(G^*, \gamma)$.

### Symmetry of $\pi$

We want to prove that $\pi$ is symmetric as defined in Definition 11. Take $a_1, a_2 \in [N]^2$ two arms and assume that $a_1 \in G'_{l_1, g_1}$ and $a_2 \in G'_{l_2, g_2}$.

First, we recall that $A_t^\pi = \kappa(A_t^{\pi'})$ so that,

$$N_{a_1}^\pi(\tau) = \sum_{s=1}^\tau \mathbb{1}_{A_s^\pi = a_1} = \sum_{s=1}^\tau \mathbb{1}_{\kappa(A_s^{\pi'}) = a_1} = N_{\kappa^{-1}(a_1)}^{\pi'}(\tau) .$$

We now use the expression of the uniform laws that follows $\kappa, \chi$ and Lemma 22,

$$\mathbb{E}_{\pi,G'}[N_{a_1}^\pi(\tau)|\mathcal{Y}] = \frac{1}{2^L\#\mathcal{S}}\sum_{\kappa\in\mathcal{S}}\sum_{\chi\in\{-1,1\}^L}\mathbb{E}_{\pi,G'}[N_{a_1}^\pi(\tau)|\mathcal{Y},\kappa,\chi]$$

$$= \frac{1}{2^L\#\mathcal{S}}\sum_{\kappa\in\mathcal{S}}\sum_{\chi\in\{-1,1\}^L}\mathbb{E}_{\pi',G'(\kappa,\chi)}[N_{\kappa^{-1}(a_1)}^{\pi'}(\tau)|\mathcal{Y}] \ .$$

We construct $\kappa'\in\mathcal{S}$ a permutation which switches the blocks of $a_1$ and $a_2$, while switching $a_1$ and $a_2$, take

$$\forall\epsilon\in\{-1,1\}, \kappa'(G'_{l_1,\epsilon g_1})=G'_{l_2,\epsilon g_2} \ ; \qquad \forall\epsilon\in\{-1,1\}, \kappa'(G'_{l_2,\epsilon g_2})=G'_{l_1,\epsilon g_1} \ ;$$

$$\kappa'(a_1)=a_2, \kappa'(a_2)=a_1 \ , \text{ and} \qquad \forall c\in[N], \text{ if } l^*(c)\notin\{l_1,l_2\}, \kappa'(c)=c \ .$$

The permutation $\kappa'$ exists because the groups of $G'$ have exactly the same size.

We also define $\chi'\in\{-1,1\}^L$ with

$$\chi'(l)=\chi(l) \text{ if } l\notin\{l_1,l_2\} \ , \qquad \chi'(l_1)=(g_1 g_2)\chi(l_2) \ , \qquad \text{and } \chi'(l_2)=(g_2 g_1)\chi(l_1) \ .$$

Note that $\kappa'\in\mathcal{S}$. When we consider $\mathcal{S}$ is a group of permutation we see that $\kappa'\mathcal{S}=\mathcal{S}$. Moreover, as the law of $\chi(1),\ldots,\chi(L)$ is exchangeable and symmetric with respect to 0, $\chi'$ and $\chi$ follow the same distribution.

It implies that we can use a change of variable in the sum,

$$\mathbb{E}_{\pi,G'}[N_{a_1}^\pi(\tau)|\mathcal{Y}] = \frac{1}{2^L\#\mathcal{S}}\sum_{\kappa\in\mathcal{S}}\sum_{\chi\in\{-1,1\}^L}\mathbb{E}_{\pi',G'(\kappa,\chi)}[N_{\kappa^{-1}(a_1)}^{\pi'}(\tau)|\mathcal{Y}]$$

$$= \frac{1}{2^L\#\mathcal{S}}\sum_{\kappa\in\mathcal{S}}\sum_{\chi\in\{-1,1\}^L}\mathbb{E}_{\pi',G'(\kappa'\kappa,\chi')}[N_{(\kappa'\kappa)^{-1}(a_1)}^{\pi'}(\tau)|\mathcal{Y}] \ .$$

Now, for any $\kappa\in\mathcal{S}$, $(\kappa'\kappa)^{-1}(a_1)=\kappa^{-1}(\kappa')^{-1}(a_1)=\kappa^{-1}(a_2)$ because $\kappa'$ exchanges $a_1$ and $a_2$.

Then, fix $\chi$ and $\kappa$ and consider the partition $G'(\kappa'\kappa,\chi')$. We want to prove that, $G'(\kappa'\kappa,\chi')=G'(\kappa,\chi)$. By definition (Lemma 22), we have to prove that $\forall b\in[N]$,

$$g'(\kappa(b))\chi(\tilde\kappa(l^*(b))) = g'(\kappa'\kappa(b))\chi'(\tilde{\kappa'}\tilde\kappa(l^*(b))) \ , \tag{24}$$

We prove Equation (24).

Take $\epsilon\in\{-1,1\}$ and $b\in\kappa^{-1}(G'_{l_1,\epsilon g_1})$, by construction, $\tilde{\kappa'}$ is the transposition $(l_1\ l_2)$, and we have

$$\chi'(\tilde{\kappa'}\tilde\kappa(l^*(b))) = \chi'(\tilde{\kappa'}(l_1)) = \chi'((l_1\ l_2)(l_1)) = \chi'(l_2) = (g_1 g_2)\chi(l_1) \ .$$

Besides, we have $\chi(\tilde\kappa(l^*(b)))=\chi(l_1)$. Moreover, $\kappa(b)\in G'_{l_1,\epsilon g_1}$ and then $\kappa'(\kappa(b))\in G'_{l_2,\epsilon g_2}$, i.e. , $g'(\kappa'\kappa(b))=\epsilon g_2$.

The equality in Equation (24) therefore holds for all $b$ in $\kappa^{-1}(G'_{l_1,\epsilon g_1})$,

$$g'(\kappa(b))\chi(\tilde\kappa(l^*(b))) = \epsilon g_2(g_1 g_2)\chi(l_1) = \epsilon g_1\chi(l_1) = g'(\kappa(b))\chi(\tilde\kappa(l^*(b))) \ .$$

The labels $l_1$ and $l_2$ play the symmetric role, so we also have the equality of Equation (24) for $b \in \kappa^{-1}(G'_{l_2, \epsilon g_2})$. Finally, if $l^*(\kappa(b)) \notin \{l_1, l_2\}$, then by construction of $\chi'$ and $\kappa'$, we have $\kappa'(\kappa(b)) = \kappa(b)$ and $\chi'(\tilde{\kappa}'\tilde{\kappa}(l^*(b))) = \chi'(\tilde{\kappa}(l^*(b))) = \chi(\tilde{\kappa}(l^*(b)))$.

Equation (24) being proved, we have finally,

$$
\begin{aligned}
\mathbb{E}_{\pi,G'}[N^\pi_{a_1}(\tau)|\mathcal{Y}] &= \frac{1}{2^L \# \mathcal{S}} \sum_{\kappa \in \mathcal{S}} \sum_{\chi \in \{-1,1\}^L} \mathbb{E}_{\pi',G'(\kappa'\kappa,\chi')}[N^{\pi'}_{(\kappa'\kappa)^{-1}(a_1)}(\tau)|\mathcal{Y}] \\
&= \frac{1}{2^L \# \mathcal{S}} \sum_{\kappa \in \mathcal{S}} \sum_{\chi \in \{-1,1\}^L} \mathbb{E}_{\pi',G'(\kappa,\chi)}[N^{\pi'}_{\kappa^{-1}(a_2)}(\tau)|\mathcal{Y}] \\
&= \mathbb{E}_{\pi,G'}[N^\pi_{a_2}(\tau)|\mathcal{Y}] \ .
\end{aligned}
$$

This proves that $\mathbb{E}_{\pi,G'}[N^\pi_{a_1}(\tau)|\mathcal{Y}]$ is independent of $a_1$ and equal to $\mathbb{E}_{\pi,G'}[\tau|\mathcal{Y}]/N$. Now, using the same method as above with $\kappa' = (a\ b_a)$, we also deduce that $\mathbb{E}_{\pi,G^*_{(a)}}[\tau|\mathcal{Y}] = \mathbb{E}_{\pi,G^*}[\tau|\mathcal{Y}]$ does not depend on $a$.

This proves that $\pi$ is symmetric as defined in Definition 11.

We have proved that $\pi$ is $\delta$-PAC and symmetric on $\mathcal{E}_{Sym}(G^*, \gamma)$. It remains to conclude for the proof of the lemma.

**Budget of $\pi$**   By construction of the algorithm, we have

$$
\mathbb{E}_{\pi,G^*}[\tau^\pi|\mathcal{Y}, \kappa, \chi] = \mathbb{E}_{\pi',G^*(\kappa,\chi)}[\tau^{\pi'}|\mathcal{Y}] \leqslant \sup_{\nu \in \mathcal{E}(\Delta,\theta,\sigma,N,K,d)} \mathbb{E}_{\pi',\nu}[\tau'] \ ,
$$

since, on the event $\mathcal{Y}$, we have $\nu(G^*(\kappa,\chi), \bar{\mu}) \in \mathcal{E}(\Delta, \theta, \sigma, N, K, d)$. We now use the fact that $\pi$ is in $\Pi_{Sym}(\delta, \mathcal{E}_{Sym}(G^*, \gamma))$, so that

$$
T^*_{Sym}(\delta, \mathcal{E}_{Sym}(G^*, \gamma)) \leqslant \mathbb{E}_{\pi,G^*}[\tau|\mathcal{Y}] \leqslant \sup_{\nu \in \mathcal{E}(\Delta,\theta,\sigma,N,K,d)} \mathbb{E}_{\pi',\nu}[\tau'] \ .
$$

Finally, we prove Lemma 12 by taking the inf over $\pi' \in \Pi(\delta, \mathcal{E}(\Delta, \theta, \sigma, N, K, d))$.

### B.4.3. PROOFS FROM APPENDIX B.3

**Proof** [Proof of Lemma 16] Let $l \in [L]$ and define $Z = \|\bar{\mu}(l)\|^2/\rho^2$. We have $\bar{\mu}(l) \sim \mathcal{N}(0, \rho^2)$ with $\rho^2 = \Delta^2/d$, then $Z \sim \chi_2(d)$ is a chi-square distribution with $d$ degrees of freedom. We apply the Laurent-Massart inequality with $x = (3/8)^2 d$ – see 43,

$$
\mathbb{P}\left(\|\bar{\mu}(l)\| < \frac{\Delta}{2}\right) = \mathbb{P}\left(Z - d < \frac{\Delta^2}{4\rho^2} - d\right) = \mathbb{P}\left(Z - d < -2\sqrt{d(3/8)^2 d}\right) \leqslant \exp(-(3/8)^2 d) \ .
$$

Then, we notice that $\beta = 4$ satisfies $(\beta - 1)^2/4 \geqslant 1 + 2(3/8)^2 + 2\sqrt{(3/8)^2}$, we have

$$
\begin{aligned}
\mathbb{P}\left(\|\bar{\mu}(l)\| > (\beta - 1)\frac{\Delta}{2}\right) &= \mathbb{P}\left(Z - d > ((\beta - 1)^2/4 - 1)d\right) \\
&\leqslant \mathbb{P}\left(Z - d > 2\sqrt{d(3/8)^2 d} + 2(3/8)^2 d\right) \ .
\end{aligned}
$$

Now, we use the other side of Laurent-Massart inequality with $x = (3/8)^2 d$ to obtain

$$\mathbb{P}\left(\|\bar{\mu}(l)\| > (\beta - 1)\frac{\Delta}{2}\right) \leqslant \exp(-(3/8)^2 d) \ .$$

We recall that we assumed that $d \geqslant (8/3)^2 \log(K/\delta)$, and so $\exp(-(3/8)^2 d) \leqslant \delta/K$. A union bound on $L = K/2$ ensures that Lemma 16 holds. ∎

**Proof** [Proof of Lemma 20]

Let $a \in G_+^*$ be an arm labelled by $(l_a^*, 1)$ in $G^*$ and let $Y, Z \sim \mathbb{P}_{-1}$ –see Definition 18. We fix an algorithm $\pi$ for the clustering with bandit feedback problem on $\mathcal{E}(G^*, \gamma_2)$. The algorithm $\pi$ is characterized by three families of measurable functions $(\pi_s, \tau_s, f_s)_{s \geqslant 1}$ where for all $s \geqslant 1$

- $A_s = \pi_s((A_1, X_1), \ldots, (A_{s-1}, X_{s-1}); U_s)$

- $\tau = \min\{t \geqslant 1 ; \tau_s((A_1, X_1), \ldots, (A_s, X_s); U_s) = 1\}$

- $\hat{g} = f_\tau((A_1, X_1), \ldots, (A_\tau, X_\tau); U_\tau)$

Here, the sequence $(U_s)$ captures the fact that $\pi$ can use some external randomness to make decisions. We define $N_{s,a} = \sum_{u=1}^s \mathbb{1}_{\{A_u \in \{a, b_a\}\}}$ and $M_s = \sum_{u=1}^s \mathbb{1}_{\{A_u \in G_{l_a^*}^*\}}$. We consider the event $B_a$ on which the inequalities $N_{\tau,a} = N_a(\tau) + N_{b_a} \leqslant t_a = t$ and $M_\tau \leqslant M_{l_a^*}(\tau) \leqslant T_a = T$ holds. Then, the data collected $(X_1, \ldots, X_\tau)$ when $\pi$ interacts with $\nu(G_{(a)}^*, \bar{\mu})$ and $\bar{\mu} \sim \gamma^{\otimes L}$ can be constructed with $Y, Z, \epsilon, U$ using the following coupling.

First, we create the observations from arms that belongs to a block different than the one of $a$, using the variables $(\epsilon_u)_{u \geqslant 1}$. We sample once and for all $(L - 1)$ centers by defining for any $l \in [L] \setminus \{l_a^*\}$,

$$\bar{\mu}(l) = \rho \epsilon_l \ ,$$

we observe that $(\bar{\mu})_{l \neq l_a^*} \sim \gamma_2^{\otimes(L-1)}$.

Then, for any $s \geqslant 1$, if $A_s \in G_l^*$ with $l \neq l_a^*$, we can create $X_s$ with the expression

$$X_s = C(l) + g\bar{\mu}(l) + \sigma \epsilon_{s+L} \ .$$

Now, for $s \geqslant 1$, when $A_s \in G_{l_a^*}^*$, we use $Y, Z$,

- $X_s = C(l_a^*) + Y_{N_{s,a}}$ if $A_s = a$

- $X_s = C(l_a^*) - Y_{N_{s,a}}$ if $A_s = b_a$

- $X_s = C(l_a^*) + g_{A_s}^* Z_{M_s}$ if $A_s \in G_{l_a^*}^* \setminus \{a, b_a\}$

We highlight that the law of $Y, Z$ is a marginal distribution that captures the fact that the data obtained from the block $G_{l_a^*}^*$ are obtained using the prior $\gamma$ for $\bar{\mu}(l_a^*)$.

From there, it is possible to give (explicitly) a function $f_a$ measurable with respect to $Y, Z, \epsilon, U$ such that $A_a \cap B_a = f(Y, Z, \epsilon, U)$ where the equality holds in law with respect to $\mathbb{P}_{\pi, G_{(a)}^*}$ (integrated with respect to $\bar{\mu}$). If we use the same measurable function $f_a$ with $X, Y \sim \mathbb{P}_1$, then $A_a \cap B_a = f(Y, Z, \epsilon, U)$ where the equality holds with respect to $\mathbb{P}_{\pi, G^*}$. ∎

**Proof** [Proof of Lemma 21 ] We recall that $\pi$ is a $\delta$-PAC algorithm for the problem of clustering with bandit feedback with an oracle. We recall that $A_a = \{\hat{G} \sim G^*_{(a)}\}$. By construction of the partitions $G^*_{(a)}$, these partitions are not equivalent (for the relation $\sim$). We highlight that this is due to the fact that all the groups contain more than two arms. The events $(A_a)_a$ are disjoints, and $\sqcup_{a \in G^*_+}(A_a \cap B_a) \subset \{\hat{G} \not\sim G^*\}$.

Now, we have directly

$$\mathbb{P}_{\pi,G^*}(\cup_{a \in [N] \setminus S} A_a \cap B_a) \leqslant \mathbb{P}_{\pi,G^*}(\hat{G} \not\sim G^*) \ .$$

By definition, $\pi$ is $\delta$-PAC on $\mathcal{E}_{Sym}(G^*, \gamma_2)$, we have

$$\mathbb{P}_{\pi,G^*}(\hat{G} \not\sim G^*) = \mathbb{P}_{\pi,G^*}(\hat{G} \not\sim G^* | \mathcal{Y})\mathbb{P}_{\gamma^{\otimes L}}(\mathcal{Y}) + \mathbb{P}_{\pi,G^*}(\hat{G} \not\sim G^* | \mathcal{Y}^c)\mathbb{P}_{\gamma^{\otimes L}}(\mathcal{Y}^c)$$
$$\leqslant \delta + \mathbb{P}_{\gamma^{\otimes L}}(\mathcal{Y}^c) \leqslant 2\delta$$

For the second point of the lemma, we fix $a \in G^*_+$.

$$\mathbb{P}_{\pi,G^*_{(a)}}((A_a \cap B_a)^c) = \mathbb{P}_{\pi,G^*_{(a)}}(A^c_a \cup B^c_a | \mathcal{Y})\mathbb{P}_{\gamma^{\otimes L}}(\mathcal{Y}) + \mathbb{P}_{\pi}(A^c_a \cup B^c_a | \mathcal{Y}^c)\mathbb{P}_{\gamma^{\otimes L}}(\mathcal{Y}^c)$$
$$\leqslant \mathbb{P}_{\pi,G^*_{(a)}}(A^c_a \cup B^c_a | \mathcal{Y}) + \mathbb{P}_{\gamma^{\otimes L}}(\mathcal{Y}^c)$$
$$\leqslant \mathbb{P}_{\pi,G^*_{(a)}}(A^c_a | \mathcal{Y}) + \mathbb{P}_{\pi,G^*_{(a)}}(B^c_a | \mathcal{Y}) + \mathbb{P}_{\gamma^{\otimes L}}(\mathcal{Y}^c) \ .$$

Now, $\pi$ is $\delta$-PAC which implies that

$$\mathbb{P}_{\pi,G^*_{(a)}}(A^c_a | \mathcal{Y}) = \mathbb{P}_{\pi,G^*_{(a)}}(\hat{G} \not\sim G^*_{(a)} | \mathcal{Y}) \leqslant \delta \ .$$

For the second term, we use Markov inequality with respect to the distribution $\mathbb{P}_{\pi,G^*_{(a)}}(\cdot | \mathcal{Y})$. We recall that $\pi$ satisfies a symmetry property and that $t = t_a = 3\mathbb{E}_{\pi,G^*_{(a)}}[N_a(\tau) + N_{b_a}(\tau) | \mathcal{Y}]$ and $T = T_a = 3\mathbb{E}_{\pi,G^*_{(a)}}[M_{l^*_a}(\tau) | \mathcal{Y}]$. We also recall that $B_a = \{N_a(\tau) + N_{b_a}(\tau) \leqslant t_a\} \cap \{M_{l^*_a}(\tau) \leqslant T_a\}$. We have with Markov inequality

$$\mathbb{P}_{\pi,G^*_{(a)}}(B^c_a | \mathcal{Y}) \leqslant \mathbb{P}_{\pi,G^*_{(a)}}(N_a(\tau) + N_{b_a}(\tau) > t_a | \mathcal{Y}) + \mathbb{P}_{\pi,G^*_{(a)}}(M_{l^*_a}(\tau) > T_a | \mathcal{Y}) \leqslant \frac{1}{3} + \frac{1}{3} = \frac{2}{3} \ .$$

This concludes the proof of Lemma 21. ∎

**Proof** [Proof of Lemma 19]

Let $g \in \{-1, 1\}$ and take $\mathbb{P}_g$ defined in Definition 18 with the Gaussian prior. We have $\mu \sim \mathcal{N}(0, \rho^2 I_d)$ and conditionally on $\mu$,

- $Y_1, \ldots, Y_t, Z_1, \ldots, Z_T$ are independent ;

- $\forall r \in [t], Y_r \sim \mathcal{N}(g\mu, \sigma^2 I_d)$

- $\forall s \in [T], Z_r \sim \mathcal{N}(\mu, \sigma^2 I_d)$.

First, $Y_1, \ldots, Y_t, Z_1, \ldots, Z_T$ have i.i.d coordinates and so has $\mu$. Then, it is enough to prove Lemma 19 in dimension 1. The general case will be obtained by multiplying by $d$ the result for dimension 1. We assume then that $d = 1$, and we want to prove that $\mathrm{KL}(\mathbb{P}_{-g}, \mathbb{P}_g) = \frac{2tT\rho^4}{\sigma^4 + \sigma^2\rho^2(T+t)}$.

Now, we specify the distribution of the vector $Y, Z$. As $\mu$ follows a Gaussian distribution, the vector $(X, Y) = Y_1, \ldots, X_t, Z_1, \ldots, Z_T$ is a Gaussian vector.

With the law of total variance, we have $Y, Z \sim \mathcal{N}(0, \Sigma_g)$ where $\Sigma_g$ is the covariance (square) matrix of size $(T + t)$. The matrix $\Sigma_g$ is defined as follows:

$$\Sigma_g = \sigma^2 I_{t+T} + \rho^2 \begin{pmatrix} J_{t,t} & g J_{t,T} \\ g J_{T,t} & J_{T,T} \end{pmatrix} =: \sigma^2 I_{(t+T)} + \rho^2 H_g \ ,$$

where $I_{(t+T)}$ is the identity matrix of size $(T + t)$, and we define $J_{t,T}$ being the rectangle matrix of size $t \times T$ where all entries are equal to 1.

We observe that $H_g$ has a particular shape, in particular, $H_g^2 = (T + t) H_g$. As a consequence, it is easy to compute its inverse. We have:

$$\Sigma_g^{-1} = \frac{1}{\sigma^2} I_{(t+T)} + \frac{1}{\tilde{\rho}^2} H_g \ ;$$

with $\tilde{\rho}^2 = -\frac{\sigma^2}{\rho^2}(\sigma^2 + \rho^2(t + T))$ .

Now,

$$\Sigma_g^{-1} \Sigma_{-g} - I_{(T+t)} = \left( \frac{1}{\sigma^2} I_{(T+t)} + \frac{1}{\tilde{\rho}^2} H_g \right) \left( \sigma^2 I_{(T+t)} + \rho^2 H_{-g} \right) - I_{(T+t)} = \frac{\rho^2}{\sigma^2} H_{-g} + \frac{\sigma^2}{\tilde{\rho}^2} H_g + \frac{\rho^2}{\tilde{\rho}^2} H_g H_{-g} \ ,$$

where we compute

$$H_g H_{-g} = (t - T) \begin{pmatrix} J_{t,t} & -g J_{t,T} \\ g J_{T,t} & -J_{T,T} \end{pmatrix} \ .$$

Finally, with the formula for the KL divergence between two multidimensional Gaussian distribution, we have

$$\mathrm{KL}(\mathbb{P}_{-g}, \mathbb{P}_g) = \frac{1}{2} \left( \log \frac{|\Sigma_g|}{|\Sigma_{-g}|} + Tr(\Sigma_g^{-1} \Sigma_{-g} - I_{(T+t)}) + 0 \Sigma_g^{-1} 0 \right)$$

$$= \frac{1}{2} \left( \frac{\rho^2}{\sigma^2} Tr(H_{-g}) + \frac{\sigma^2}{\tilde{\rho}^2} Tr(H_g) + \frac{\rho^2}{\tilde{\rho}^2} Tr(H_g H_{-g}) \right)$$

$$= \frac{1}{2} \left( \frac{\rho^2(t + T)}{\sigma^2} - \frac{\rho^2(T + t)}{\sigma^2 + \rho^2(T + t)} - \frac{\rho^4(T - t)^2}{\sigma^2(\sigma^2 + \rho^2(t + T))} \right)$$

$$= \frac{1}{2} \left( \frac{\rho^4((t + T)^2 - (T - t)^2)}{\sigma^2(\sigma^2 + \rho^2(T + t))} \right) = \frac{2tT\rho^4}{\sigma^4 + \sigma^2\rho^2(T + t)} \ .$$

This concludes the computation of $\mathrm{KL}(\mathbb{P}_{-g}, \mathbb{P}_g)$. ∎

## Appendix C. Analysis of ACB

In this section, we establish that ACB 1 is $\delta$-PAC and we control its budget thereby proving the part of Theorem 5 pertaining to ACB.

**Theorem 24** *Let $\delta > 0$. Let $\Delta > 0$, $\theta > 0$ be the two parameters used in the design of ACB, such that $\mathcal{E}(\Delta, \theta, \sigma, N, K, d) \neq \emptyset$. The ACB algorithm (1) is $\delta$-PAC on $\mathcal{E}(\Delta, \theta, \sigma, N, K, d)$.*
*Moreover, define $\tau_{ACB}$ for the budget of $ACB(\delta, \Delta, \theta)$. There exist two universal constants $c$ and $c'$ (with $c$ small), independent of all the parameters $\Delta, \theta, \sigma, N, K, d$ and such that for any environment $\nu$ in $\mathcal{E}(\Delta, \theta, \sigma, N, K, d)$, if we assume that $\frac{\log(K)}{\theta} \leqslant N$, then $\mathbb{E}_{ACB,\nu}[\tau_{ACB}] \leqslant cN + c'A$, and $\tau_{ACB} \leqslant cN + c'(A + B)$ almost surely, where*

$$A = \frac{\sigma^2}{\Delta^2}\left[ N \log\left(N/\delta\right) + \sqrt{dNK \log\left(N/\delta\right)} + \sqrt{d}\frac{\log(K)}{\theta}\right]$$

$$B = \frac{\log(K/\delta)}{\theta} + \frac{\sigma^2}{\Delta^2}\frac{1}{\theta}\log\left(\frac{K}{\delta}\right)\left[\sqrt{d} + \log\log\left(\frac{1}{\theta\delta}\right)\right] \ .$$

In fact, Theorem 24 is a straightforward consequence of the two following lemmas that separately consider the two sub-routines SRI and ADC.

**Lemma 25 (Analysis of SRI)**

*Let $\delta > 0$ be fixed. Let $\Delta > 0$ and $1/K > \theta > 0$ and let $\hat{S} = SRI(\delta, \Delta, \theta)$ be the output of Algorithm SRI applied to an environment in $\mathcal{E}(\Delta, \theta, \sigma, N, K, d)$. Let $\tau_{SRI}$ be the number of samples used by the SRI routine to compute $\hat{S}$. with probability higher than $1 - \delta$, it holds that $\hat{S}$ contains exactly one arm by group.*

*Moreover, there exist two universal constant $c$ and $c'$ (independent of all the parameters) such that almost surely, we have*

$$\tau_{SRI} \leqslant c\frac{1}{\theta}\log\left(\frac{K}{\delta}\right) + c'\frac{\sigma^2}{\Delta^2}\frac{1}{\theta}\log\left(\frac{K}{\delta}\right)\left[\log(K) + \sqrt{d} + \log\log\left(\frac{1}{\theta\delta}\right)\right] \ . \tag{25}$$

*Also, the expected budget satisfies*

$$\mathbb{E}_{\nu}[\tau_{SRI}] \leqslant c\frac{\log(K)}{\theta} + c'\frac{\sigma^2}{\Delta^2}\left[\frac{\log(K)}{\theta}\log\left(\frac{1}{\theta\delta}\right) + \frac{\log(K)}{\theta} + \sqrt{dK\frac{\log(K)}{\theta}\log\left(\frac{K}{\delta}\right)}\right] \ . \tag{26}$$

**Lemma 26 (Analysis of ADC)** *Let $\nu$ be an environment in $\mathcal{E}(\Delta, \theta, \sigma, N, K, d)$. Let $S$ be a set of $K$ arms containing exactly one arm belonging to each of the $K$ groups. Let $\hat{G} = ADC(\delta, \Delta, S)$ be the output of the ACD routine, and $\tau_{ADC}$ be the budget of ADC, i.e., the number of samples used to compute $\hat{G}$. First, with probability larger than $1 - \delta$, $\hat{G}$ is a perfect clustering, that is*

$$\mathbb{P}_{ADC,\nu}(\hat{G} \sim G^*) \geqslant 1 - \delta \ .$$

*Second, there exists a universal constant $c$ such that*

$$\tau_{ADC} \leqslant 2N + c\frac{\sigma^2}{\Delta^2}N\log\left(\frac{N}{\delta}\right) + c\frac{\sigma^2}{\Delta^2}\sqrt{dKN\log\left(\frac{N}{\delta}\right)} \ . \tag{27}$$

## C.1. Analysis of the SRI subroutine

In this section, we prove theorem 25. We organize the proof in several steps. As a warm-up, we discuss the intuition behind the SRI routine in Appendix C.1. We also further explanation, along with notation. Then, we provide guarantees on SRI which holds even if the parameters $\Delta$, $\theta$ used to calibrate SRI are larger than the trues parameters $\Delta_*$, $\theta_*$. We bound the probability that SRI would reject a good candidate for being a representative, or add a bad one to the set $S$. Then, we prove Lemma 25 by proving its correction and bounding its budget. The proofs of some technical lemmas are postponed to Appendix C.2. Finally, we establish Lemma 26 in Appendix C.3.

STEP 1: EXPLANATION AND NOTATION.

In this section, we fix $\Delta > 0$, and $\theta > 0$ the two parameters used in the design of the SRI routine. We also fix $\delta > 0$ and $\sigma > 0$. We consider then the algorithm $SRI = SRI(\delta, \Delta, \theta)$, where the parameters of the algorithm $U$, $(n_s)_s$, $n_{\max}$ and $r$ are computed with $\sigma$, $\Delta$, $\theta$ and $\delta$, using the expressions from Remark 27. We denote by $\mathbb{P}_\nu$ for the probability induced by $SRI(\delta, \Delta, \theta)$ and an environment $\nu$.

Let $\nu$ be an environment with a hidden partition $G^* = G_1^*, \ldots, G_K^*$ and the centers of the groups $\mu(1), \ldots, \mu(K)$, with $\sigma$-subGaussian noises Assumption 1. We associate to $G^*$ the labels $(k(a))_{a \in [N]}$ such that the mean of $a$ is $\mu_a = \mu(k(a))$ and $a \in G_{k(a)}^*$. We recall that $\Delta_*$ denotes the minimal gap of $\nu$ and $\theta_*$ is the proportion of arms in the smallest group. We want to study how $SRI = SRI(\delta, \Delta, \theta)$ behaves when it interacts with the environment $\nu$. For now, $\nu$ denotes any environment in the hidden partition model, with subGaussian noises of parameters $\sigma$ – see Assumption 1 and Assumption 3. In particular, for now, we do not assume anything about $\Delta_*$ and $\theta_*$.

In the algorithm, there are some parameters defined in (10)–(12) that we recall here.

**Remark 27** *For any $s \geqslant 1$,*

$$U = \left\lceil \frac{8}{\theta} \log\left(\frac{8K}{\delta}\right) \right\rceil \quad ,$$

$$r = \left\lceil \log_2(\log(4U/\delta)) \right\rceil \quad ,$$

$$n_s = \left\lceil c_1 \frac{\sigma^2}{\Delta^2}\left(2^s + \log(12K)\right) \right\rceil \vee \left\lceil c_2 \frac{\sigma^2}{\Delta^2}\sqrt{d(2^s + \log(6))} \right\rceil \quad ,$$

$$n_{\max} = n_r \vee \left\lceil c_3 \frac{\sigma^2}{\Delta^2}\sqrt{d}\log(2K) \right\rceil \quad ,$$

$$s_0 = r \wedge \min\{s \geqslant 1; n_s \geqslant 2\} \quad ,$$

*where the universal constants $c_1, c_2, c_3$ are respectively defined by $c_1 = 32^2 \vee 8c_{HW}$, $c_2 = 16\sqrt{c_{HW}/2} \vee 32\sqrt{2}$, and $c_3 = 32\sqrt{2}$ where $c_{HW}$ is the constant of Hanson-Wright inequality –see Appendix E. Also the maximum budget $T_{\max}$ (13) is defined as*

$$T_{\max} = 2K\left(n_{\max} + \sum_{s=s_0+1}^{r} n_s\right) + 2Un_{s_0} + 2U\sum_{s=s_0+1}^{r} \frac{n_s}{2^{s-4}} \quad ,$$

*and thereby only depends on $\theta$, $\Delta$, $K$, and $\delta$*

We refer as an epoch of the algorithm, the successive passage in the $u$ loop in the SRI routine. We introduce some notation, taking into account the dependency on $u$.

At the beginning of the $u$-th epoch, the arm $a_u$ is taken randomly and uniformly on the set $[N]$ of arms (independently of everything else). We denote by $S_u$ for the set of arms selected as representatives before the $u$-th epoch. Before the first epoch, we initialize $S_1 = \{a_0\}$. During the $u$-th epoch, the algorithm decides to add $a_u$ to $S_u$ or not by performing a sequence of tests – see Line 5 to 11 in SRI routine. If $a_u$ is added to $S_u$, it compute (Line 11) two empirical means $\hat{\mu}_{a_u}$ and $\hat{\mu}'_{a_u}$ using $2n_{\max}$ samples.

We say that

- the arm $a_u$ is bad if there exists $a \in S_u$ such that $\|\mu_{a_u} - \mu_a\| \leqslant \Delta/4$;

- the arm $a_u$ is good if for any arm $a \in S_u$ then $\|\mu_{a_u} - \mu_a\| \geqslant \Delta$.

**Remark 28** *If $\Delta \geqslant \Delta_*$, it is possible that some arms are neither good nor bad. Nonetheless, if $\Delta_* \geqslant \Delta$, then all arms from $\nu$ are good or bad. Moreover, in this case, the arm $a_u$ is bad if and only if $a_u$ is already represented in $S_u$.*

We want to add $a_u$ to $S_u$ if $a_u$ is good, but we allow the algorithm to reject some good arms if it does not affect the budget (up to a numerical constant). Anyway, we want to reject every bad arm, and reject them as quickly as possible.

For $s \geqslant 1$, we define as $\phi_s^u$ for the output of REPRESENTEDTEST$(a_u, (\hat{\mu}_b, \hat{\mu}'_b)_{b \in S_u}, \Delta, n_s)$ computed during the $u$-th epoch and for the $s$-th step . We call it the test $(u, s)$. We further write,

$$\phi_s^u := \mathbb{1}\kern-0.6em\diagup_{\left\{\min_{a \in S_u} \langle \bar{\mu}_{u,s} - \hat{\mu}_a, \bar{\mu}'_{u,s} - \hat{\mu}'_a \rangle \leqslant \frac{\Delta^2}{2}\right\}} \ ,$$

where $\bar{\mu}_{u,s}$ and $\bar{\mu}'_{u,s}$ denotes the two empirical means of arm $a_u$ computed with $2n_s$ samples, when REPRESENTEDTEST$(a_u, (\hat{\mu}_b, \hat{\mu}'_b)_{b \in S_u}, \Delta, n_s)$ is called. Remark that these empirical means are only used for the test $(u, s)$.

We start with some $s_0$ equal to $r \wedge \min\{s \geqslant 1; n_s \geqslant 2\}$ so that $n_s$ strictly increases at each iteration $s \to s + 1$. If, at some test $s_0 \leqslant s \leqslant r$, it holds that $\phi_s^u = 1$, then $a_u$ is rejected and considered as a bad arm (Line 7). If $a_u$ is rejected, we denote by $\tau_u$ for the time of rejection of $a_u$, $\tau_u := \min\{s_0 \leqslant s \leqslant r ; \phi_s^u = 1\}$. If for all $s = s_0, \ldots, r$, $\phi_s^u$ is equal to zero (False) (condition in Line 9), then $a_u$ is added to $S_u$ (Line 10) and considered as a new representative. If $a_u$ is not rejected, $\tau_u = +\infty$ by convention. The empirical mean $\hat{\mu}_{a_u}$ (resp. $\hat{\mu}'_a$) denotes the estimator of $\mu_{a_u}$ computed once and for all when $a_u$ is added to $S_u$ (Line 11), and used in every test that follows.

**Remark 29** *In REPRESENTEDTEST, the condition $\langle \bar{\mu}_{u,s} - \hat{\mu}_a, \bar{\mu}'_{u,s} - \hat{\mu}'_a \rangle \leqslant \frac{\Delta^2}{2}$ is natural, because $\mathbb{E}_\nu[\langle \bar{\mu}_{u,s} - \hat{\mu}_a, \bar{\mu}'_{u,s} - \hat{\mu}'_a \rangle] = \|\mu_{a_u} - \mu_a\|^2$, which is equal to zero if $a$ and $a_u$ are in the same group, and is larger than $\Delta_*$ else. This is a benefit of sub-sampling.*

STEP 2: CONTROL THE PROBABILITY OF REJECTING A GOOD ARM OR ADDING A BAD ARM TO $S$

In order to use the subGaussian noise assumption– see Assumption 1, we define $\epsilon_a = \frac{\sqrt{n_{\max}}}{\sigma}(\hat{\mu}_a - \mu_a)$ and $\epsilon_{u,s} = \frac{\sqrt{n_s}}{\sigma}(\bar{\mu}_{u,s} - \mu_{a_u})$ (and respectively $\epsilon'_{u,s}, \epsilon'_a$). We refer to Corollary 45 for concentration inequalities on these variables.

With these notation, we develop the statistic $\langle \bar{\mu}_{u,s} - \hat{\mu}_a, \bar{\mu}'_{u,s} - \hat{\mu}'_a \rangle$ as follows

$$
\begin{aligned}
\langle \bar{\mu}_{u,s} - \hat{\mu}_a, \bar{\mu}'_{u,s} - \hat{\mu}'_a \rangle = \|\mu_{a_u} - \mu_a\|^2 &+ \frac{\sqrt{2}\sigma}{\sqrt{n_{\max}}} \left\langle \frac{\epsilon_a + \epsilon'_a}{\sqrt{2}}, \mu_a - \mu_{a_u} \right\rangle + \frac{\sigma^2}{n_{\max}} \left\langle \epsilon_a, \epsilon'_a \right\rangle \\
&- \frac{\sigma^2}{\sqrt{n_{\max} n_s}} \left\langle \epsilon_{u,s}, \epsilon'_a \right\rangle - \frac{\sigma^2}{\sqrt{n_{\max} n_s}} \left\langle \epsilon'_{u,s}, \epsilon_a \right\rangle \qquad (28) \\
&+ \frac{\sqrt{2}\sigma}{\sqrt{n_s}} \left\langle \frac{\epsilon_{u,s} + \epsilon'_{u,s}}{\sqrt{2}}, \mu_{a_u} - \mu_a \right\rangle + \frac{\sigma^2}{n_s} \left\langle \epsilon_{u,s}, \epsilon'_{u,s} \right\rangle \quad .
\end{aligned}
$$

We will use concentration inequalities in order to control all deviations of $\langle \bar{\mu}_{u,s} - \hat{\mu}_a, \bar{\mu}'_{u,s} - \hat{\mu}'_a \rangle$ around its mean $\|\mu_{a_u} - \mu_a\|^2$.

**Remark 30** *In order to estimate the means of the representatives added to S, we compute once and for all $(\hat{\mu}_a, \hat{\mu}'_a)$ when arm $a$ is added to S (Line 11 of the SRI routine. It implies that the test statistics $(\phi_s^u)_{s,u}$ are not independent. This is why we condition on the event $\mathcal{Y}$ defined below, which controls once and for all the deviation of the random variables $\epsilon_a$ and $\epsilon'_a$).*

We define $\mathcal{Y}$ as the event:

$$
\begin{aligned}
\mathcal{Y} = &\left\{ \forall a \in \hat{S}, \forall k \in [K] \setminus \{k(a)\}, \left| \left\langle \frac{\epsilon_a + \epsilon'_a}{\sqrt{2}}, \frac{\mu_a - \mu(k)}{\|\mu_a - \mu(k)\|} \right\rangle \right| \leqslant \frac{1}{16} \sqrt{\frac{\Delta^2}{2\sigma^2}} \sqrt{n_{\max}} \right\} \\
&\bigcap \left\{ \forall a \in \hat{S}, |\langle \epsilon_a, \epsilon'_a \rangle| \leqslant \frac{1}{16} \frac{\Delta^2}{\sigma^2} n_{\max} \right\} \qquad (29) \\
&\bigcap \left\{ \forall a \in \hat{S}, \ \|\epsilon'_a\|^2 \vee \|\epsilon_a\|^2 - \mathbb{E}[\|\epsilon_a\|^2] \leqslant c_{HW} \log(12K/\delta) \vee \sqrt{c_{HW} d \log(12K/\delta)} \right\} \ ,
\end{aligned}
$$

where $c_{HW}$ is the universal constant from Hanson-Wright inequality (Lemma 44).

**Lemma 31** *For any environment $\nu$, we have,*

$$
\mathbb{P}_\nu(\mathcal{Y}) \geqslant 1 - \delta/4 \ .
$$

We leave the proof of this technical lemma to Appendix C.2; it is a consequence of the concentration of subGaussian random variables, in particular Hanson-Wright inequality (Lemmas 44 and Corollary 45).

We now give an auxiliary lemma that will be used in the rest of the proof as an elementary brick. For every test $(u, s)$, we define the event $\mathcal{Z}_{u,s}$ as

$$
\begin{aligned}
\mathcal{Z}_{u,s} = &\left\{ \exists k \in [K] \setminus \{k(a_u)\} \ ; \left| \left\langle \frac{\epsilon_{u,s} + \epsilon'_{u,s}}{\sqrt{2}}, \frac{\mu_{a_u} - \mu(k)}{\|\mu_{a_u} - \mu(k)\|} \right\rangle \right| \geqslant \frac{1}{16} \frac{\Delta}{\sigma} \sqrt{\frac{n_s}{2}} \right\} \\
&\bigcup \left\{ |\langle \epsilon_{u,s}, \epsilon'_{u,s} \rangle| \geqslant \frac{1}{16} \frac{\Delta^2}{\sigma^2} n_s \right\} \qquad (30) \\
&\bigcup \left\{ \exists a \in \hat{S} \ ; |\langle \epsilon_{u,s}, \rho'_a \rangle| + |\langle \epsilon'_{u,s}, \rho_a \rangle| \geqslant \frac{1}{4} \frac{\Delta^2}{\sigma^2} \sqrt{\frac{n_s n_{\max}}{4d + c_{HW} l \vee \sqrt{c_{HW} dl}}} \right\} \ ,
\end{aligned}
$$

with $l = \log(12K/\delta)$ and $c_{HW}$ is the constant from Lemma 44.

**Lemma 32** *The sequence of events $(\mathcal{Z}_{u,s})_{u \geqslant 1, s \geqslant s_0}$ satisfies four properties.*

1. *Conditionally on the random directions $(\rho_a, \rho'_a)_a := \left( \frac{\epsilon_a}{\|\epsilon_a\|}, \frac{\epsilon'_a}{\|\epsilon'_a\|} \right)_a$, with $a \in \hat{S}$ the events $\mathcal{Z}_{u,s}$ are independent (for all test $(u,s)$).*

2. *For all $u \geqslant 1$ and $\forall s_0 \leqslant s \leqslant r$, the inclusion $\mathcal{Y} \cap \{a_u \text{ is good and } \phi_s^u = 1\} \subset \mathcal{Z}_{u,s}$ holds.*

3. *For all $u \geqslant 1$ and $\forall s_0 \leqslant s \leqslant r$, the inclusion $\mathcal{Y} \cap \{a_u \text{ is bad and } \phi_s^u = 0\} \subset \mathcal{Z}_{u,s}$ also holds.*

4. *Finally, we have $\forall u \geqslant 1$ and $\forall s_0 \leqslant s \leqslant r$, $\mathbb{P}_\nu(\mathcal{Z}_{u,s}) \leqslant \exp(-2^s)$.*

This results is important to prove that the SRI routine actually rejects bad arms and add good arms to S. We recall that $\phi_s^u = 1$ implies that the test $(u, s)$ would reject $a_u$.
**Proof** [Sketch of proof] The terminology bad and good was introduced in the previous paragraph. Let $u \geqslant 1$ and $s_0 \leqslant s \leqslant r$. The variables $(\epsilon_{u,s}, \epsilon'_{u,s})$ are mutually independent (for any test $(u, s)$, and the event $\mathcal{Z}_{u,s}$ is measurable with respect to $\left( (\rho_a, \rho'_a)_{a \in \hat{S}}, \epsilon_{u,s}, \epsilon'_{u,s} \right)$, the first point of Lemma 32 is clear.

The construction of the event $\mathcal{Z}_{u,s}$ follows from the decomposition in Equation (28). We notice that if the event $\mathcal{Y}$ holds, the estimation of all centers $(\mu_a, \mu'_a)$ for $a$ in $\hat{S}$ are concentrated around the true centers. The points two and three follow from this observation. Moreover, the deviation of $\bar{\mu}_{u,s}$ around $\mu_{a_u}$ are subGaussian, the point 4 will follow from subGaussian concentration inequalities. We postpone the proof of this result to Appendix C.2. ∎

Now, in the next lemma, we prove that $r$ is large enough to ensure that, within the procedure, every bad arm is rejected with large probability.

**Lemma 33** *Recall that $r = \lceil \log_2(\log(4U/\delta)) \rceil$. If $\mathcal{Y}$ holds, then, with probability higher than $1 - \delta/4$, we do not add bad arms to S, i.e.*

$$\mathbb{P}_\nu \left( \{ \exists a, b \in \hat{S}, \|\mu_a - \mu_b\| \leqslant \Delta/4 \} \cap \mathcal{Y} \right) \leqslant \frac{\delta}{4} \ .$$

**Proof** Within the procedure, the algorithm picks at most $U$ arms (without counting $a_0$) – see Line 3 in SRI routine. If there exists $a, b \in \hat{S}$ such that $\|\mu_a - \mu_b\| \leqslant \Delta/4$, it means that there exists an epoch $1 \leqslant u \leqslant U$, where the last test statistic $\phi_r^u$ is equal to zero, although $a_u$ is bad. If the events $\mathcal{Y}$ holds, using the third point of Lemma 32, the event $\mathcal{Z}_{u,r}$ holds.

In terms of probability, with a simple union bound, we have

$$\mathbb{P}_\nu \left( \{ \exists a, b \in \hat{S}, \|\mu_a - \mu_b\| \leqslant \Delta/4 \} \cap \mathcal{Y} \right) \leqslant \mathbb{P}_\nu ( \{ \exists 1 \leqslant u \leqslant U, a_u \text{ is bad}, \phi_r^u = 0 \} \cap \mathcal{Y} )$$

$$\leqslant \sum_{u=1}^{U} \mathbb{P}_\nu(\mathcal{Z}_{u,r}) \ .$$

We recall that the probability of $\mathcal{Z}_{u,r}$ is smaller than $\exp(-2^r)$ and we conclude with the expression of $r$.

$$\mathbb{P}_\nu \left( \{ \exists a, b \in \hat{S}, \|\mu_a - \mu_b\| \leqslant \Delta/4 \} \cap \mathcal{Y} \right) \leqslant U \exp(-2^r) \leqslant \frac{\delta}{4} \ .$$

∎

STEP 3: SRI IS $\delta$-PAC

For all epochs $u \geqslant 1$ and $s_0 \leqslant s \leqslant r$, we denote $H_{s,u} := \sum_{v=1}^{u-1} \mathbb{1}_{\{s \leqslant \tau_v \leqslant r\}}$ as the number of arms that are rejected with a time of rejection larger than $s$ within the epochs $1, \ldots, u-1$. We highlight that $H_{s_0,u}$ is the total number of arms rejected before epoch $u$.

Now, we define $M = \inf\{u \geqslant 1; |S_u| = K \text{ or } u > U \text{ or Budget} > T_{\max}\}$ as the stopping time (i.e. the number of epochs) of SRI. It corresponds to the number of arms taken randomly from $[N]$, (namely $a_0, \ldots, a_{M-1}$) to build the set $\hat{S}$. When $M$ is reached, SRI outputs $\hat{S} = S_M$, whether or not it contains $K$ arms.

We will prove that, with probability high than $1 - \delta$, we have , on $\mathcal{E}(\Delta, \theta, \sigma, N, K, d)$, we will have $|S_M| = K$ and $S_M$ contains each representant of each cluster.

Now, assume that $\theta_* \geqslant \theta$ and $\Delta_* \geqslant \Delta$. We use Appendix C.1 to prove that the SRI routine outputs a set with exactly one arm by group when it interacts with the environment $\nu \in \mathcal{E}(\Delta, \theta, \sigma, N, K, d)$.

First, we define

$$\mathcal{X} := \bigcap_{s=s_0+1}^{r} \{H_{s,M} < \frac{1}{2^{s-4}}U\} \ . \tag{31}$$

The definition (13) of $T_{\max}$ ensures that, on the event $\mathcal{X}$, the stopping condition of the SRI routine reduces to the condition $\{|S_u| = K\} \cup \{u > U\}$ and then $M = (U+1) \wedge \min\{u \geqslant 1; |S_u| = K\}$. It turns out that the event $\mathcal{X}$ has a large probability when it is intersected with $\mathcal{Y}$.

**Lemma 34** *Let $1 + s_0 \leqslant s \leqslant r$ and recall that $H_{s,M} = \#\{u \in [|1; M-1|], s \leqslant \tau_u \leqslant r\}$. It holds that*

$$\mathbb{P}_\nu \left( \{H_{s,M} \geqslant \frac{1}{2^{s-4}}U\} \cap \mathcal{Y} \right) \leqslant \exp(-U/2) \ . \tag{32}$$

*This implies that*

$$\mathbb{P}_\nu(\mathcal{Y} \cap \mathcal{X}^c) \leqslant \frac{\delta}{8} \ .$$

**Proof** [Proof of Lemma 34] We start with the first statement of Lemma 34, take $s$ such that $s_0 < s \leqslant r$.

If $s = 1, 2$ or $3$, the inequality is trivial because $H_{s,M} \leqslant U$, we assume that $s > 3 \vee s_0$. Recall that $\nu \in \mathcal{E}(\Delta, \theta, \sigma, N, K, d)$, so that all arms are either good or bad. By definition of $\tau_u$, if $s \leqslant \tau_u \leqslant r$, it means that at some test $t \in [s, r]$, $\phi_t^u = 1$ but $\phi_t^u = 0$ for $t < s$. Moreover, each arm is either good or bad because $\Delta_* \geqslant \Delta$. The following inclusion holds then,

$$\{s \leqslant \tau_u \leqslant r\} \cap \mathcal{Y} = (\{s \leqslant \tau_u \leqslant r\} \cap \{a_u \text{ is good }\} \cap \mathcal{Y}) \bigsqcup (\{s \leqslant \tau_u \leqslant r\} \cap \{a_u \text{ is bad }\} \cap \mathcal{Y})$$

$$\subset (\cup_{s \leqslant t \leqslant r}\{\phi_t^u = 1\} \cap \{a_u \text{ is good}\} \cap \mathcal{Y}) \bigsqcup (\{\phi_{s-1}^u = 0\} \cap \{a_u \text{ is bad}\} \cap \mathcal{Y}) \ .$$

We use the points 2 of Lemma 32 to get the inclusion $\{\phi_t^u = 1\} \cap \{a_u \text{ is good}\} \cap \mathcal{Y} \subset \mathcal{Z}_{u,t}$ valid for any $t \in [s, r]$. Using the point 3 of the same lemma with $s - 1 \geqslant s_0$, we have also $\{\phi_{s-1}^u = 0\} \cap \{a_u \text{ is bad}\} \cap \mathcal{Y} \subset \mathcal{Z}_{u,s-1}$. Then,

$$\{s \leqslant \tau_u \leqslant r\} \cap \mathcal{Y} \subset \bigcup_{s-1 \leqslant t \leqslant r} \mathcal{Z}_{u,t} \ , \tag{33}$$

and we recall that the events $(\bigcup_{s-1\leqslant t\leqslant r}\mathcal{Z}_{u,t})_{u\geqslant 1}$ are independent according to Lemma 32 (if we condition on the random directions $(\rho_a, rho'_a)_a$. Now, we use a union bound on $t$ and the bound $\mathbb{P}_\nu(\mathcal{Z}_{u,t}) \leqslant \exp(-2^t)$ valid for any $t \in [s_0, r]$, and we have,

$$\mathbb{P}_\nu\left(\bigcup_{s-1\leqslant t\leqslant r}\mathcal{Z}_{u,t}\right) \leqslant \exp(-2^{s-2}) \ .$$

From the inequality $H_{s,M} = \sum_{u=1}^{M-1}\mathbb{K}_{\{s\leqslant\tau_u\leqslant r\}} \leqslant \sum_{u=1}^{U}\mathbb{K}_{\{s\leqslant\tau_u\leqslant r\}}$, we deduce that $H_{s,M}\mathbb{K}_\mathcal{y}$ is stochastically dominated by $\mathcal{B}(U, q_s)$ where $q_s := \exp(-2^{s-2})$.

We use Chernoff bound with $\alpha \geqslant \sqrt{q_s}$, we have

$$\mathbb{P}_\nu\left(H_{s,M}\mathbb{K}_\mathcal{y} \geqslant (1+\alpha/q_s)q_sU\right) \leqslant \left[\frac{e^{\alpha/q_s}}{(1+\alpha/q_s)^{1+\alpha/q_s}}\right]^{q_sU}$$
$$= \exp\left[\alpha U\left(1 - \log(1+\alpha/q_s)(1+q_s/\alpha)\right)\right]$$
$$= (1+\alpha/q_s)^{-\alpha U/2}\exp[\alpha U(1-\log(1+\alpha/q_s)/2) - Uq_s\log(1+\alpha/q_s)] \ .$$

As $\alpha \geqslant \sqrt{q_s}$ and $s \geqslant 4$, we have $\frac{\alpha}{q_s} \geqslant \frac{1}{\sqrt{q_s}} = \exp(2^{s-3}) \geqslant e^2 - 1$ and then $1 - \log(1 + \alpha/q_s)/2 \leqslant 0$. It follows that

$$\mathbb{P}_\nu\left(H_{s,M}\mathbb{K}_\mathcal{y} \geqslant 2\alpha U\right) \leqslant (1+\alpha/q_s)^{-\alpha U/2}$$
$$\leqslant (1+1/\sqrt{q_s})^{-\alpha U/2}$$
$$\leqslant \exp\left(-\frac{\alpha U}{2}2^{s-3}\right) = \exp(-\alpha U 2^{s-4}) \ .$$

Finally, taking $\alpha = 1/2^{s-3} \geqslant \exp(-2^{s-3}) = \sqrt{q_s}$, we deduce the first result of Lemma 34

$$\mathbb{P}_\nu\left(\{H_{s,M} \geqslant \frac{1}{2^{s-4}}U\} \cap \mathcal{Y}\right) \leqslant \mathbb{P}_\nu\left(H_{s,M}\mathbb{K}_\mathcal{y} \geqslant \frac{1}{2^{s-4}}U\right) \leqslant \exp(-U/2) \ .$$

Directly,

$$\mathcal{Y} \cap \mathcal{X}^c = \bigcup_{s=1}^{r}\mathcal{Y} \cap \{H_{s,M} \geqslant \frac{1}{2^{s-4}}U\} \ ,$$

then, we use the first part of the lemma and a union bound,

$$\mathbb{P}\left(\mathcal{Y} \cap \mathcal{X}^c\right) \leqslant r\exp(-U/2) \leqslant \delta/8 \ ,$$

where we conclude with the expression of $U \geqslant \frac{8}{\theta}\log\left(\frac{8K}{\delta}\right) \geqslant 2\log\left(\frac{8r}{\delta}\right)$. The last inequality follows from the expression of $r$.

∎

Now, we study the probability of adding $K$ arms to $S$, before reaching the maximum number of epochs $U$.

**Lemma 35** *Recall that $\hat{S}$ denotes the output of SRI, consider a group $G_k^*$, it holds that*

$$\mathbb{P}_\nu(\{\hat{S} \cap G_k^* = \emptyset\} \cap \{\forall a, b \in \hat{S}, \mu_a \neq \mu_b\} \cap \mathcal{Y} \cap \mathcal{X}) \leqslant 2 \exp\left(-\frac{U\theta}{8}\right) \ .$$

**Proof** [Proof of Lemma 35] Let $k \in [K]$, we study the event $\{\hat{S} \cap G_k^* = \emptyset\}$, event where the group $G_k^*$ is not represented in $\hat{S}$ (the output of the SRI routine). We use here the assumption that the groups are nonempty. If $\hat{S}$ contains arms from different groups but no arm from the group $G_k^*$, it implies that $|\hat{S}| < K$ and then, if the event $\mathcal{X}$ also holds, the algorithm has passed every epochs from $u = 1$ to $U$, i.e., $M = U + 1$. In particular, the algorithm rejected every arm from $G_k^* \cap \{a_1, \ldots, a_U\}$. We have the inclusion between events,

$$\{\hat{S} \cap G_k^* = \emptyset\} \cap \{\forall a \neq b \in \hat{S}, \mu_a \neq \mu_b\} \cap \mathcal{X} \subset \bigcap_{u \in B_k} \{a_u \text{ rejected}\} \ ,$$

where $B_k := G_k^* \cap \{a_1, \ldots, a_U\}$ and denote $X_k := |B_k|$. As $\{a_1, \ldots, a_U\}$ are i.i.d and uniform on $[N]$ then $X_k$ is binomial with parameters $U$ and $\theta_k = |G_k|/N \geqslant \theta_* \geqslant \theta$. Using Hoeffding's bound and taking $\alpha \in (0, 1)$ to be specified later, it holds that

$$\mathbb{P}_\nu(X_k \leqslant \theta U(1 - \alpha)) \leqslant \exp(-2\alpha^2 U\theta) \ . \tag{34}$$

Then, as $\Delta_* \geqslant \Delta$, the arms in $G_k \cap \{a_1, \ldots, a_U\}$ are good until one of them is added to $S$. In particular, if none are added to $S$, they are all good. We then have

$$\{|\hat{S}| \cap G_k^* = \emptyset\} \cap \{\forall a \neq b \in \hat{S}, \mu_a \neq \mu_b\} \cap \mathcal{Y} \cap \mathcal{X} \subset \bigcap_{u \in B_k} \{a_u \text{ good and rejected}\} \cap \mathcal{Y} \ .$$

If $a_u$ is rejected, it means that $\phi_s^u = 1$ for some $s_0 \leqslant s \leqslant r$, we then have the inclusion $\{a_u \text{ is good and rejected}\} \subset \bigcup_{s \geqslant s_0} \{a_u \text{ is good and } \phi_s^u = 1\}$. According to the second point of Lemma 32, we have $\{a_u \text{ is good and rejected}\} \cap \mathcal{Y} \subset \cup_{s \geqslant s_0} \mathcal{Z}_{u,s}$. We denote $\mathcal{Z}_u = \cup_{s \geqslant s_0} \mathcal{Z}_{u,s}$.

In terms of probability, we have $\mathbb{P}_\nu\left(\cap_{u \in B_k} \{a_u \text{ rejected}\} \cap \mathcal{Y}\right) \leqslant \mathbb{P}_\nu\left(\cap_{u \in B_k} \mathcal{Z}_u\right)$.

Then, we also have $\mathbb{P}_\nu(\mathcal{Z}_{u,s}) \leqslant \exp(-2^s)$ for any $s \in [s_0, r]$, and with a union bound, $\mathbb{P}_\nu(\mathcal{Z}_u) \leqslant 1/2$. Moreover, with the first point of Lemma 32, we deduce that the events $(\mathcal{Z}_u)_u$ are independent (if we condition on $\rho_a, \rho_a'$).

If $X_k > \theta U(1 - \alpha)$ and using the independence of the events $(\mathcal{Z}_u)_u$, we have

$$\mathbb{P}_\nu\left(\{X_k > \theta U(1 - \alpha)\} \cap \left(\bigcap_{u \in B_k} \mathcal{Z}_u\right)\right) \leqslant \left(\frac{1}{2}\right)^{\theta U(1-\alpha)} \ .$$

Finally, with Equation (34), we have

$$\mathbb{P}_\nu(\{\hat{S} \cap G_k^* = \emptyset\} \cap \{\forall a \neq b \in \hat{S}, \mu_a \neq \mu_b\} \cap \mathcal{Y} \cap \mathcal{X})$$

$$\leqslant \mathbb{P}_\nu\left(\{X_k > \theta U(1 - \alpha)\} \cap \left(\bigcap_{u \in B_k} \mathcal{Z}_u\right)\right) + \mathbb{P}_\nu(X_k \leqslant \theta U(1 - \alpha))$$

$$\leqslant \exp\left(-\log(2)\theta U(1 - \alpha)\right) + \exp\left(-2\alpha^2 U\theta\right)$$

$$\leqslant 2 \exp(-U\theta/8) \ .$$

In the last line, we took $\alpha = 1/2$. ∎

We now have all the tools that we need to prove that $SRI(\delta, \Delta, \theta)$ is $\delta$-PAC on $\mathcal{E}(\Delta, \theta, \sigma, N, K, d)$ for the representative identification problem, which means that with probability higher than $1 - \delta$, SRI outputs a set of $K$ representatives for environments that are $\mathcal{E}(\Delta, \theta, \sigma, N, K, d)$.

**Proof** [Proof of the first statement of Lemma 25.] recall that in this subsection, $\nu \in \mathcal{E}(\Delta, \theta, \sigma, N, K, d)$.

As a direct consequence of Lemma 33, we know that with high probability, $\hat{S}$ does not contain two arms from the same group.

$$\mathbb{P}_\nu \left( \{ \exists a, b \in \hat{S}, \mu_a = \mu_b \} \cap \mathcal{Y} \right) \leqslant \mathbb{P}_\nu \left( \{ \exists a, b \in \hat{S}, \|\mu_a - \mu_b\| \leqslant \Delta/4 \} \cap \mathcal{Y} \right) \leqslant \frac{\delta}{4} \ .$$

Now, with Lemma 35, for all $k \in [K]$,

$$\mathbb{P}_\nu \left( \{ \hat{S} \cap G_k^* = \emptyset \} \cap \{ \forall a \neq b \in \hat{S}, \mu_a \neq \mu_b \} \cap \mathcal{Y} \cap \mathcal{X} \right) \leqslant 2 \exp(-U\theta/8) \ .$$

Now, we recall that $U \geqslant \frac{8}{\theta} \log \left( \frac{8K}{\delta} \right)$ and then $\exp(-U\theta/8) \leqslant \delta/8K$. If the set $\hat{S}$ contains strictly less than $K$ arms then at least one group is not represented. With a union bound on $k \in [K]$,

$$\mathbb{P}_\nu \left( \{ |\hat{S}| < K \} \cap \{ \forall a \neq b \in \hat{S}, \mu_a \neq \mu_b \} \cap \mathcal{Y} \cap \mathcal{X} \right) \leqslant 2K \exp(-U\theta/8) \leqslant \delta/4 \ .$$

Finally, together with Lemmas 31 and 34, we conclude that

$$
\begin{aligned}
\mathbb{P}_\nu \left( \forall k \in [K], \exists! a \in \hat{S} ; \mu_a = \mu(k) \right) \geq\ & 1 - \mathbb{P}_\nu(\mathcal{Y}^c) - \mathbb{P}_\nu(\mathcal{Y} \cap \mathcal{X}^c) \\
& - \mathbb{P}_\nu \left( |\hat{S}| < K \} \cap \{ \forall a \neq b \in \hat{S}, \mu_a \neq \mu_b \} \cap \mathcal{Y} \cap \mathcal{X} \right) \\
& - \mathbb{P}_\nu \left( \{ \exists a, b \in \hat{S}, \mu_a = \mu_b \} \cap \mathcal{Y} \right) \\
\geq\ & 1 - \delta \ .
\end{aligned}
$$

In summary, we have proved that SRI is $\delta$-PAC on $\mathcal{E}(\Delta, \theta, \sigma, N, K, d)$ for the Representatives identification problem. ∎

STEP 4: UPPER BOUND ON THE BUDGET $\tau_{SRI}$

We establish here the bounds on the budget $\tau_{SRI}$ of Lemma 25.

**Explanation on $T_{\max}$**

By definition (13) of $T_{\max}$, we know that, almost surely, the total budget $\tau_{SRI}$ satisfies:

$$\tau_{SRI} \leq T_{\max} = 2K \left( n_{\max} + \sum_{s=s_0+1}^{r} n_s \right) + 2U n_{s_0} + 2U \sum_{s=s_0+1}^{r} \frac{n_s}{2^{s-4}} \ . \tag{35}$$

Although plugging the values of $n_{\max}$, $n_s$, $U$, and $s_0$, will lead to (25), we start by gently describing the budget of SRI in order to give intuition on the definition of $T_{\max}$.

To analyze the budget, we divide the budget in two parts, $\tau_{SRI} = \tau_{SRI}^a + \tau_{SRI}^r$, where $\tau_{SRI}^a$ is the number of samples used for arms that are added to $\hat{S}$ and $\tau_{SRI}^r$ is the number of samples uses for arms that are rejected.

First, we study $\tau_{SRI}^a$. From the algorithm, the arms that are selected in $\hat{S}$ are arms that pass successfully all the tests, and after the tests, they are sampled again $2n_{\max}$ times. In total, each arm in $\hat{S}$ is sampled $2n_{\max} + \sum_{s=s_0}^r 2n_s$, the factor 2 comes from the fact that we compute two empirical means at each time. We have then

$$\tau_{SRI}^a = 2|\hat{S}|n_{\max} + (|\hat{S}| - 1) \sum_{s=s_0}^r 2n_s \leqslant 2K \left( n_{\max} + \sum_{s=s_0+1}^r n_s \right) + 2(|\hat{S}| - 1)n_{s_0} \ . \quad (36)$$

Now, consider the the budget spent for arms that are ultimately rejected during the procedure. For $s_0 \leqslant s \leqslant r$ , we defined previously $H_{s,M} = \sum_{u=1}^{M-1} \mathbb{1}_{\{s \leqslant \tau_u \leqslant r\}}$, as the number of arms rejected after at least $s$ tests in the procedure. The algorithm outputs $\hat{S}$ after $M - 1$ epochs. In particular, $H_{s_0,M} = M - |\hat{S}|$. Besides, if the candidate $a_u$ is rejected, it is sampled $\sum_{s=s_0}^{\tau_u} 2n_s$ times. This leads us to the equality

$$\tau_{SRI}^r = \sum_{u=1}^{M-1} \sum_{s=s_0}^{\tau_u} 2n_s \mathbb{1}_{\{a_u \text{ is rejected }\}} = \sum_{s=s_0}^r 2H_{s,M}n_s = 2\left( M - |\hat{S}| \right) n_{s_0} + 2 \sum_{s=s_0+1}^r H_{s,M}n_s \ . \quad (37)$$

This justifies the definition $T_{\max}$ (13), as under the large probability event $\mathcal{X}$, $\tau_{SRI} \leqslant T_{\max}$ is directly implied by (36) and (37).

**Upper bound on $T_{\max}$**

In order to upper bound $T_{\max}$ (13), we need the following lemmas, whose proofs are postponed to Appendix C.2. These lemmas are direct consequences of the expressions of $n_s$, $n_{\max}$, $U$ and $r$ from equations (10),(12).

**Lemma 36** *Using the explicit expression of $r$, $n_s$, $n_{max}$ and $s_0$ from Remark 27, we have, up to a universal constant $c$, the inequality*

$$K \left( n_{\max} + \sum_{s=s_0+1}^r n_s \right) \leqslant K + c\frac{\sigma^2}{\Delta^2} \left[ \frac{\log(K)}{\theta} \log(K/\delta) + \sqrt{dK \frac{\log(K)}{\theta} \log(K/\delta)} \right]$$

$$\leqslant K + c\frac{\sigma^2}{\Delta^2} \frac{1}{\theta} \log\left( \frac{K}{\delta} \right) \left[ \log(K) + \sqrt{d} \right] \ .$$

**Lemma 37** *Using the explicit expression of $r$, $U$, $n_s$ and $s_0$ from Remark 27, we have*

$$U \sum_{s=s_0+1}^r \frac{n_s}{2^{s-4}} \leqslant c\frac{\sigma^2}{\Delta^2} \frac{1}{\theta} \log\left( \frac{K}{\delta} \right) \left[ \log(K) + \sqrt{d} + \log\log\left( \frac{1}{\theta\delta} \right) \right] \ ,$$

*where $c$ is a numerical constant.*

Now, by combining (35) with Lemmas 36 and 37, and bounding $Un_{s_0}$ by $U + A$ where $A$ is the right side of Lemma 37, we conclude that

$$\tau_{SRI} \leq T_{\max} \leqslant 2(U + K) + c\frac{\sigma^2}{\Delta^2} \frac{1}{\theta} \log\left( \frac{K}{\delta} \right) \left[ \log(K) + \sqrt{d} + \log\log\left( \frac{1}{\theta\delta} \right) \right] \ , \quad (38)$$

where $c$ is a numerical constant. We have proved (25).

**Upper bound on $\mathbb{E}[\tau_{SRI}]$**

We now upper bound the expectation of $\tau_{SRI}$. For that purpose, we now assume that $\nu \in \mathcal{E}(\Delta, \theta, \sigma, N, K, d)$. We will prove (26).

With the same decomposition on the budget $\tau_{SRI} = \tau_{SRI}^a + \tau_{SRI}^r$, and by linearity of the expectation, we deduce from (36) and (37) that

$$
\begin{aligned}
\mathbb{E}_\nu[\tau_{SRI}] &= \mathbb{E}_\nu[\tau_{SRI}\mathbb{1}_{\mathcal{Y}^c}] + \mathbb{E}_\nu[\tau_{SRI}\mathbb{1}_{\mathcal{Y}}] \\
&\leq T_{\max}\delta + 2n_{s_0}\mathbb{E}_\nu\left[(M-1)\mathbb{1}_{\mathcal{Y}}\right] + \sum_{s=s_0+1}^{r} 2n_s\mathbb{E}_\nu\left[H_{s,M}\mathbb{1}_{\mathcal{Y}}\right] \\
&\quad + 2\mathbb{E}_\nu[|\hat{S}|\mathbb{1}_{\mathcal{Y}}]\left(n_{\max} + \sum_{s=s_0+1}^{r} n_s\right) \\
&\leq T_{\max}\delta + 2K\left(n_{\max} + \sum_{s=s_0+1}^{r} n_s\right) + 2n_{s_0}\mathbb{E}_\nu\left[(M-1)\mathbb{1}_{\mathcal{Y}}\right] + \sum_{s=s_0+1}^{r} 2n_s\mathbb{E}_\nu\left[H_{s,M}\mathbb{1}_{\mathcal{Y}}\right] .
\end{aligned}
$$
(39)

Let us focus on the terms $\mathbb{E}_\nu\left[(M-1)\mathbb{1}_{\mathcal{Y}}\right]$ and $\mathbb{E}_\nu\left[H_{s,M}\mathbb{1}_{\mathcal{Y}}\right]$.

Recall that $H_{s,M} = \sum_{u=1}^{M-1}\mathbb{1}_{\{s\leqslant\tau_u\leqslant r\}}$. We also recall Equation (33), valid for $s > s_0$, and which is a consequence of Lemma 32 and the fact that $\Delta_* \geqslant \Delta$,

$$
\{s \leqslant \tau_u \leqslant r\} \cap \mathcal{Y} \subset \bigcup_{s-1\leqslant t\leqslant r} \mathcal{Z}_{u,t} .
$$

We have then,

$$
\mathbb{E}_\nu\left[H_{s,M}\mathbb{1}_{\mathcal{Y}}\right] = \mathbb{E}_\nu\left[\sum_{u=1}^{M-1}\mathbb{1}_{\{\{s\leqslant\tau_u\leqslant r\}\cap\mathcal{Y}\}}\right] \leqslant \mathbb{E}_\nu\left[\sum_{u=1}^{M-1}\mathbb{1}_{\{\bigcup_{s-1\leqslant t\leqslant r}\mathcal{Z}_{u,t}\}}\right] .
$$

We can use Wald's equation. Indeed, if we condition on the direction of the estimated centers $(\rho_a, \rho'_a)$, the random variables $\mathbb{1}_{\{\bigcup_{s-1\leqslant t\leqslant r}\mathcal{Z}_{u,t}\}}$ are independent and identically distributed for $u = 1, \ldots, U$. Moreover $\mathbb{P}_\nu(\bigcup_{s-1\leqslant t\leqslant r}\mathcal{Z}_{u,t}) \leqslant \exp(-2^{s-4})$ thanks to Lemma 32. We observe that $M$ is a stopping time with respect to the filtration naturally associated to the sequence of epochs, and the sequence $(\bigcup_{s-1\leqslant t\leqslant r}\mathcal{Z}_{u,t})_u$ is adapted to this filtration. With Wald's equation, we deduce that

$$
\mathbb{E}_\nu\left[H_{s,M}\mathbb{1}_{\mathcal{Y}}\right] \leqslant \mathbb{E}_\nu[(M-1)\mathbb{1}_{\mathcal{Y}}]\exp(-2^{s-4}) .
$$
(40)

Hence, we conclude that

$$
\sum_{s=s_0+1}^{r} 2n_s\mathbb{E}_\nu\left[H_{s,M}\mathbb{1}_{\{\mathcal{Y}\}}\right] \leqslant \sum_{s=s_0+1}^{r} 2n_s\mathbb{E}_\nu[(M-1)\mathbb{1}_{\mathcal{Y}}]\exp(-2^{s-4}) .
$$
(41)

It remains to bound $\mathbb{E}_\nu[(M-1)\mathbb{1}_{\mathcal{Y}}]$. We will bound stochastically $M-1$ by a sum of geometric random variables. We recall that $S_u$ is the state of the set of representatives at the beginning of the epoch $u$ and $\hat{S} = S_M$. We define for $k \in [K-1]$, $M_k = \sum_{u=1}^{M-1}\mathbb{1}_{\{|S_u|=k\}}$. The number $M_k$ is the

number of epochs necessary to add the $(k+1)$-th arm to $S$. Once $|S_u| = K$, the algorithm stops, so that

$$M - 1 = \sum_{k=1}^{K-1} M_k \ .$$

Fix now $k \in [K-1]$. If $|\hat{S}| < k$, we have $M_k = 0$. We assume that $|\hat{S}| \geqslant k$, and we condition on $S^{(k)}$ and $\mathcal{Y}$, the set containing the $k$ first arms that were added to $\hat{S}$. Let $u$ such that $|S_u| = k$. Thanks to the second point of Lemma 32, it holds that

$$\{a_u \text{ is good }\} \cap \mathcal{Y} \cap \left(\cap_{s \in [s_0, r]} \mathcal{Z}_{u,s}^c\right) \subset \cap_{s \in [s_0, r]}\{\phi_s^u = 0\} = \{a_u \text{ is added to } S\} \ .$$

Then, the events $\{a_u \text{ is good }\}$, $\mathcal{Y}$ and $\cap_{s \in [r]} \mathcal{Z}_{u,s}^c$ are independent. Moreover, $\mathbb{P}_\nu(a_u \text{ is good }) \geqslant (K-k)\theta_* \geqslant (K-k)\theta$ because it remains at least $(K-k)$ groups not represented in $S^{(k)}$ and all these groups have a proportion larger than $\theta_*$. We also have thanks to Lemma 31 and Lemma 32, $\mathbb{P}_\nu(\mathcal{Y}) \geqslant 1 - \delta/4$ and $\mathbb{P}_\nu(\cap_{s \in [r]} \mathcal{Z}_{u,s}^c) \geqslant 1/2$.

Conditionally on $S_u = S^{(k)}$, and on the estimated centers of the representatives in $S^{(k)}$, the event $\{a_u \text{ is added to } S\}$ are independent and of probability larger than $(1 - \delta/4)(K-k)\theta/2$. Then, $M_k$ is stochastically dominated by a geometric random variable of parameter $(K-k)\theta/4$. Finally, $\mathbb{E}_\nu[M_k \mathbb{1}_\mathcal{Y}] \leqslant \frac{4}{(K-k)\theta}$ and

$$\mathbb{E}_\nu[(M-1)\mathbb{1}_\mathcal{Y}] = \sum_{k=1}^{K-1} \mathbb{E}_\nu[M_k \mathbb{1}_\mathcal{Y}] \leqslant \sum_{k=1}^{K-1} \frac{4}{(K-k)\theta} = \frac{4}{\theta} \sum_{k=1}^{K-1} \frac{1}{k} \leqslant \frac{4}{\theta}(1 + \log(K)) \ . \quad (42)$$

Now, Equation (41) becomes

$$\sum_{s=s_0+1}^{r} 2n_s \mathbb{E}_\nu\left[H_{s,M}\mathbb{1}_\mathcal{Y}\right] \leqslant \sum_{s=s_0+1}^{r} 2n_s \left[\frac{4}{\theta}(1 + \log(K)) \exp(-2^{s-4})\right] \ .$$

We bound the previous expression, using the same computation as Lemma 37, and we state the bound as a lemma proved later.

**Lemma 38** *Using the explicit expression of $r$, $U$, $n_s$, $n_{max}$ and $s_0$ from Remark 27, we have*

$$\sum_{s=s_0}^{r} n_s \left[\frac{4}{\theta} \cdot \frac{1 + \log(K)}{\exp(2^{s-4})}\right] \leqslant c \frac{\sigma^2}{\Delta^2} \frac{\log(K)}{\theta} \left[\log(K) + \sqrt{d}\right] \ ,$$

*where $c$ is a universal constant.*

We come back to Equation (39), using Equations (41),(42), we have

$$\begin{aligned}
\mathbb{E}_\nu[\tau_{SRI}] \leqslant & T_{\max}\delta + 2K\left(n_{\max} + \sum_{s=s_0+1}^{r} n_s\right) \\
& + \frac{8n_{s_0}}{\theta}(1 + \log(K)) + \sum_{s=s_0+1}^{r} 2n_s \left[\frac{4}{\theta}(1 + \log(K)) \exp(-2^{s-4})\right] \ .
\end{aligned}$$

Now, if we define $A' = \frac{\sigma^2}{\Delta^2} \left[ \frac{\log(K)}{\theta} \log(1/(\theta\delta)) + \frac{\log(K)}{\theta}\sqrt{d} + \sqrt{dK\frac{\log(K)}{\theta}\log(K/\delta)} \right]$, Lemma 36 and Lemma 38 implies that we can choose $c$ large enough (and universal) such that

$$\sum_{s=s_0+1}^{r} 2n_s \left[ \frac{4}{\theta}(1 + \log(K))\exp(-2^{s-4}) \right] + 2K\left( n_{\max} + \sum_{s=s_0+1}^{r} n_s \right) \leqslant 2K + cA' \ .$$

By definition of $n_{s_0}$, we can also see that

$$\frac{8n_{s_0}}{\theta}(1 + \log(K)) \leqslant 16\frac{\log(K)}{\theta} + cA' \ .$$

Finally, it follows from (38) and $\delta\log(1/\delta) \leq 1$ that

$$T_{\max}\delta \leq c'\frac{\log(K)}{\theta} + cA' \ .$$

In summary, we have the desired bound in expectation,

$$\mathbb{E}_\nu[\tau_{SRI}] \leqslant c'\frac{\log(K)}{\theta} + c\frac{\sigma^2}{\Delta^2} \left[ \frac{\log(K)}{\theta}\log(1/(\theta\delta)) + \frac{\log(K)}{\theta}\sqrt{d} + \frac{\log^2(K)}{\theta} \right] \ .$$

## C.2. Proofs of technical lemmas

**Proof [Proof of Lemma 31]** We want to prove that $\mathbb{P}_\nu(\mathcal{Y}^c) \leqslant \delta/4$, where $\mathcal{Y}^c$ is equal to

$$\mathcal{Y}^c = \left\{ \exists a \in \hat{S}, \exists k \in [K] \setminus \{k(a)\}, \left| \left\langle \frac{\epsilon_a + \epsilon_a'}{\sqrt{2}}, \frac{\mu_a - \mu(k)}{\|\mu_a - \mu(k)\|} \right\rangle \right| > \frac{1}{16}\sqrt{\frac{\Delta^2}{2\sigma^2}}\sqrt{n_{\max}} \right\}$$

$$\bigcup \left\{ \exists a \in \hat{S}, |\langle \epsilon_a, \epsilon_a' \rangle| > \frac{1}{16}\frac{\Delta^2}{\sigma^2}n_{\max} \right\}$$

$$\bigcup \left\{ \exists a \in \hat{S}, \ \|\epsilon_a'\|^2 \vee \|\epsilon_a\|^2 - \mathbb{E}[\|\epsilon_a\|^2] > c_{HW}\log(12K/\delta) \vee \sqrt{c_{HW}d\log(12K/\delta)} \right\} \ .$$

From the definition of $n_s$ and $n_{\max}$ as given in (11), (12), it holds that $n_{\max} \geqslant n_r$, with

$$n_s \geqslant c_1\frac{\sigma^2}{\Delta^2}\left(2^s + \log(12K)\right) \vee c_2\frac{\sigma^2}{\Delta^2}\sqrt{d(2^s + \log(6))} \ .$$

With the definition of $r$ in (10), we have also $2^r \geqslant \log(4U/\delta)$ and $U \geqslant 8K\log(K/\delta)$. We prove that, for $c_1 = 32^2 \vee 8c_{HW}$ and $c_2 = 16\sqrt{c_{HW}/2} \vee c_3$, then $n_{\max}$ is large enough to ensure Lemma 31 (the value of $c_3$ will be useful later).

We start with a union bound on $\hat{S} \times [K]$, where $|\hat{S}| \leqslant K$. We have

$$\mathbb{P}_\nu\left( \exists a \in \hat{S}, \exists k \in [K] \setminus \{k(a)\}, \left| \left\langle \frac{\epsilon_a + \epsilon_a'}{\sqrt{2}}, \frac{\mu_a - \mu(k)}{\|\mu_a - \mu(k)\|} \right\rangle \right| > \sqrt{\frac{n_{\max}\Delta^2}{2 \cdot 16^2\sigma^2}} \right)$$

$$\leqslant K^2 \mathbb{P}_\nu\left( \left| \left\langle \frac{\epsilon_a + \epsilon_a'}{\sqrt{2}}, \frac{\mu_a - \mu(k)}{\|\mu_a - \mu(k)\|} \right\rangle \right| > \sqrt{\frac{n_{\max}\Delta^2}{2 \cdot 16^2\sigma^2}} \right) \ .$$

From the assumption on the noise (Assumption 1), it is easy to see that $\left\langle \frac{\epsilon_a + \epsilon_a'}{\sqrt{2}}, \frac{\mu_a - \mu(k)}{\|\mu_a - \mu(k)\|} \right\rangle$ is 1-subGaussian, proceeding as in the same proof of Corollary 45. With the standard concentration inequality Lemma 42 for subGaussian variables, we have

$$\mathbb{P}_\nu \left( \left| \left\langle \frac{\epsilon_a + \epsilon_a'}{\sqrt{2}}, \frac{\mu_a - \mu(k)}{\|\mu_a - \mu(k)\|} \right\rangle \right| > \sqrt{\frac{n_{\max}\Delta^2}{2 \cdot 16^2 \sigma^2}} \right) \leqslant 2 \exp\left( -\frac{n_{max}\Delta^2}{(32\sigma)^2} \right) \ ,$$

Now, $c_1 \geqslant 32^2$ and $2^r \geqslant \log(4K/\delta)$ so that $n_{\max} \geqslant c_1 \frac{\sigma^2}{\Delta^2}(2^r + \log(12K)) \geqslant 32^2 \frac{\sigma^2}{\Delta^2} \log\left( \frac{48K^2}{\delta} \right)$. Finally, with the union bouund above, we have

$$\mathbb{P}_\nu \left( \exists a \in \hat{S}, \exists k \in [K] \setminus \{k(a)\}, \ \left| \left\langle \frac{\epsilon_a + \epsilon_a'}{\sqrt{2}}, \frac{\mu_a - \mu(k)}{\|\mu_a - \mu(k)\|} \right\rangle \right| > \sqrt{\frac{n_{\max}\Delta^2}{2 \cdot 16^2 \sigma^2}} \right) \leqslant \frac{\delta}{24} \ . \quad (43)$$

Now, as proved in Corollary 45, Hanson Wright inequality imply a bound for $\langle \epsilon_a, \epsilon_a' \rangle$ and then with the constant $c_{HW}$ from Lemma 44, we have:

$$\mathbb{P}_\nu \left( \exists a \in \hat{S}, , |\langle \epsilon_a, \epsilon_a' \rangle| > \frac{\Delta^2 n_{\max}}{16\sigma^2} \right) \leqslant 2K \exp\left( -\frac{2}{c_{HW}} \left( \frac{\Delta^2 n_{max}}{16\sigma^2} \wedge \frac{\Delta^4 n_{\max}^2}{16^2 \sigma^4 d} \right) \right) \leqslant \frac{\delta}{24} \ , \quad (44)$$

because the definition of $c_1$, $c_2$ and $r$, implies that

$$n_{\max} \geqslant 8 c_{HW} \frac{\sigma^2}{\Delta^2} \log\left( \frac{48K}{\delta} \right) \vee 16\sqrt{\frac{c_{HW}}{2}} \frac{\sigma^2}{\Delta^2} \sqrt{d \log\left( \frac{48K}{\delta} \right)} \ .$$

Finally, a direct application of Hanson-Wright inequality (Lemma 44) ensures that

$$\mathbb{P}_\nu \left( \exists a \in \hat{S}, \ \|\epsilon_a'\|^2 \vee \|\epsilon_a\|^2 - \mathbb{E}[\|\epsilon_a\|^2] \geq c_{HW} \log(12K/\delta) \vee \sqrt{c_{HW} d \log(12K/\delta)} \right)$$
$$\leqslant 2K \exp(-\log(12K/\delta)) \leqslant \frac{\delta}{6} \ , \quad (45)$$

This concludes the proof of Lemma 31, using a union bound and inequalities (43) to (45). ∎

### Proof [Proof of Lemma 32 ]

We recall that in this lemma, $\nu$ is an environment with minimal gap $\Delta_*$ and balancedness $\theta_*$, and $\nu$ is not necessary in $\mathcal{E}(\Delta, \theta, \sigma, N, K, d))$. Let $u \geqslant 1$ and $s \in [s_0, r]$.

The first point of the lemma is a direct consequence of the expression of $\mathscr{Z}_{u,s}$ and the mutual independence of the series of empirical means $(\bar{\mu}_{u,s}, \bar{\mu}_{u,s}')$.

Second point of Lemma 32

We assume that $a_u$ is a good arm rejected by the test $(u, s)$, which by definition of the test statistic means that for all $a \in S_u$, then $|\mu_{a_u} - \mu_a| \geqslant \Delta$, while the test statistic $\phi_s^u$ is equal to 1. It implies that there exists $a \in S_u$ such that

$$\left\langle \bar{\mu}_{u,s} - \hat{\mu}_a, \bar{\mu}_{u,s}' - \hat{\mu}_a' \right\rangle \leqslant \frac{\Delta^2}{2} \ .$$

From the decomposition of 28, and conditionally on the event $\mathcal{Y}$ 29, we have:

$$\frac{\Delta^2}{2} \geqslant \|\mu_{a_u} - \mu_a\|^2 - \frac{\Delta}{16}\|\mu_{a_u} - \mu_a\| - \frac{\Delta^2}{16}$$

$$- \frac{\sigma^2}{\sqrt{n_{\max} n_s}} \langle \epsilon_{u,s}, \rho_a' \rangle \|\epsilon_a'\| - \frac{\sigma^2}{\sqrt{n_{\max} n_s}} \langle \epsilon_{u,s}', \rho_a \rangle \|\epsilon_a\| \tag{46}$$

$$+ \frac{\sqrt{2}\sigma}{\sqrt{n_s}} \left\langle \frac{\epsilon_{u,s} + \epsilon_{u,s}'}{\sqrt{2}}, \mu_{a_u} - \mu_a \right\rangle + \frac{\sigma^2}{n_s} \langle \epsilon_{u,s}, \epsilon_{u,s}' \rangle \quad . \tag{47}$$

We also have on $\mathcal{Y}$, $\|\epsilon_a\|^2 \leqslant \mathbb{E}[\|\epsilon_a\|^2] + c_{HW} l \vee \sqrt{c_{HW} dl}$ with $l := \log\left(\frac{12K}{\sigma}\right)$. Moreover $\mathbb{E}[\|\epsilon_a\|^2] \leqslant 4d$ is a direct consequence of the subGaussian assumption (see Rigollet and Hütter, 2023, Lemma 1.4) . We have

$$(46) \geqslant -\frac{\sigma^2}{\sqrt{n_{\max} n_s}} \left[ \left| \langle \epsilon_{u,s}', \rho_a \rangle \right| + \left| \langle \epsilon_{u,s}, \rho_a' \rangle \right| \right] \sqrt{4d + c_{HW} l \vee \sqrt{c_{HW} dl}} \quad .$$

We also have directly,

$$(47) \geqslant -\sqrt{\frac{2\sigma^2 \|\mu_{a_u} - \mu_a\|^2}{n_s}} \left| \left\langle \frac{\epsilon_{u,s} + \epsilon_{u,s}'}{\sqrt{2}}, \frac{\mu_{a_u} - \mu_a}{\|\mu_{a_u} - \mu_a\|} \right\rangle \right| - \frac{\sigma^2}{n_s} \left| \langle \epsilon_{u,s}, \epsilon_{u,s}' \rangle \right| \quad .$$

We recall that $\Delta \leqslant \|\mu_{a_u} - \mu_a\|$, because $a_u$ is a good arm. Hence, we have $\|\mu_{a_u} - \mu_a\|^2 - \frac{\Delta}{16}\|\mu_{a_u} - \mu_a\| - \frac{\Delta^2}{16} - \frac{\Delta^2}{2} \geqslant \frac{3}{8}\|\mu_{a_u} - \mu_a\|^2 \geq \frac{3}{8}\Delta^2$.

From there, we state that if $a_u$ is good and $\phi_s^u = 1$ then there exists $a \in \hat{S}$ such that at least one of the these three inequalities holds:

$$\left| \langle \epsilon_{u,s}, \rho_a' \rangle \right| + \left| \langle \epsilon_{u,s}', \rho_a \rangle \right| \geqslant \frac{1}{4} \frac{\Delta^2}{\sigma^2} \sqrt{\frac{n_s n_{\max}}{4d + \sqrt{c_{HW} dl} \vee c_{HW} l}} \quad ,$$

$$\left| \langle \epsilon_{u,s}, \epsilon_{u,s}' \rangle \right| \geqslant \frac{1}{16} \frac{\Delta^2}{\sigma^2} n_s \quad ,$$

$$\left| \left\langle \frac{\epsilon_{u,s} + \epsilon_{u,s}'}{\sqrt{2}}, \frac{\mu_{a_u} - \mu_a}{\|\mu_{a_u} - \mu_a\|} \right\rangle \right| \geqslant \frac{1}{16} \frac{\Delta}{\sigma} \sqrt{\frac{n_s}{2}} \quad .$$

By definition, $\mathcal{Z}_{u,s}$ is the event where one of these three inequalities hold for some $a \in \hat{S}$, so the inclusion $\mathcal{Y} \cap \{a_u \text{ is good and } \phi_s^u = 1\} \subset \mathcal{Z}_{u,s}$ is proved.

Third point of Lemma 32

We now assume that $a_u$ is bad, but the $s$-th test accept $a_u$, i.e., $\phi_s^u = 0$. As $a_u$ is bad, there exists $a \in S_u$ such that $\|\mu_{a_u} - \mu_a\| \leqslant \Delta/4$. As $\phi_s^u = 0$, for this specific arm $a$, we have: $\langle \bar{\mu}_{u,s} - \hat{\mu}_a, \bar{\mu}_{u,s}' - \hat{\mu}_a' \rangle > \frac{\Delta^2}{2}$.

Assume that $0 < \|\mu_{a_u} - \mu_a\| \leqslant \Delta/4$, with the same computation as in the first case, we have

$$\frac{\Delta^2}{2} \leqslant \|\mu_{a_u} - \mu_a\|^2 + \frac{\Delta}{16}\|\mu_{a_u} - \mu_a\| + \frac{\Delta^2}{16}$$

$$+ \frac{\sigma^2}{\sqrt{n_{\max} n_s}} \sqrt{4d + c_{HW} l \vee \sqrt{c_{HW} dl}} \left( \left| \langle \epsilon_{u,s}, \rho_a' \rangle \right| + \left| \langle \epsilon_{u,s}', \rho_a \rangle \right| \right)$$

$$+ \sqrt{\frac{2\sigma^2 \|\mu_{a_u} - \mu_a\|^2}{n_s}} \left| \left\langle \frac{\epsilon_{u,s} + \epsilon_{u,s}'}{\sqrt{2}}, \frac{\mu_{a_u} - \mu_a}{\|\mu_{a_u} - \mu_a\|} \right\rangle \right| + \frac{\sigma^2}{n_s} \left| \langle \epsilon_{u,s}, \epsilon_{u,s}' \rangle \right| \quad .$$

In the last line, we upper bound $\|\mu_{a_u} - \mu_a\|$ by $\Delta/4$. Now, consider the constant terms, $\frac{\Delta^2}{2} - \|\mu_{a_u} - \mu_a\|^2 - \frac{\Delta}{16}\|\mu_{a_u} - \mu_a\| - \frac{\Delta^2}{16} \geqslant \frac{\Delta^2}{2} - \frac{\Delta^2}{16} - \frac{\Delta^2}{64} - \frac{\Delta^2}{16} \geqslant \Delta^2\left(\frac{1}{4} + \frac{1}{16} + \frac{1}{4\cdot16}\right)$. As above, we deduce that at least one of the three inequalities defining $\mathcal{Z}_{u,s}$ holds. If $\|\mu_{a_u} - \mu_a\| = 0$, there are simply less terms in the equality. This proves the inclusion $\mathcal{Y} \cap \{a_u \text{ is bad and } \phi_s^u = 0\} \subset \mathcal{Z}_{u,s}$.

Probability of $\mathcal{Z}_{u,s}$

We prove now that the probability of $\mathcal{Z}_{u,s}$ decrease exponentially fast with $s$. Fix $s_0 \leqslant s \leqslant r$.

We recall the expression of $n_s$, and $n_{\max}$

$$n_s \geqslant c_1 \frac{\sigma^2}{\Delta^2}\left(2^s + \log(12K)\right) \vee c_2 \frac{\sigma^2}{\Delta^2}\sqrt{d(2^s + \log(6))} \ ,$$

where $c_1 = 32^2 \vee 8c_{HW}$, $c_2 = 16\sqrt{c_{HW}/2} \vee 32\sqrt{2}$, $c_3 = 32\sqrt{2}$, and $n_{\max} \geqslant n_r \vee c_3 \frac{\sigma^2}{\Delta^2}\sqrt{d}\log(2K)$.

Let $k \in [K]$ such that $\mu_{a_u} \neq \mu(k)$, as a consequence of Assumption 1, the one-dimensional variable $\left\langle \frac{\epsilon_{u,s} + \epsilon'_{u,s}}{\sqrt{2}}, \frac{\mu_{a_u} - \mu(k)}{\|\mu_{a_u} - \mu(k)\|}\right\rangle$ is 1-subGaussian. With standard concentration of subGaussian variables (Lemma 42), we have

$$\mathbb{P}_\nu\left(\left|\left\langle \frac{\epsilon_{u,s} + \epsilon'_{u,s}}{\sqrt{2}}, \frac{\mu_{a_u} - \mu(k)}{\|\mu_{a_u} - \mu(k)\|}\right\rangle\right| \geqslant \frac{1}{16}\frac{\Delta}{\sigma}\sqrt{\frac{n_s}{2}}\right) \leqslant 2\exp\left(-\frac{\Delta^2}{\sigma^2}\frac{n_s}{32^2}\right) \leqslant \frac{1}{3K}\exp(-2^s) \ ,$$

because $n_s \geqslant 32^2 \frac{\sigma^2}{\Delta^2}(2^s + \log(6K))$.

Now, with a union bound over $k \in [K]$, it holds that

$$\mathbb{P}\left(\exists k \in [K]\setminus\{k(a_u)\} \ ; \left|\left\langle \frac{\epsilon_{u,s} + \epsilon'_{u,s}}{\sqrt{2}}, \frac{\mu_{a_u} - \mu(k)}{\|\mu_{a_u} - \mu(k)\|}\right\rangle\right| \geqslant \frac{1}{16}\frac{\Delta}{\sigma}\sqrt{\frac{n_s}{2}}\right) \leqslant \frac{1}{3}\exp(-2^s) \ , \quad (48)$$

Then, $\langle \epsilon_{u,s}, \epsilon'_{u,s}\rangle$ is the inner product of two independent vectors, for which the assumptions from Corollary 45 holds. We use this corollary of Hanson-Wright inequality, and obtain

$$\mathbb{P}_\nu\left(\left|\langle \epsilon_{u,s}, \epsilon'_{u,s}\rangle\right| \geqslant \frac{\Delta^2}{\sigma^2}\frac{n_s}{16}\right) \leqslant 2\exp\left(-\frac{2}{c_{HW}}\left(\frac{\Delta^2}{\sigma^2}\frac{n_s}{16} \wedge \frac{1}{d}\frac{\Delta^4}{\sigma^4}\frac{n_s^2}{16^2}\right)\right) \leqslant \frac{1}{3}\exp(-2^s) \ , \quad (49)$$

where the last inequality follows from the definition of $n_s$ (and $c_1, c_2$) where $n_s \geqslant 8c_{HW}\frac{\sigma^2}{\Delta^2}(2^s + \log(6)) \vee 16\sqrt{\frac{c_{HW}}{2}}\frac{\sigma^2}{\Delta^2}\sqrt{d(2^s + \log(6))}$.

Finally, we want to upper bound the probability that the cross term between $\epsilon_a$ and $\epsilon_{u,s}$ is too large. By conditioning with respect to the random variables $(\rho_a, \rho'_a)_a$, we consider these variables as constants. We start with a union bound and the inequality $a + b \leqslant 2a \vee b$.

$$\mathbb{P}_\nu\left(\exists a \in \hat{S} \ ; \left|\langle \epsilon_{u,s}, \rho'_a\rangle\right| + \left|\langle \epsilon'_{u,s}, \rho_a\rangle\right| \geqslant \frac{1}{4}\frac{\Delta^2}{\sigma^2}\sqrt{\frac{n_s n_{\max}}{4d + \sqrt{c_{HW}dl} \vee c_{HW}l}}\right)$$
$$\leqslant 2K\mathbb{P}_\nu\left(\left|\langle \epsilon_{u,s}, \rho\rangle\right| \geqslant \frac{1}{8}\frac{\Delta^2}{\sigma^2}\sqrt{\frac{n_s n_{\max}}{4d + \sqrt{c_{HW}dl} \vee c_{HW}l}}\right) \ ,$$

with $\rho$ of norm 1. Then $\langle \epsilon_{u,s}, \rho\rangle$ is a 1-dimensional subGaussian random variable. We use therefore the concentration inequality in Lemma 42, and we state that

$$\mathbb{P}_\nu \left( \exists a \in \hat{S} \; ; \left| \langle \epsilon_{u,s}, \rho_a' \rangle \right| + \left| \langle \epsilon_{u,s}', \rho_a \rangle \right| \geqslant \frac{1}{4} \frac{\Delta^2}{\sigma^2} \sqrt{\frac{n_s n_{\max}}{4d + \sqrt{c_{HW} dl} \vee c_{HW} l}} \right)$$

$$\leqslant 4K \exp \left( -\frac{1}{2 \cdot 8^2} \frac{\Delta^4}{\sigma^4} \frac{n_s n_{\max}}{4d + \sqrt{c_{HW} dl} \vee c_{HW} l} \right)$$

$$\leqslant 4K \exp \left( -\frac{1}{16^2} \frac{\Delta^4}{\sigma^4} \frac{n_s n_{\max}}{4d \vee \sqrt{c_{HW} dl} \vee c_{HW} l} \right) \; ,$$

in the last line, we use again the inequality $a + b \leqslant 2a \vee b$.

Now, we need to bound this last expression by $\frac{1}{3} \exp(-2^s)$ by using the definition of $n_s$ and $n_{\max}$ We recall that $l = \log(12K/\delta) \leqslant 2^r$.

Now, with our choice for $c_1$ and $c_2$, it holds that $n_s \geqslant 32^2 \frac{\sigma^2}{\delta^2}(2^s + \log(12K))$ and $n_{\max} \geqslant c_{HW} l \vee \sqrt{c_{HW} dl}$, so that

$$n_s n_{\max} \geqslant 16^2 \frac{\sigma^4}{\Delta^2} (\sqrt{c_{HW} dl} \vee c_{HW} l)(2^s + \log(12K)) \; .$$

We finally use the assumption that $c_2 \geqslant 32\sqrt{2}$ and $c_3 = 32\sqrt{2}$, so that $n_s \geqslant 16\sqrt{2} \frac{\sigma^2}{\Delta^2} \sqrt{(4d)(2^s + \log(6))}$, $n_{\max} \geqslant 16\sqrt{2} \frac{\sigma^2}{\Delta^2} \sqrt{4d} \sqrt{2^s + \log(6)}$ and $n_{\max} \geqslant 16\sqrt{2} \frac{\sigma^2}{\Delta^2} \sqrt{4d} \log(2K)$. Then, with the inequality $a \vee b \geqslant (a + b)/2$, we have

$$n_s n_{\max} \geqslant 16^2 \frac{\sigma^4}{\Delta^2} (4d)(2^s + \log(12K)) \; .$$

We combine these lower bound on $n_s n_{\max}$ to deduce that

$$n_s n_{\max} \geqslant 16^2 \frac{\sigma^4}{\Delta^2} (4d \vee \sqrt{c_{HW} dl} \vee c_{HW} l)(2^s + \log(12K)) \; .$$

This allows us to conclude that

$$\mathbb{P}_\nu \left( \exists a \in \hat{S} \; ; \left| \langle \epsilon_{u,s}, \rho_a' \rangle \right| + \left| \langle \epsilon_{u,s}', \rho_a \rangle \right| \geqslant \frac{1}{4} \frac{\Delta^2}{\sigma^2} \sqrt{\frac{n_s n_{\max}}{4d + \sqrt{c_{HW} dl} \vee c_{HW} l}} \right)$$

$$\leqslant 4K \exp \left( -\frac{1}{16^2} \frac{\Delta^4}{\sigma^4} \frac{n_s n_{\max}}{4d \vee \sqrt{c_{HW} dl} \vee c_{HW} l} \right) \leqslant \frac{1}{3} \exp(-2^s) \; . \tag{50}$$

We finish the proof with a union bound, gathering the inequalities (48) to (50),

$$\mathbb{P}_\nu(\mathcal{Z}_{u,s}) \leqslant \frac{1}{3} \exp(-2^s) + \frac{1}{3} \exp(-2^s) + \frac{1}{3} \exp(-2^s) = \exp(-2^s) \; .$$

∎

**Proof** [ **Proof of Lemma 36**]

Throughout the proofs of Lemmas 36, 37, 38, $c$ is a universal constant changing from one line to another. Also, we use that, by the definition of $s_0$ and $n_s$, it turns out, that if $s > s_0$ then

$$n_s \leqslant 2c_1 \frac{\sigma^2}{\Delta^2} (2^s + \log(12K)) \vee 2c_2 \frac{\sigma^2}{\Delta^2} \sqrt{d(2^s + \log(6))} \; . \tag{51}$$

We now bound $K\left(\sum_{s=s_0+1}^r n_s + n_{\max}\right)$. Relying on the expression of $n_s$ above, and the sums $\sum_{s=1}^r 2^s \leqslant 2^{r+1}$ and $\sum_{s=1}^r \sqrt{2}^s \leqslant \sqrt{2}^{r+1}(1+\sqrt{2})$, we deduce that

$$K \sum_{s=s_0+1}^r n_s \leqslant 2c_1 \frac{\sigma^2}{\Delta^2} K(2^{r+1} + \log(12K)r) \bigvee 2c_2 \frac{\sigma^2}{\Delta^2} K\sqrt{d}\left(\sqrt{2^r}(2+\sqrt{2}) + \sqrt{\log(6)}r\right) \quad .$$

Now, from the expression of $r$ and $U$, we have $2^r \leqslant 2\log(4U/\delta) \leqslant c\log(1/(\theta\delta))$. It leads to the bound

$$K \sum_{s=s_0+1}^r n_s \leqslant c\frac{\sigma^2}{\Delta^2}\left[K\log(1/(\theta\delta)) + K\log(K)\log\log(1/(\theta\delta)) + K\sqrt{d\log(1/(\theta\delta))}\right] \quad .$$

As $1/\theta \geqslant K$, we have $K\log\left(\frac{1}{\theta}\right) \leqslant \frac{1}{\theta}\log(K)$, so that we can bound the term above by

$$K \sum_{s=s_0+1}^r n_s \leqslant c\frac{\sigma^2}{\Delta^2}\left[\frac{1}{\theta}\log(K/\delta) + \sqrt{dK\frac{1}{\theta}\log(K/\delta)}\right] \quad .$$

We also compute $n_{\max}$, we have $Kn_{max} \leqslant K + c\frac{\sigma^2}{\Delta^2}\left[Kn_r + K\log(K)\sqrt{d}\right]$, where $Kn_r$ is upper bounded by the same bound as $K\sum_{s=s_0+1}^r n_s$.

Finally, it implies that

$$K \sum_{s=s_0+1}^r n_s + Kn_{\max} \leqslant K + c\frac{\sigma^2}{\Delta^2}\left[\frac{\log(K)}{\theta}\log(K/\delta) + \sqrt{dK\frac{\log(K)}{\theta}\log(K/\delta)}\right] \quad .$$

The second inequality in Lemma 36 is clear, and the lemma is proved. ∎

**Proof** [ Proof of Lemma 37] With the bound on $n_s$ for $s > s_0$ from Equation (51), we simplify the terms in $2^s$ and obtain,

$$\sum_{s=s_0+1}^r \frac{U}{2^{s-4}}n_s \leqslant 2c_1\frac{16U\sigma^2}{\Delta^2}\sum_{s=s_0+1}^r\left(1 + \frac{1}{2^s}\log(12K)\right) \bigvee 2c_2\frac{16U\sigma^2}{\Delta^2}\sum_{s=s_0+1}^r\left(\frac{1}{\sqrt{2}^s}\sqrt{d} + \frac{1}{2^s}\sqrt{d}\sqrt{\log(6)}\right)$$

$$\leqslant 2c_1\frac{16U\sigma^2}{\Delta^2}\left(r + 2\log(12K)\right) \bigvee 2c_2\frac{16U\sigma^2}{\Delta^2}\left((2+\sqrt{2})\sqrt{d} + 2\sqrt{\log(6)}\sqrt{d}\right)$$

$$\leqslant c\frac{\sigma^2}{\Delta^2}U\left[\log\log\left(\frac{1}{\theta\delta}\right) + \log(K) + \sqrt{d}\right] \quad ,$$

because $\sum_{s\geqslant 1} 1/2^s \leqslant 2$ and $\sum_{s\geqslant 1} 1/\sqrt{2}^s = 2 + \sqrt{2}$. We also use in the last inequality that $\log(U/\delta) \leqslant 2\log(8/\theta\delta)$, so that $r \leqslant \log(2\log(8/\theta\delta)) \leqslant c\log\log(1/\theta\delta)$.

From the previous bound, we conclude that

$$\sum_{s=s_0+1}^r \frac{U}{2^{s-4}}n_s \leqslant c\frac{\sigma^2}{\Delta^2}U\left[\log\log\left(\frac{1}{\theta\delta}\right) + \log(K) + \sqrt{d}\right] + c\frac{\sigma^2}{\Delta^2}\sqrt{d\log(K)KU} \quad .$$

Moreover, we have by definition of $U$ (10), $U \geqslant K \log(K)$, and then

$$\frac{\sigma^2}{\Delta^2} \sqrt{d \log(K) K U} \leqslant \frac{\sigma^2}{\Delta^2} U \sqrt{d} \ .$$

Finally, using the expression of $U$ (10), we have

$$\sum_{s=s_0+1}^{r} \frac{U}{2^{s-4}} n_s \leqslant c \frac{\sigma^2}{\Delta^2} \frac{1}{\theta} \log(K/\delta) \left[ \log(K) + \sqrt{d} + \log\log\left(\frac{1}{\theta\delta}\right) \right] \ .$$

∎

**Proof** [ Proof of Lemma 38] With the same computation as in Lemma 37, we obtain

$$\sum_{s=s_0+1}^{r} n_s \left[ \frac{4}{\theta}(1 + \log(K)) \exp(-2^{s-4}) \right] \leqslant c \frac{\sigma^2}{\Delta^2} \frac{\log(K)}{\theta} \left[ \log(K) + \sqrt{d} \right] + c \frac{\sigma^2}{\Delta^2} \sqrt{dK \frac{\log(K)}{\theta} \log(K)}$$

$$\leqslant c \frac{\sigma^2}{\Delta^2} \frac{\log(K)}{\theta} \left[ \log(K) + \sqrt{d} \right] \ ,$$

where we use $K \leqslant 1/\theta$ in the last inequality. ∎

## C.3. Proof of Lemma 26

In this section, we want to prove that the subroutine ADC outputs the exact partition with probability larger than $1 - \delta$, for environments in $\mathcal{E}(\Delta, \theta, \sigma, N, K, d)$. Let $\nu$ be an environment with a minimal gap smaller than $\Delta$, following Assumptions 1 and 3. We highlight that the algorithm ADC uses $\Delta$, $\sigma$, $N$, $K$ and $d$ as parameters but not $\theta$. Let $S = \{b_1, \ldots, b_K\}$ be a set of $K$ arms containing one representative by group. The objective is to find the groups $G_1^*, \ldots, G_K^*$ up to permutation. Without loss of generality, we fix the label of the groups so that $G_k^* = \{a \in [N], \mu_a = \mu_{b_k}\}$. We denote by $k(a)$ as the corresponding label of any arm $a$ $a$ ($a \in G_{k(a)}^*$). With this convention, making an error of clustering is equivalent of making an error of labeling.

We denote $\hat{G}$ for the output of the ADC routine. The algorithm labels the arms in $S$ so that $b_k \in \hat{G}_k$ for $k \in [K]$ (see Line 7). Then, it labels each arm $a \in [N] \setminus S$ by $\hat{k}(a)$ defined (Equation (15)) by

$$\hat{k}(a) \in \underset{j=1,\ldots,K}{\operatorname{argmin}} \left\langle \hat{\mu}_a - \hat{\mu}(j), \hat{\mu}_a' - \hat{\mu}'(j) \right\rangle \ .$$

We have $\{\hat{G} \sim G^*\} = \{\exists a \in [N] \setminus S \, ; \hat{k}(a) \neq k(a)\}$.

Consider $j \in [K]$ a group and $a \in [N]$ an arm. As explained in the introduction, the statistic $\hat{d}_{a,j}^2 := \left\langle \hat{\mu}_a - \hat{\mu}(j), \hat{\mu}_a' - \hat{\mu}'(j) \right\rangle$ is a natural non-biased estimator of $\|\mu_a - \mu(j)\|^2$ where $\mu(j) = \mu_{b_j}$ is the center of $G_j^*$. In the expression of $\hat{k}(a)$, $\hat{\mu}(j)$ [resp. $\hat{\mu}'(j)$] is the empirical mean of representative $b_j$ computed with $J = \left\lceil c_4 \frac{\sigma^2}{\Delta^2} L \vee c_5 \frac{\sigma^2}{\Delta^2} \sqrt{dL \frac{N}{K}} \right\rceil$ samples –see Equation (14) and Line 6– and $L = \log(6NK/\delta)$. The random variable $\hat{\mu}_a$ [resp. $\hat{\mu}_a'$] is the empirical mean of the arm $a$ computed with $I = \left\lceil c_4 \frac{\sigma^2}{\Delta^2} L \vee c_5 \frac{\sigma^2}{\Delta^2} \sqrt{dL \frac{K}{N}} \right\rceil$ samples – see Line 10. We emphasise

that in high dimension, $J \asymp NI/K$ is much larger than $I$. We want to bound the probability of misclassification for a single arm in $[N] \setminus S$. Let $a \in [N] \setminus S$ such that $a$ belongs to the group $G^*_{k(a)}$. The misclassification probability for the arm $a$ using the classifier $\hat{k}(a)$ of Equation (15) is

$$
\begin{aligned}
\mathbb{P}_\nu(\hat{k}(a) \neq k(a)) =& \mathbb{P}_\nu\Big(\exists j = 1, \ldots, K , j \neq k(a);\ \hat{d}^2_{a,j} < \hat{d}^2_{a,k(a)}\Big) \\
\leqslant& \sum_{\substack{j \neq k(a)}}^K \mathbb{P}_\nu\Big(\hat{d}^2_{a,j} < \hat{d}^2_{a,k(a)}\Big) \ .
\end{aligned}
\tag{52}
$$

We used here a first union bound over $j \in [1; K] \setminus k(a)$, and now, we upper bound each term on the sum.

**Lemma 39** *For all $a \in [N] \setminus S$, and $j \in [K]$, if $\mu(j) \neq \mu_a$ then*

$$
\mathbb{P}_\nu\Big(\hat{d}^2_{a,j} < \hat{d}^2_{a,k(a)}\Big) \leqslant \frac{\delta}{(K-1)(N-K)} \ .
$$

This lemma easily leads to the desired result (Lemma 26) by a union bound on $a \in [N] \setminus S$. With Equation (52) and Lemma 39, we have indeed

$$
\begin{aligned}
\mathbb{P}_\nu\left(\hat{G} \not\sim G^*\right) =& \mathbb{P}_\nu\left(\exists a \in [N] \setminus [S] ; \hat{k}(a) \neq k(a)\right) \\
\leqslant& \sum_{a \in [N] \setminus S} \sum_{j \in [K] \setminus \{k(a)\}} \mathbb{P}_\nu\Big(\hat{d}^2_{a,j} < \hat{d}^2_{a,k(a)}\Big) \leqslant \delta \ .
\end{aligned}
$$

Moreover, the budget $\tau_{ADC}$ used to compute ADC is deterministic and equal to $2(N-K)I + 2KJ$ with the notation of the algorithm which leads to the second part of the lemma directly.

We have indeed the (deterministic) bound on the budget of ADC

$$
\tau_{ADC} = 2(N-K)I + 2KJ \leqslant 2N + 2c_4 \frac{\sigma^2}{\Delta^2} NL \vee 4c_5 \frac{\sigma^2}{\Delta^2}\sqrt{dKNL} \ .
$$

It remains now to prove the auxiliary lemma.

**Proof** [Proof of Lemma 39
]

Without loss of generality, we assume that $\mu_a = \mu(1)$ and consider $j = 2$. We write

$$
\hat{\mu}_a = \mu_a + \frac{\sigma}{\sqrt{I}}\varepsilon_a = \mu(1) + \frac{\sigma}{\sqrt{I}}\varepsilon_a \ ,
$$

where $\varepsilon_a := \frac{\sqrt{I}}{\sigma}(\hat{\mu}_a - \mu_a)$. We define in the same way $\varepsilon(1) := \frac{\sqrt{J}}{\sigma}(\hat{\mu}(1) - \mu(1))$ and also $\varepsilon(2), \varepsilon'_a$, $\varepsilon'(1)$ and $\varepsilon'(2)$.

From direct computation, reorganising the terms, we wright the event $\{\hat{d}^2_{a,2} < \hat{d}^2_{a,1}\}$ as

$$
\begin{aligned}
&\left\langle \hat{\mu}_a - \hat{\mu}(2), \hat{\mu}'_a - \hat{\mu}'(2)\right\rangle < \left\langle \hat{\mu}_a - \hat{\mu}(1), \hat{\mu}'_a - \hat{\mu}'(1)\right\rangle \Leftrightarrow \\
&\frac{\sqrt{2}\sigma\|\mu(1) - \mu(2)\|}{\sqrt{I}}A + \frac{\sqrt{2}\sigma\|\mu(1) - \mu(2)\|}{\sqrt{J}}B + \frac{\sqrt{2}\sigma^2}{\sqrt{IJ}}(C+D) + \frac{\sigma^2}{J}(E+F) > \|\mu(1) - \mu(2)\|^2 \ ,
\end{aligned}
\tag{53}
$$

where

$$A := - \left\langle \frac{\mu(1) - \mu(2)}{\|\mu(1) - \mu(2)\|}, \frac{\varepsilon'_a + \varepsilon_a}{\sqrt{2}} \right\rangle; \qquad C := - \left\langle \varepsilon_a, \frac{\varepsilon'(1) - \varepsilon'(2)}{\sqrt{2}} \right\rangle; \quad E := - \langle \varepsilon(2), \varepsilon'(2) \rangle;$$

$$B := - \left\langle \frac{\mu(2) - \mu(1)}{\|\mu(2) - \mu(1)\|}, \frac{\varepsilon'(2) + \varepsilon(2)}{\sqrt{2}} \right\rangle; \quad D := - \left\langle \varepsilon'_a, \frac{\varepsilon(1) - \varepsilon(2)}{\sqrt{2}} \right\rangle; \quad F := - \langle \varepsilon'(1), \varepsilon(1) \rangle \ .$$

Let us control the variation of each of these terms.

First, by Assumption 1, as in the proofs of Appendix C.1, $A$ and $B$ are subGaussian. With the concentration inequality (Lemma 42) for subGaussian (real) variables, we have

$$\mathbb{P}_\nu(A > \sqrt{2L}) \leqslant \exp(-L) \qquad \text{and} \qquad \mathbb{P}(B > \sqrt{2L}) \leqslant \exp(-L) \ .$$

For the other terms, we use Hanson-Wright inequality (Corollary 45) with $c_{HW}$ the universal constant from the lemma. The scalar products $C$, $D$, $E$ and $F$ verifies all the assumptions for Corollary 45, and for instance,

$$\mathbb{P}_\nu \left( C > \frac{c_{HW} L}{2} \vee \sqrt{c_{HW} \frac{dL}{2}} \right) \leqslant \exp(-L) \ ,$$

and we have the same bound for $D$, $E$ and $F$.

We recall the expression $L = \log\left(\frac{6NK}{\delta}\right)$ defined after Equation (14), in particular, $\exp(-L) \leqslant \frac{\delta}{6NK}$. With a union bound on these 6 errors, it holds that with probability larger than $1 - \delta/NK$ we have

$$\frac{\sqrt{2}\sigma\|\mu(1) - \mu(2)\|}{\sqrt{I}} A + \frac{\sqrt{2}\sigma\|\mu(1) - \mu(2)\|}{\sqrt{J}} B + \frac{\sqrt{2}\sigma^2}{\sqrt{IJ}}(C + D) + \frac{\sigma^2}{J}(E + F)$$

$$\leqslant \frac{\sqrt{2}\sigma\|\mu(1) - \mu(2)\|}{\sqrt{I}} \sqrt{2L} + \frac{\sqrt{2}\sigma\|\mu(1) - \mu(2)\|}{\sqrt{J}} \sqrt{2L} + \frac{2\sqrt{2}\sigma^2}{\sqrt{IJ}} \left( \frac{c_{HW} L}{2} \vee \sqrt{c_{HW} \frac{dL}{2}} \right)$$

$$+ \frac{2\sigma^2}{J} \left( \frac{c_{HW} L}{2} \vee \sqrt{c_{HW} \frac{dL}{2}} \right) \ .$$

The parameters $I,J$ are defined as

$$I = \left\lceil \frac{\sigma^2}{\Delta^2} \left( c_4 L \vee c_5 \sqrt{\frac{K}{N} dL} \right) \right\rceil; \quad J = \left\lceil \frac{\sigma^2}{\Delta^2} \left( c_4 L \vee c_5 \sqrt{\frac{N}{K} dL} \right) \right\rceil \ ,$$

with $c_4$ and $c_5$ two universal constants defined as $c_4 = 8^2 \vee 4\sqrt{2} c_{HW}$ and $c_5 = 8\sqrt{c_{HW}}$ with $c_{HW}$ the universal constant in Hanson-Wright inequality (Lemma 44). Now, each term in the last sum is smaller than $\|\mu(1) - \mu(2)\|\Delta/4$, or $\Delta^2/4$. As $\nu \in \mathcal{E}(\Delta, \theta, \sigma, N, K, d)$, we have $\Delta_* \geqslant \Delta$ and $\|\mu(1) - \mu(2)\| \geqslant \Delta$. It implies that with probability larger than $1 - \delta/NK$, it holds that

$$\frac{\sqrt{2}\sigma\|\mu(1) - \mu(2)\|}{\sqrt{I}} A + \frac{\sqrt{2}\sigma\|\mu(1) - \mu(2)\|}{\sqrt{J}} B + \frac{\sqrt{2}\sigma^2}{\sqrt{IJ}}(C + D) + \frac{\sigma^2}{J}(E + F) \leqslant \|\mu(1) - \mu(2)\|^2 \ .$$

From there, eq. (53) assures that

$$\mathbb{P}_\nu \left( \left\langle \hat{\mu}_1 - \hat{\mu}(2), \hat{\mu}'_1 - \hat{\mu}'(2) \right\rangle < \left\langle \hat{\mu}_a - \hat{\mu}(k(a)), \hat{\mu}'_a - \hat{\mu}'(k(a)) \right\rangle \right) \leqslant \frac{\delta}{KN} \ .$$

■

## Appendix D. Analysis of $ACB^*$

In this section, we prove the part of Theorem 5 pertaining to $ACB^*$. In fact, this result is a straightforward consequence of the following theorem

**Theorem 40** *Let $\delta > 0$. For any environment $\nu$, $ACB^*$ Algorithm 2 is $\delta$-PAC. There exist positive numerical constants $c$, $c'$, and $c''$ such that the following holds.*

$$\mathbb{P}_{ACB^*,\nu}\Big[\tau_{ACB^*} \leqslant cN + c'\frac{\sigma^2}{\Delta_*^2\theta_*}L_*\log\left(\frac{L_*K}{\delta}\right)\Big[\log(K) + \sqrt{d} + \log\log(L_*) + \log\log(N/\delta)\Big]$$

$$+ c'\frac{L_*}{\theta_*}\log\left(\frac{L_*K}{\delta}\right) + c''\frac{\sigma^2}{\Delta_*^2}\Big[N\log\left(N/\delta\right) + \sqrt{dNK\log\left(N/\delta\right)}\Big]\Big] \geq 1 - \delta\,, \qquad (54)$$

*where*

$$L_* := \left\lceil\log_2\left(\frac{1}{\theta_*K}\left[\left(\frac{\Delta_0^2}{\Delta_*^2} \vee 1\right)\right]\right)\right\rceil\,. \qquad (55)$$

We set the numerical constant $c_6$ in the definition (17) of $n_p'$ as

$$c_6 = 2048 \vee 64c_{HW} \vee 92\sqrt{c_{HW}}\,, \qquad (56)$$

where $c_{HW}$ is the constant arising in Hanson-Wright inequality –see Lemma 44.

### D.1. Analysis of SRI for $\Delta \leq 4\Delta_*$

We explained in Appendix C.1 how the algorithm $\hat{S} =$SRI$(\delta, \Delta, \theta)$ behaves for environments that are not in $\mathcal{E}(\Delta, \theta, \sigma, N, K, d)$. If $\Delta_* \geqslant \Delta$ then the identification of $K$ representatives goes well but, if $\Delta_* \gg \Delta$, the budget will be unnecessarily large. If $\Delta_* \leqslant \Delta$, then the set of representative $\hat{S}$ may contain less than $K$ representative. The following lemma summarizes the properties of SRI.

**Lemma 41** *Take $\nu$ an environment with a minimal gap $\Delta_*$ and a balancedness $\theta_*$. Consider $\hat{S} =$SRI$(\delta, \Delta, \theta)$ the output of the SRI routine, designed with $\Delta > 0$ and $\theta > 0$. With probability $\mathbb{P}_{SRI,\nu}$ larger than $1 - \delta$, the follwoing holds*

- *the set $\hat{S}$ does not contain two arms from the same cluster,*

- *if $\Delta_* \leqslant \Delta/4$, then $\hat{S}$ contains strictly less than $K$ arms,*

- *if $\Delta_* \geqslant \Delta$ and $\theta_* \geqslant \theta$ then $\hat{S}$ contains exactly one arm by group.*

**Proof** The first point is a consequence of Lemma 31 and Lemma 33. The third point is exactly the result of Lemma 25.

For the second point, recall that by definition, a candidate $a_u$ is bad if there exists an arm $a$ in the set $S$ such that $\|\mu_{a_u} - \mu_a\| \leqslant \Delta/4$. In Lemma 33, we prove that with probability larger than $1 - \delta$, no bad arms would be added to $S$. Moreover, if $\Delta_* \leqslant \Delta/4$, then there exists at least one group whose arms are bad during all the procedure, and hence, the second point is also a consequence of Lemma 31 and Lemma 33. ∎

### D.2. Proof of Theorem 40

D.2.1. $ACB^*$ IS $\delta$-PAC

We consider separately two cases $\Delta_* \leq \Delta_0$ and $\Delta_* > \Delta_0$. We first focus on the case where $\Delta_* \leqslant \Delta_0$.

We remind the reader that the procedure $ACB_*$ consists on a sequence of calls for SRI, with different parameters, we remind these parameters as defined in (16), (17)

$$\Delta_0^2 = \sigma^2[\log(K) + \sqrt{d} + \log\log(6N/\delta)], \qquad \delta_l = \frac{\delta}{6(l+1)^3}$$

$$\theta_{p,l} = \frac{1}{K2^{l-p}}, \quad \Delta_p = \Delta_0\sqrt{\frac{1}{2^p}}, \quad n_p' = \left\lceil c_6\frac{\sigma^2}{\Delta_p^2}\left(\log(3K^2/\delta) + \sqrt{d\log(3K^2/\delta)}\right)\right\rceil .$$

For short, we write $SRI(p,l)$ for SRI routine with parameters $\delta_l$, $\Delta_p$, and $\theta_{p,l}$. For $l \geqslant 0$ and $p = 0,\ldots,l$, we define $\mathcal{E}_{p,l}$ as the event of probability larger than $1 - \delta_l$ under $\mathbb{P}_{SRI(p,l),\nu}$ defined in Lemma 41. We write $\mathcal{E}$ for the intersection of these events.

From Lemma 41, the event $\mathcal{E}_{p,l}$ has a probability larger than $1 - \delta_l$. With a union bound, and the definition of $\delta_l$ (16), we deduce that

$$\mathbb{P}(\mathcal{E}) = \mathbb{P}\left(\bigcap_{p,l}\mathcal{E}_{p,l}\right) \geqslant 1 - \sum_{l\geqslant 1}\sum_{p=0}^{l}\delta_l = 1 - \sum_{l\geqslant 0}\frac{\delta}{6(l+1)^2} \geqslant 1 - \delta/3 .$$

We write $(l',p')$ the first value of $(l,p)$ in Algorithm 2 such that $|S_{l,p}| = K$. On the event $\mathcal{E}$, we have that $\hat{S} = S_{l',p'}$ contains exactly one arm by cluster – see again Lemma 41.

Even, if on the event $\mathcal{E}$, we know that $\Delta_* \geq \Delta_{p'}/4$ (see also Lemma 41). This lower bound on $\Delta_*$ could be used to parameterize the ADC, however, we prefer to estimate $\Delta_*$ directly in Algorithm 2 before applying the routine ADC.

Recall that $n_p' = c_6\frac{\sigma^2}{\Delta_p^2}\left(\log(3K^2/\delta) + \sqrt{d\log(3K^2/\delta)}\right)$. We use $2Kn_{p'}'$ samples to estimate $\Delta_*$–see $\widehat{\Delta}$ in Line 8 of Algorithm 2. Arguing as in the proof of Lemma 31, we deduce from the definition (56) of $c_6$, that, on the intersection of the event $\mathcal{E}$ with an event of probability higher than $1 - \delta/3$, we have

$$\frac{1}{4}\Delta_*^2 \leqslant \frac{1}{2}\hat{\Delta}^2 \leqslant \Delta_*^2$$

Since, on this event, we have $2^{-1/2}\hat{\Delta} \leq \Delta_*$, we are in position to apply Lemma 26 to $ADC(\delta/3, 2^{-1/2}\hat{\Delta}, \hat{S})$. In summary, we have proved that $ACB^*$ is $\delta$-PAC.

D.2.2. CONTROL OF THE BUDGET OF $ACB^*$

We now bound the budget of $ACB^*$ under the same event as in the previous subsection.

The key observation was proven page 22 of (Jamieson et al., 2016), taking $T_l = 2^l$, it holds that

$$\{\theta \in (0,1/K), \Delta \in (0,\Delta_0); \frac{\Delta_0^2}{K\theta\Delta^2} \leqslant 2^l\} \subset \bigcup_{p=0}^{l-1}\{(\theta,\Delta) : \theta \geqslant \theta_{p,l}, \Delta \geqslant \Delta_p\} .$$

In particular, if $2^l \geqslant \frac{\Delta_0^2}{K\theta_*\Delta_*^2}$, then, there exists $p \in [l-1]$ such that $\theta_{p,l} \leqslant \theta_*$ and $\Delta_{p,l} \leqslant \Delta_*$. From this result and from Lemma 41, we get that, on the event $\mathcal{E}$, the stopping time $l'$ satisfies $l' \leqslant L_* = \left\lceil \log_2 \left( \frac{\Delta_0^2}{\theta_* K \Delta_*^2} \right) \right\rceil$ –recall that $L_*$ is defined in (55).

We write $\tau_1$ at the total budget we have spent for computing $\widehat{S}$. Recall that the budget of the routine SRI is almost surely bounded by $T_{\max}$ –see (13) – and we upper bounded $T_{\max}$ in (38). In order to emphasize the dependency of this budget on $(\delta, \Delta, \theta)$ we write $T_{\max}(\delta, \Delta, \theta)$ in the sequel.

By (38), on the event $\mathcal{E}$, we have

$$
\tau_1 \leqslant \sum_{l=0}^{L_*} \sum_{p=0}^{l} T_{\max}(\delta_l, \Delta_p, \theta_{p,l} \vee 1/N)
$$

$$
\leqslant \sum_{l=0}^{L_*} \sum_{p=0}^{l} 2 \left( \left\lceil \frac{8}{\theta_{p,l}} \log \left( \frac{8K}{\delta_l} \right) \right\rceil + K \right) + c' \frac{\sigma^2}{\Delta_p^2} \frac{1}{\theta_{p,l}} \log \left( \frac{K}{\delta_l} \right) \left[ \log(K) + \sqrt{d} + \log\log \left( \frac{N}{\delta_l} \right) \right] \quad .
$$

We observe that, in $ACB_*$ Line 3, we use SRI with $\theta_{p,l} \vee 1/N$ because any environment has necessary a balancedness larger than $1/N$. It allows us to bound the $\log\log$-term in Equation (38) by $\log\log(N/\delta)$.

Now, by definition (17), $\theta_{p,l} = \frac{1}{K2^{l-p}}$ and $\Delta_p^2 = \frac{\Delta_0^2}{2^p}$ so that $\frac{1}{\Delta_p^2} \frac{1}{\theta_{p,l}} = \frac{K2^l}{\Delta_0^2}$ and then

$$
\tau_1 \leqslant \sum_{l=0}^{L_*} \sum_{p=0}^{l} \left( 16K \cdot 2^p \log \left( \frac{8K}{\delta_l} \right) + 2(K+1) \right)
$$

$$
+ \sum_{l=0}^{L_*} \sum_{p=0}^{l} c' \frac{\sigma^2}{\Delta_0^2} K 2^l \log \left( \frac{K}{\delta_l} \right) \left[ \log(K) + \sqrt{d} + \log\log \left( \frac{N}{\delta_{L_*}} \right) \right]
$$

$$
\leqslant 2(L_*+1)^2(K+1) + cK2^{L^*} \log \left( \frac{8K}{\delta_{L_*}} \right)
$$

$$
+ c' \frac{\sigma^2}{\Delta_0^2} K(L_*+1) 2^{L_*} \log \left( \frac{K}{\delta_{L_*}} \right) \left[ \log(K) + \sqrt{d} + \log\log \left( \frac{N}{\delta_{L_*}} \right) \right] \quad .
$$

Now, $2^{L_*} \leqslant 2 \frac{\Delta_0^2}{\theta_* K \Delta_*^2}$, so that

$$
\tau_1 \leqslant 2(L_*+1)^2(K+1) + c \frac{\Delta_0^2}{\theta_* \Delta_*^2} \log \left( 8 \frac{K(L_*+1)^3}{\delta} \right)
$$

$$
+ c'(L_*+1) \frac{\sigma^2}{\theta_* \Delta_*^2} \log \left( \frac{6K(L_*+1)^3}{\delta} \right) \left[ \log(K) + \sqrt{d} + \log\log \left( \frac{6N(L_*+1)^3}{\delta} \right) \right]
$$

$$
\leqslant cL_*^2 K + c'L_* \frac{\sigma^2}{\theta_* \Delta_*^2} \log \left( \frac{KL_*}{\delta} \right) \left[ \log(K) + \sqrt{d} + \log\log \left( \frac{NL_*}{\delta} \right) \right] \quad . \tag{57}
$$

In the last inequality, we used the expression of $\Delta_0^2$ (16) which implies that

$$
\frac{\Delta_0^2}{\theta_* \Delta_*^2} \log \left( 8 \frac{K(L_*+1)^3}{\delta} \right) \leqslant c' \frac{\sigma^2}{\theta_* \Delta_*^2} \log \left( \frac{KL_*}{\delta} \right) \left[ \log(K) + \sqrt{d} + \log\log \left( \frac{N}{\delta} \right) \right] \quad .
$$

Let us now consider the budget $\tau_2$ dedicated to the estimator of $\Delta_*$. Since $\Delta_{p'}^{-2} \leq 2^{L_*}\Delta_0^{-2}$, we deduce that

$$\tau_2 = 2Kn_{p'} \leq 2K + c\frac{\sigma^2}{\theta_*\Delta_*^2}\left(\log(3K^2/\delta) + \sqrt{d\log(3K^2/\delta)}\right) \ . \tag{58}$$

Finally, as we are working under the event $\widehat{\Delta}^2/\Delta_*^2 \in [1/2, 2]$, we deduce from Lemma 26 that the budget $\tau_3$ incurred by ADC is smaller or equal to

$$\tau_3 \leq 2N + c\frac{\sigma^2}{\Delta_*^2}N\log\left(\frac{N}{\delta}\right) + c\frac{\sigma^2}{\Delta_*^2}\sqrt{dKN\log\left(\frac{N}{\delta}\right)} \ . \tag{59}$$

The total budget is obtained by summing the bounds (57), (58), and (59).

It remains to consider the case where $\Delta_* \geq \Delta_0$. In that case, under the events of the previous subsection, the first phase of the algorithm stops at the latest as $(l, p) = (L_*, 0)$, where $L_* = \lceil\log_2(1/(\theta_*K))\rceil$. Arguing as above, we deduce that $\tau_1$ satisfies

$$\tau_1 \leq cKL_*^2 + c'L_*\frac{1}{\theta_*}\log\left(\frac{KL_*}{\delta}\right) \ . \tag{60}$$

Regarding the second step of the algorithm, we know that $p' \leq L_*$ so that $\Delta_{p'}^{-2} \leq 2^{L_*}\Delta_0^{-2} \leq \frac{\Delta_0^{-2}}{\theta_*K}$. We deduce that

$$\tau_2 \leq 2K + c\frac{1}{\theta_*}\frac{\log(3K^2/\delta) + \sqrt{d\log(3K^2/\delta)}}{[\log(K) + \sqrt{d} + \log\log(6N/\delta)]} \leq 2K + c'L_*\frac{1}{\theta_*}\log\left(\frac{KL_*}{\delta}\right) \ . \tag{61}$$

Finally, the budget $\tau_3$ is still given by (59). Gathering (60), (61), and (59) allows us to conclude.

## Appendix E. Concentration inequalities

We now give a few concentration inequalities used in the paper.

First, a consequence of the definition of $\sigma$-subGaussian random variables given in Assumption 1 is the following,

**Lemma 42** *Let $Y \in \mathbb{R}$ be subGaussian, then for all $x > 0$,*

$$\mathbb{P}(X > x) \leqslant \exp(-\frac{x^2}{2}), \text{ and } \mathbb{P}(X < -x) \leqslant \exp(-\frac{x^2}{2}) \ .$$

Here is Laurent and Massart inequality , page 1325 of (Laurent and Massart, 2000).

**Lemma 43 (Laurent & Massart)** *Let $Z \sim \chi_d^2$ a chi-square distribution, where $d \geqslant 1$ is the degree of freedom, then for any $x > 0$,*

$$\mathbb{P}(Z \geqslant d + 2\sqrt{dx} + 2x) \leqslant \exp(-x), \text{ and } \mathbb{P}(Z \leqslant d - 2\sqrt{dx}) \leqslant \exp(-x)$$

We now give the Hanson-Wright inequality for the concentration of scalar products of subGaussian random variables – see (Rudelson and Vershynin, 2013) for the proof.

**Lemma 44 (Hanson-Wright inequality)** *Let $Y$ be a $d$-dimensional vector in $\mathbb{R}^d$ with independent, centered and $1$-subGaussian components. Let $A$ be an $d \times d$ matrix. Then, there exists a constant $c_{HW}$ such that for any $x \geqslant 0$,*

$$\mathbb{P}(Y^T A Y - \mathbb{E}[Y^T A Y] > x) \leqslant \exp\left(-\frac{1}{c_{HW}}\left(\frac{x^2}{\|A\|_F^2} \wedge \frac{x}{\|A\|_{op}}\right)\right) \ ,$$

*where $\|A\|_{op}$ is the operator norm of $A$, $\|A\|_F$ is the Frobenius norm.*

We use in this paper the following corollary,

**Corollary 45** *Let $\nu_1$ and $\nu_2$ be two probability distribution, with respective expectations $\mu_1$ and $\mu_2$. We assume that there exists $\Sigma_1$ and $\Sigma_2$ two symmetric $d \times d$ matrices such that, for $a = 1, 2$, under $\nu_a$, $E = \Sigma_a^{-1/2}[X - \mu_a]$ is a vector with independent subGaussian random variables. Assume also that $\|\Sigma_1\|_{op} \leqslant \sigma^2$ and $\|\Sigma_2\|_{op} \leqslant \sigma^2$.*

*Let $m_1 \in \mathbb{N}^*$ and $m_2 \in \mathbb{N}^*$ be two integer. Consider $X_{1,1}, \ldots, X_{1,n_1}$ be i.i.d variables distributed as $\nu_1$, and $X_{2,1}, \ldots, X_{2,n_2}$ i.i.d variables distributed as $\nu_2$, independent from the observations of $a_2$.*

*If $\epsilon_1 := \frac{\sqrt{n_1}}{\sigma}\left(\frac{1}{n_1}\sum_{i=1}^{n_1} X_{1,i} - \mu_1\right)$, and $\epsilon_2 := \frac{\sqrt{n_2}}{\sigma}\left(\frac{1}{n_2}\sum_{i=1}^{n_2} X_{2,i} - \mu_2\right)$, then, for any $x \geqslant 1$,*

$$\mathbb{P}(\langle \epsilon_1, \epsilon_2 \rangle > x) \leqslant \exp\left(-\frac{2}{c_{HW}}\left(\frac{x^2}{d} \vee x\right)\right) \ .$$

*Similarly, for any $x > 0$, we have*

$$\mathbb{P}\left(\langle \epsilon_1, \epsilon_2 \rangle > \frac{c_{HW}}{2}x \vee \sqrt{\frac{c_{HW}}{2}dx}\right) \leqslant \exp\left(-x\right) \ .$$

**Proof**

Let $a = 1, 2$. We specify the rotation $\Sigma_a$ in the expression of $\epsilon_a$,

$$\epsilon_a = \frac{\sqrt{n_a}}{\sigma}\left(\frac{1}{n_a}\sum_{i=1}^{n_a} X_{a,i} - \mu_a\right) = \frac{1}{\sigma}\Sigma_a^{1/2}\frac{1}{\sqrt{N}}\sum_{t=1}^{n_a}\Sigma_a^{-1/2}[X_{a,i} - \mu_a] \ .$$

Now, by assumption on the distribution $\nu_a$, for all $i \in [n_a]$, the vector $\Sigma_a^{-1/2}[X_{a,i} - \mu_a]$ has independent and subGaussian entries. By independence of the random variables $(X_{a,1}, \ldots, X_{a,n_a})$, the vector $\frac{1}{\sqrt{N}}\sum_{t=1}^{n_a}\Sigma_a^{-1/2}[X_{a,i} - \mu_a]$ has independent entries. By independence and using the definition of subGaussian variables given in Assumption 1, $Y_a := \frac{1}{\sqrt{N}}\sum_{t=1}^{n_a}\Sigma_a^{-1/2}[X_{a,i} - \mu_a]$ is composed of independent and subGaussian entries. It holds then that

$$\langle \epsilon_1, \epsilon_2 \rangle = Y_1^T \frac{\Sigma_1^{1/2}\Sigma_2^{1/2}}{\sigma^2} Y_2 = \begin{bmatrix} Y_1 \\ Y_2 \end{bmatrix}^T S \begin{bmatrix} Y_1 \\ Y_2 \end{bmatrix} \ ,$$

where the matrix $S := \frac{1}{2}\begin{bmatrix} 0 & \frac{\Sigma_1^{1/2}\Sigma_2^{1/2}}{\sigma^2} \\ \frac{\Sigma_1^{1/2}\Sigma_2^{1/2}}{\sigma^2} & 0 \end{bmatrix}$ is a $2d \times 2d$ matrix. We can then apply Lemma 44, noticing that $\mathbb{E}[\langle \epsilon_1, \epsilon_2 \rangle] = 0$, $\|S\|_{op} = \|\Sigma_1^{1/2}\Sigma_2^{1/2}\frac{1}{\sigma^2}\|_{op}/2 \leqslant 1/2$ and $\|S\|_F^2 \leqslant d/2$. ∎

