# OpenReview forum: "Clustering with bandit feedback: breaking down the computation/information gap"
_algorithmiclearningtheory.org/ALT/2025/Conference — ALT 2025_

### Official Review · Reviewer_q6yb · 2024-11-04
**Review and Comments**

**Rating:** 6
**Confidence:** 3

**Review:**

This work consider N-armed stochastic bandit with subGaussian feedback, where the mean vectors are from a set of cardinality K. The goal is to reveal this unknown partition using a minimum budget. The authors propose a novel ACB algorithm, whose budget matches the lower bound in many regimes. Also, the authors find there is no computation-information gap in the bandit setting, as opposed to the batch setting.

Some comments and questions about the paper:
1. In Appendix C.2, the authors have shown that |S_M|=K w.p. at least 1-\delta, which is important for the algorithm to work. But this has not been mentioned in the main paper.
2. The Appendix is not well organized, for example, STEP1 to STEP4 span Appendix C.1 and C.2. Also, the authors can consider introducing the purpose of each part in Appendix at the beginning of the appendix.
3. Quoted from the abstract: "we establish that there is no computation-information gap in the bandit setting". But based on the introduction, the computation-information gap can come from whether repeated measurements are allowed, instead of "bandit vs batch".
4. Do the authors invent the estimator in Eq (9)? If not, please cite the corresponding reference.
5. Regarding Section 5, while a kmeans++ initialization is better than maximin, the maximin guarantees a 2-approximation to the global optimum of kmeans (Celebi et al., 2013). This property is important in establishing the theoretical results in Yang et al. (2024), but kmeans++ does not have this property.

After reading the rebuttal, I have adjusted the score accordingly.

**Paper Award:**

No

---

### Official Review · Reviewer_gC9r · 2024-11-07

**Rating:** 6
**Confidence:** 2

**Review:**

This paper studies clustering identification with bandit feedback. The setting is as follows: K arms – each associated with a random multi-dimensional distribution – are partitioned into $K$ clusters, with the property that two arms share the same cluster if and only if their distributions share the same average. A learner is tasked with uncovering the clustering. To do so, it is allowed to query arms and observe i.i.d. samples from their associated distribution. Given a precision $\delta$, the goal is to design an optimal learning strategy, in the sense that needs as few queries/samples as possible to uncover the correct clustering, with probability at least $1-\delta$.

In the batch version of the problem – when data are generated by a mixture of $K$ gaussians that need to be identified – it is believed that in high-dimension there is a gap between the sample complexity needed by poly-time algorithms and the information theoretical requirement.
The contribution of the authors is two-fold. First, they construct a poly-time algorithm which, in the balanced case (all groups have a similar size), outperforms the the simple batch algorithm. Second, they prove that such bound is information-theoretical optimal, meaning that there is no computation-information gap for clustering with bandit feedback in high-dimension.

The algorithm combines in a nice way many ideas, which seem to be quite standard. Although I am not an expert I enjoyed reading the paper and followed easily the intuition behind the technical part. On the negative side, I am not totally convinced by the relevance of the question: recovering clustering structure is an important task, but I am not sure how relevant is perfectly identifying all the clusters, i.e., also the ones that are extremely close. For instance, imagine that there are 3 clusters, one very far apart and two extremely close. The algorithm and the results presented in this paper depends polynomially on the distance between the two close centers, while the well separated cluster can be easily identified. This is not captured by the model, nor by the techniques.

**Paper Award:**

No

---

### Official Review · Reviewer_quSZ · 2024-11-12
**Review for the paper "Clustering with Bandit Feedback: Breaking Down the Computation/Information Gap"**

**Rating:** 7
**Confidence:** 3

**Review:**

This work explores the Clustering with Bandit Feedback Problem (CBP), where a learner engages with an \(N\)-armed stochastic bandit system providing \(d\)-dimensional sub-Gaussian feedback, in order to uncover a hidden partition of the arms into \(K\) groups. Each group shares a common mean vector for all arms within it. The primary objective for the learner is to identify this hidden partition with a minimum observation budget and with an error probability below a specified threshold, $\delta$.

The paper established a Non-Asymptotic Lower Bound on Observation Budget for the observation budget required to achieve the specified error threshold, which matches the information theoretical lower bound up to multiplicative log factors.

The main algorithm innovation is from combining the Sequential Representative Identification and Active Distance-based Classification subroutines together, being able to greatly reduce the number of observations compared to the batched sampling. This adaptive approach provides a substantial improvement in the performance, ensuring that clustering is achieved with minimal budget expenditure.

A notable finding in this paper is the demonstration that, contrary to the batch setting, there is no computation-information gap in the sequential setting.

 The paper is very well written and organized.  I examined the mathematical proofs presented in the main body. Empirical evidences are included to support the claims. Overall, this paper makes meaningful theoretical advancements in understanding the observation budget and computational requirements for clustering in the bandit feedback framework.

Therefore, I recommend this paper for acceptance.

**Paper Award:**

No

---

### Author Rebuttal · Authors · 2024-11-22

### **Rebuttal to reviewer gC9r**

We would like to thank you for your time and effort in reviewing our paper.

- **Novelty of the paper.** The main difficulties for applying bandit theory to clustering are twofold. First, we go beyond classical bandit ideas as we consider a $d$-dimensional clustering problem, where $d$ is not assumed to be constant as the budget grows, and where we focus on obtaining the right dependency in $d$. Achieving this is not standard in bandit theory, and far from being trivial: in particular in terms of lower bounds, this requires the development of lower bound techniques in an high dimensional sequential setting, which is non-standard and challenging. Second, we go beyond classical clustering ideas as a main advantage of sequential learning - as opposed to batch learning - is the possibility to sub-sample efficiently some entries of interest and avoid computational bottlenecks coming from combining information in several entries. For this reason, we are able to prove that there is no statistical-computational gap in sequential clustering, unlike what happens in batch.

- **Heterogeneity of the group distances.** In this paper, we considered the problem of exactly recovering the partition of the arms as this problem formulation matches the usual framework of the clustering literature.

   If the learner is aiming at recovering a partition of the arms where the groups are distant at least by some $\Delta$, it is quite easy to adapt the procedures $ACB$ and $ACB^*$ to  build only groups distant by at least $\Delta$. In your example, running such a procedure with $K=2$ would enable to recover this coarse-grain structure with a small and optimal budget which only depends on the distance between the distant groups. We will add a remark in our manuscript explaining how to do this extension.

  Now beyond this answer, it would of course be interesting to have an algorithm who fully adapts to all inter-groups gaps - and not just to the minimal one, or to some target distance. This is a relevant and interesting direction for future works, but it goes beyond the present work whose main aim was to disprove the existence of a statistical-computational gap in sequential clustering.

---

### Author Rebuttal · Authors · 2024-11-22

### **Rebuttal to Reviewer q6yb**

We first would like to thank you for your time and effort in reviewing our paper. We will use your insightful questions and remarks to improve the presentation.
- **1) Explanation on SRI subroutine --** Indeed, the $SRI$  Subroutine of $ACB$ and $ACB_*$ has been crafted in order to identify $K$ arms [from distinct clusters] with probability larger than $1-\delta$. This is an important step of the proof of our main theorem (Theorem 5). We did not emphasize that in the main text mainly because of page constraints. In the revision, we further emphasize this in the main text.

- **2) Appendix organization --**
    We appreciate the suggestion to reorganize the structure of the appendix. In section C.1, we provide results on SRI which are valid for any choice of the parameter $\Delta$ and $\theta$ used in the routine SRI. Then, in section C.2, we assume that the true minimal gap $\Delta_*$ is larger than $\Delta$, and we conclude that in this case, SRI identifies one representative by cluster. We will merge Appendices C.1 and C.2. Additionally, we will add a short introduction to this section to outline the purpose of each Step.

---

> ### Author Rebuttal · Authors · 2024-11-22
>
> - **3) Bandit vs batch --** In order to make it easier to compare the batch and the sequential setting, we considered in the introduction a version of a batch setting where $T$ observations are performed uniformly, so that the learner samples $T/N$ times each arm - so that if $T\geq 2N$, repeated measurements are performed.
>    Consider for simplicity the Gaussian model with known variance $\sigma^2$, as computational lower bounds are established in this setting. A sufficient statistic would be to average all observations of the same arm, so that after computing it, we would be back in the classical batch setting with noise variance shrunk by a factor $T/N$ - namely $N\sigma^2/T$ instead of $\sigma^2$. And in this model, a computation-information gap is strongly conjectured and is even established in the low-degree computational framewrok, see [1].
>     So that it is not just repeated measurements that allow us to bypass the computation-information gap in the sequential setting, but also the fact that we can allocate samples in a sequential way to well-chosen sub-samples of interesting arms. In this way, we can identify in a faster way, and without needing to solve too costly combinatorial optimization problems, interesting arms. But repeated measurements alone would not be enough, so we insist on the necessity of a sequential and active setting to bypass the computation-information gap. We find this interesting, as while sequential sampling is generally praised for being able to improve the information theoretic performances on some problems, its effect on bypassing computation-information gap has, to the best of our knowledge, not be very much studied.
>
> - **4) Estimator from Equation (9) --**  The construction of this estimator belongs to the statistical folklore for the problem of estimating the square $l_2$ norm of the mean of a random vector when the covariance structure -- see e.g.[2] for a previous occurence. In the simpler case where the covariance structure would be known, one could instead the simpler estimator from~[3].
>
> - **5)Initialization of kmeans --**    Indeed, Yang et al. use a maxi-min initialization for kmeans to have theoretical guaranties. However, kmeans++ is known to have in practice better performances, as discussed in [3]. As the purpose of Section 5 is to compare numerical performances, we used  kmeans++.

---

> > ### Author Rebuttal · Authors · 2024-11-22
> >
> > [1]. Bertrand Even, Christophe Giraud, and Nicolas Verzelen. Computation-information gap in high-dimensional clustering. In Aaron Roth Shipra Agrawal, editor, Proceedings of Thirty Seventh Conference on Learning Theory, Proceedings of Machine Learning Research. PMLR, 30 Jun – 3 Jul 2024
> >
> > [2]. Alexandra Carpentier. Testing the regularity of a smooth signal. Bernoulli 21 (1) 465 - 488, February 2015. https://doi.org/10.3150/13-BEJ575
> >
> > [3]. Collier, O., and Dalalyan, A. S. (2019). Multidimensional linear functional estimation in sparse Gaussian models and robust estimation of the mean.
> >
> > [4]. Celebi, M. E., Kingravi, H. A., et Vela, P. A. (2013). A comparative study of efficient initialization methods for the k-means clustering algorithm. Expert systems with applications, 40(1), 200-210.

---

### Meta-Review · Area_Chair_mE56 · 2024-12-06

**Recommendation:** Accept
**Confidence:** 5

**Metareview:**

The reviewers are unanimous that this is an interesting contribution to online clustering in the high-dimensional setting. I particularly appreciated the discussion about computational-statistical trade-offs in various regimes for this problem.

The authors adressed well the reviewer's questions. I recommend to accept

**Paper Award:**

No